# How Does the Task Landscape Affect MAML Performance?

## Abstract

Model-Agnostic Meta-Learning (MAML) has become increasingly popular for training models that can quickly adapt to new tasks via one or few stochastic gradient descent steps. However, the MAML objective is significantly more difficult to optimize compared to standard non-adaptive learning (NAL), and little is understood about how much MAML improves over NAL in terms of the fast adaptability of their solutions in various scenarios. We analytically address this issue in a linear regression setting consisting of a mixture of easy and hard tasks, where hardness is related to the rate that gradient descent converges on the task. Specifically, we prove that in order for MAML to achieve substantial gain over NAL, (i) there must be some discrepancy in hardness among the tasks, and (ii) the optimal solutions of the hard tasks must be closely packed with the center far from the center of the easy tasks optimal solutions. We also give numerical and analytical results suggesting that these insights apply to two-layer neural networks. Finally, we provide few-shot image classification experiments that support our insights for when MAML should be used and emphasize the importance of training MAML on hard tasks in practice.

## 1 Introduction

Large-scale learning models have achieved remarkable successes in domains such as computer vision and reinforcement learning. However, their high performance has come at the cost of requiring huge amounts of data and computational resources for training, meaning they cannot be trained from scratch every time they are deployed to solve a new task. Instead, they typically must undergo a single pre-training procedure, then be fine-tuned to solve new tasks in the wild.

*Meta-learning* aims to address this problem by extracting an inductive bias from a large set of pre-training, or meta-training, tasks that can be used to improve test-time adaptation. *Model-agnostic meta-learning* (MAML) (Finn et al., 2017), one of the most popular meta-learning frameworks, formulates this inductive bias as an initial model for a few gradient-based fine-tuning steps. Given that for the model can be fine-tuned on any meta-test task before evaluating its performance, MAML aims to learn the best initialization for fine-tuning among a set of meta-training tasks. To do so, it executes an episodic meta-training procedure in which the current initialization is adapted to specific tasks in an *inner loop*, then the initialization is updated in an *outer loop*. This procedure has led to impressive few-shot learning performance in many settings (Finn et al., 2017; Antoniou et al., 2018).

However, in some settings MAML's inner loop adaptation and the second derivative calculations resulting thereof may not justify their added computational cost compared to traditional, no-inner-loop pre-training. Multiple works have suggested that MAML's few-shot learning performance may not drop when executing the inner loop adaptation for only a fraction of the model parameters (Raghu et al., 2019; Oh et al., 2020) or, in some cases, by ignoring the inner loop altogether (Bai et al., 2020; Chen et al., 2019). Unfortunately, the underlying reasons for MAML's behavior are still not well-understood, making it difficult to determine when inner loop adaptation during meta-training yields meaningful benefit.

In this work, we investigate when and why MAML's inner loop updates provide significant gain for meta-test time adaptation. To achieve this, we focus on how the meta-training loss landscape induced by the MAML objective affects adaptation performance compared to the loss landscape induced by the classical objective without inner loop adaptation, which we term the Non-Adaptive

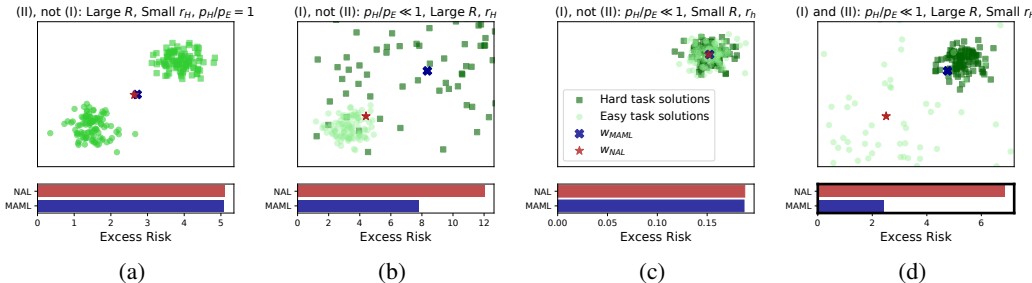

Figure 1: MAML, NAL excess risks and optimal solutions in various environments.

Learning (NAL) method. Notably, MAML should always perform at least as well as NAL because its meta-training procedure aligns with the meta-test time evaluation of performance after a few steps of task-specific SGD. Thus, our goals are to quantify the gain provided by MAML and determine the scenarios in which this gain is most significant. We start by studying the multi-task linear regression setting since it allows us to compare the exact solutions and excess risks for MAML and NAL. In doing so, we obtain novel insights on how MAML deals with hard tasks differently than NAL, and show that MAML achieves significant gain over NAL only if certain conditions are satisfied relating to the hardness and geography of the tasks. These observations provide a new understanding of MAML that is distinct from the representation learning perspective considered in recent works (Raghu et al., 2019; Du et al., 2020; Tripuraneni et al., 2020). We then give theoretical and empirical evidence generalizing these insights to neural networks.

In particular, our main observations are best captured in the setting in which tasks are either "hard" or "easy". We let $\rho_H$ be the hardness parameter for the hard tasks, and $\rho_E$ be the hardness parameter for the easy tasks, where $\rho_E > \rho_H$ (smaller hardness parameter means more hard). As we measure the hardness of a task by the rate at which gradient descent converges for the task, in this case, the hardness parameter is the task loss function's strong convexity parameter. For a particular task environment, we let $R$ be the dimension-normalized distance between the average of easy tasks' optimal solutions and the average of hard tasks' optimal solutions, and let $r_H$ quantify the variance, or dispersion, of the hard tasks' optimal solutions. Assuming that $\frac{\rho_H}{\rho_E}(1 - \frac{\rho_H}{\rho_E})^2 \gg \frac{d}{m}$, where $d$ is the problem dimension and $m$ is the number of samples used for adaptation, then the ratio of the expected excess risks after task-specific adaptation of the NAL and MAML solutions is approximately (Corollary 2): $\frac{\mathcal{E}_m(\mathbf{w}_{ERM})}{\mathcal{E}_m(\mathbf{w}_{MAML})} \approx 1 + \frac{R^2}{r_H}$. Thus, the largest gain for MAML over NAL occurs when the task environment satisfies (informal):

I. *Hardness discrepancy:* the hard tasks are significantly more difficult than the easy tasks $\frac{\rho_H}{\rho_E} \ll 1$, without being impossibly hard ($\rho_H \geq c > 0$), and

II. *Benign geography:* the optimal solutions of the hard tasks have small variance $r_H$, and the distance between the hard and easy task centers $R$ is large.

Figure 1 summarizes observations (I) and (II) by plotting locations of the easy and hard tasks' optimal solutions sampled from four distinct task environments and the corresponding solutions and excess risks for NAL and MAML. The environment in subfigure (a) violates (I) since $\frac{\rho_H}{\rho_E} = 1$. Subfigures (b)-(c) show environments with either small $R$ or large $r_h$, so (II) is not satisfied. In contrast, the environment in subfigure (d) has small $r_H$, large $R$, and $\frac{\rho_H}{\rho_E} = 0.2$, in which case as expected MAML achieves the largest gain over NAL, $\frac{\mathcal{E}_m(\mathbf{w}_{NAL})}{\mathcal{E}_m(\mathbf{w}_{MAML})} \approx 3$.

**Summary – Why MAML.** We show theoretically and empirically that MAML outperforms standard training (NAL) in linear and nonlinear settings by finding a solution that excels on the more-difficult-to-optimize (hard) tasks, as long as the hard tasks are sufficiently similar. Our work thus highlights the importance of task hardness and task geography to the success of MAML.

### 1.1 RELATED WORK

A few works have explored why MAML is effective. Raghu et al. (2019) posed two hypotheses for MAML's success in training neural networks: *rapid learning*, meaning that MAML finds a set of parameters advantageous for full adaptation on new tasks, and *feature reuse*, meaning that

MAML learns a set of reusable features, among the tasks. The authors' experiments showed that MAML learns a shared representation, supporting the feature reuse hypothesis. Motivated by this idea, Saunshi et al. (2020) proved that Reptile (Nichol et al., 2018), a similar algorithm to MAML, can learn a representation that reduces the new-task sample complexity in a two-task, linear setting. Conversely, Oh et al. (2020) gave empirical evidence that MAML performance does not degrade when forced to learn unique representations for each task, and Goldblum et al. (2020) showed that while some meta-learning algorithms learn more clustered features compared to standard training, the same cannot be said of MAML. Moreover, a series of works have shown that removing the inner loop and learning a distinct classifier for each class of images in the environment can yield representations as well-suited for few-shot image classification as MAML's (Chen et al., 2019). In this work, we take a more general, landscape-based perspective on the reasons for MAML's success, and show that MAML can still achieve meaningful gain without necessarily learning a representation.

Much of the theory surrounding MAML and meta-learning more broadly has focused on linear settings, specifically multi-task linear regression or the related linear centroid problem (Denevi et al., 2018). Some works have studied how to allocate data amongst and within tasks (Cioba et al., 2021; Bai et al., 2020; Saunshi et al., 2021). Denevi et al. (2018) considered meta-learning a common mean for ridge regression, and Kong et al. (2020) studied whether many small-data tasks and few large-data tasks is sufficient for meta-learning. Other works have examined the role of the inner loop SGD step size in meta-learning approaches (Bernacchia, 2021; Charles & Konečnỳ, 2020; Wang et al., 2020b), while Gao & Sener (2020) studied the trade-off between the accuracy and computational cost of the NAL and MAML training solutions. However, unlike our work, these works either did not consider or did not provide any interpretable characterization of the types of task environments in which MAML is effective. Other theoretical studies of MAML and related methods have focused on convergence rates in general settings (Fallah et al., 2020; Rajeswaran et al., 2019; Zhou et al., 2019; Ji et al., 2020a;b; Collins et al., 2020; Wang et al., 2020a) and excess risk bounds for online learning (Finn et al., 2019; Balcan et al., 2019; Khodak et al., 2019). Like the current work, Fallah et al. (2020) noticed that the MAML solution should be closer than the NAL solution to the hard tasks' global minima, but the authors neither quantified this observation nor further compared MAML and NAL.

## 2 PROBLEM SETUP: TRAINING TO ADAPT

We aim to determine when and why MAML yields models that are significantly more adaptable compared to models obtained by solving the traditional NAL problem in multi-task environments. To this end, we consider the gradient-based meta-learning setting in which a meta-learner tries to use samples from a set of tasks observed during meta-training to compute a model that performs well after one or a few steps of SGD on a new task drawn from the same environment at meta-test time.

Specifically, we follow Baxter (1998) by considering an environment $p$ which is a distribution over tasks. Each task $\mathcal{T}_i$ is composed of a data distribution $\mu_i$ over an input space $\mathcal{X}$ and a label space $\mathcal{Y}$. We take our model class to be the family of functions $\{h_{\mathbf{w}} : \mathcal{X} \to \mathcal{Y} : \mathbf{w} \in \mathbb{R}^D\}$ where $h_{\mathbf{w}}$ is a model parameterized by $\mathbf{w}$. The population loss $f_i(\mathbf{w})$ on task $i$ is the expected value of the loss of $h_{\mathbf{w}}$ on samples drawn from $\mu_i$, namely $f_i(\mathbf{w}) := \mathbb{E}_{(\mathbf{x},y) \sim \mu_i}[\ell(h_{\mathbf{w}}(\mathbf{x}), y)]$, where $\ell$ is some loss function, such as the squared or cross entropy loss.

During training, the meta-learner samples $T$ tasks from $p$ and $n$ points $\{(\mathbf{x}_i^j, y_i^j)\}_{j=1}^n \sim \mu_i^n$ for each sampled task $\mathcal{T}_i$. The meta-learner uses this data to compute an initial model $\mathbf{w}$, which is then evaluated as follows. First, the meta-learner samples a new task $\mathcal{T}_i \sim p$ and $m$ labeled points $\{(\hat{\mathbf{x}}_i^j, \hat{y}_i^j)\}_{j=1}^m \sim \mu_i^m$. Next, it updates $\mathbf{w}$ with one step of stochastic gradient descent (SGD) on the loss function $f_i$ using those $m$ samples and step size $\alpha$. Namely, letting $\hat{\mathbf{X}}_i \in \mathbb{R}^{m \times d}$ and $\hat{\mathbf{y}}_i \in \mathbb{R}^m$ denote the matrix and vector containing the $m$ feature vectors and their labels, respectively, the update of $\mathbf{w}$ is given by $\mathbf{w}_i := \mathbf{w} - \alpha \nabla_{\mathbf{w}} \hat{f}_i(\mathbf{w}; \hat{\mathbf{X}}_i, \hat{\mathbf{y}}_i)$, where $\hat{f}_i(\mathbf{w}; \hat{\mathbf{X}}_i, \hat{\mathbf{y}}_i) := \frac{1}{m} \sum_{j=1}^m \ell(h_{\mathbf{w}}(\hat{\mathbf{x}}_i^j), \hat{y}_i^j)$ is the empirical average of the loss on the $m$ samples in $(\hat{\mathbf{X}}_i, \hat{\mathbf{y}}_i)$. The test loss of $\mathbf{w}$ is the expected population loss of $\mathbf{w}_i$, where the expectation is taken over tasks and the $m$ samples, specifically

$$F_m(\mathbf{w}) := \mathbb{E}_i \mathbb{E}_{(\hat{\mathbf{X}}_i, \hat{\mathbf{y}}_i)}\big[f_i(\mathbf{w} - \alpha \nabla \hat{f}_i(\mathbf{w}; \hat{\mathbf{X}}_i, \hat{\mathbf{y}}_i))\big] \tag{1}$$

where we have used the shorthand $\mathbb{E}_i := \mathbb{E}_{\mathcal{T}_i \sim p}$ and $\mathbb{E}_{(\hat{\mathbf{X}}_i, \hat{\mathbf{y}}_i)} := \mathbb{E}_{(\hat{\mathbf{X}}_i, \hat{\mathbf{y}}_i) \sim \mu_i^m}$. For fair evaluation, we measure solution quality by the excess risk

$$\mathcal{E}_m(\mathbf{w}) := F_m(\mathbf{w}) - \inf_{\mathbf{w}' \in \mathbb{R}^D} F_m(\mathbf{w}') \tag{2}$$

The excess risk is the difference between the average performance of $\mathbf{w}$ after one step of task-specific adaptation from the performance of the best possible initialization for fast adaptation. To find a model with small excess risk, one can solve one of two problems during training, NAL or MAML.

**NAL.** NAL minimizes the loss $f_i(\mathbf{w})$ on average across the training tasks, which may yield small excess risk $\mathcal{E}_m(\mathbf{w})$ with less computational cost than MAML (Gao & Sener, 2020). Denoting the $n$ training examples for the $i$-th task as $\mathbf{X}_i \in \mathbb{R}^{n \times d}$ and their labels as $\mathbf{y}_i \in \mathbb{R}^n$, NAL solves

$$\min_{\mathbf{w} \in \mathbb{R}^D} F_{NAL}^{tr}(\mathbf{w}) := \tfrac{1}{T} \sum_{i=1}^T \hat{f}_i(\mathbf{w}; \mathbf{X}_i, \mathbf{y}_i), \tag{3}$$

which is a surrogate for the expected risk minimization problem defined as $\min_{\mathbf{w} \in \mathbb{R}^D} \mathbb{E}_i[f_i(\mathbf{w})]$. We let $\mathbf{w}_{NAL}^*$ denote the unique solution of the expected risk minimization problem, and let $\mathbf{w}_{NAL}$ denote the unique solution to (3). We emphasize that in our study we evaluate the solution of NAL by its expected error after running one step of SGD using the $m$ samples from a new task that are released at test time, and this error is captured by the excess risk $\mathcal{E}_m(\mathbf{w}_{NAL})$, defined in (2).

**MAML.** In contrast to NAL, MAML minimizes a surrogate loss of (1) during training. According to the MAML framework, the $n$ samples per task are divided into $\tau$ training "episodes", each with $n_2$ "inner" samples for the inner SGD step and $n_1$ "outer" samples for evaluating the loss of the fine-tuned model. Thus, we have that $\tau(n_1 + n_2) = n$. We denote the matrices that contain the outer samples for the $j$-th episode of the $i$-th task as $\mathbf{X}_{i,j}^{out} \in \mathbb{R}^{n_1 \times d}$ and $\mathbf{y}_{i,j}^{out} \in \mathbb{R}^{n_1}$, and the matrices that contain the inner samples as $\mathbf{X}_{i,j}^{in} \in \mathbb{R}^{n_2 \times d}$ and $\mathbf{y}_{i,j}^{in} \in \mathbb{R}^{n_2}$. The MAML objective is then

$$\min_{\mathbf{w} \in \mathbb{R}^D} F_{MAML}^{tr}(\mathbf{w}) := \tfrac{1}{T\tau} \sum_{i=1}^T \sum_{j=1}^\tau \hat{f}_i(\mathbf{w} - \alpha \nabla \hat{f}_i(\mathbf{w}; \mathbf{X}_{i,j}^{in}, \mathbf{y}_{i,j}^{in}); \mathbf{X}_{i,j}^{out}, \mathbf{y}_{i,j}^{out}). \tag{4}$$

We denote the unique solution to (4) by $\mathbf{w}_{MAML}$ and the unique solution to the population version (1) by $\mathbf{w}_{MAML}^*$. We expect $\mathcal{E}_m(\mathbf{w}_{MAML}) \leq \mathcal{E}_m(\mathbf{w}_{NAL})$ since (4) is a surrogate for the true objective of minimizing (1). However, the gain of MAML over NAL may not be significant enough to justify its added computational cost (Gao & Sener, 2020), necessitating a thorough understanding of the relative behavior of MAML and NAL.

## 3 MULTI-TASK LINEAR REGRESSION

We first explore the relative behaviors of MAML and NAL in a setting in which we can obtain closed-form solutions: multi-task linear regression. Here, the model $h_\mathbf{w}$ maps inputs $\mathbf{x}$ to predicted labels by taking the inner product with $\mathbf{w}$, i.e. $h_\mathbf{w}(\mathbf{x}) = \langle \mathbf{w}, \mathbf{x} \rangle$. The loss function $\ell$ is the squared loss, therefore $f_i(\mathbf{w}) = \tfrac{1}{2} \mathbb{E}_{(\mathbf{x}_i, y_i) \sim \mu_i}[(\langle \mathbf{w}, \mathbf{x} \rangle - y_i)^2]$ and $\hat{f}_i(\mathbf{w}; \mathbf{X}_i, \mathbf{y}_i) = \tfrac{1}{2} \|\mathbf{X}_i \mathbf{w} - \mathbf{y}_i\|_2^2$ for all $i$. We consider a realizable setting in which the data for the $i$-th task is Gaussian and generated by a ground truth model $\mathbf{w}_{i,*} \in \mathbb{R}^d$. That is, points $(\mathbf{x}_i, y_i)$ are sampled from $\mu_i$ by first sampling $\mathbf{x}_i \sim \mathcal{N}(\mathbf{0}, \boldsymbol{\Sigma}_i)$, then sampling $y_i \sim \mathcal{N}(\langle \mathbf{w}_{i,*}, \mathbf{x}_i \rangle, \nu^2)$, i.e. $y_i = \langle \mathbf{w}_{i,*}, \mathbf{x}_i \rangle + z_i$ where $z_i \sim \mathcal{N}(0, \nu^2)$. In this setting the population-optimal solutions for NAL and MAML are given by:

$$\mathbf{w}_{NAL}^* = \mathbb{E}_i[\boldsymbol{\Sigma}_i]^{-1} \mathbb{E}_i[\boldsymbol{\Sigma}_i \mathbf{w}_i^*], \quad \text{and} \quad \mathbf{w}_{MAML}^* = \mathbb{E}_i[\mathbf{Q}_i^{(n_2)}]^{-1} \mathbb{E}_i[\mathbf{Q}_i^{(n_2)} \mathbf{w}_i^*], \tag{5}$$

where for any $s \in \mathbb{N}$, we define $\mathbf{Q}_i^{(s)} := (\mathbf{I}_d - \alpha \boldsymbol{\Sigma}_i) \boldsymbol{\Sigma}_i (\mathbf{I}_d - \alpha \boldsymbol{\Sigma}_i) + \tfrac{\alpha^2}{s}(\text{tr}(\boldsymbol{\Sigma}_i^2) \boldsymbol{\Sigma}_i + \boldsymbol{\Sigma}_i^3)$. Note that these $\mathbf{Q}_i^{(s)}$ matrices are composed of two terms: a preconditioned covariance matrix $\boldsymbol{\Sigma}_i (\mathbf{I}_d - \alpha \boldsymbol{\Sigma}_i)^2$, and a perturbation matrix due to the stochastic gradient variance. We provide expressions for the empirical solutions $\mathbf{w}_{NAL}$ and $\mathbf{w}_{MAML}$ for this setting and show that they converge to $\mathbf{w}_{NAL}^*$ and $\mathbf{w}_{MAML}^*$ as $n, T, \tau \to \infty$ in Appendix D. Since our focus is ultimately on the nature of the solutions sought by MAML and NAL, not on their non-asymptotic behavior, we analyze $\mathbf{w}_{NAL}^*$ and $\mathbf{w}_{MAML}^*$ in the remainder of this section, starting with the following result.

It is most helpful to interpret the solutions $\mathbf{w}_{MAML}^*$ and $\mathbf{w}_{NAL}^*$ and their corresponding excess risks through the lens of task hardness. In this strongly convex setting, we naturally define **task hardness**

**as the rate at which gradient descent converges to the optimal solution for the task**, with harder tasks requiring more steps of gradient descent to reach an optimal solution. For step size $\alpha$ fixed across all tasks, the rate with which gradient descent traverses each $f_i$ is determined by the minimum eigenvalue of the Hessian of $f_i$, namely $\lambda_{\min}(\boldsymbol{\Sigma}_i)$. So, for ease of interpretation, we can think of the easy tasks as having data with large variance in all directions (all the eigenvalues of their $\boldsymbol{\Sigma}_i$ are large), while the hard tasks have data with small variance in all directions (all $\lambda(\boldsymbol{\Sigma}_i)$ are small).

Note that both $\mathbf{w}^*_{MAML}$ and $\mathbf{w}^*_{NAL}$ are normalized weighted sums of the task optimal solutions, with weights being functions of the $\boldsymbol{\Sigma}_i$'s. For simplicity, consider the case in which $m$ and $n_2$ are large, thus the $\mathbf{Q}_i$ matrices are dominated by $\boldsymbol{\Sigma}_i(\mathbf{I}_d - \alpha\boldsymbol{\Sigma}_i)^2$. Since the weights for $\mathbf{w}^*_{NAL}$ are proportional to the $\boldsymbol{\Sigma}_i$ matrices, $\mathbf{w}^*_{NAL}$ **is closer to the easy task optimal solutions**, as $\boldsymbol{\Sigma}_i$ has large energy for easy tasks and small energy for hard tasks. Conversely, the MAML weights are determined by $\boldsymbol{\Sigma}_i(\mathbf{I}_d - \alpha\boldsymbol{\Sigma}_i)^2$, which induces a relatively larger weight on the hard tasks, so $\mathbf{w}^*_{MAML}$ **is closer to the hard task optimal solutions**.

Note that easy tasks can be approximately solved after one step of gradient descent from far away, which is not true for hard tasks. We therefore expect (I) MAML to perform well on both hard and easy tasks since $\mathbf{w}^*_{MAML}$ is closer to the optimal solutions of hard tasks, and (II) NAL to perform well on easy tasks but struggle on hard tasks since $\mathbf{w}^*_{NAL}$ is closer to the optimal solutions of easy tasks. We explicitly compute the excess risks of $\mathbf{w}^*_{NAL}$ and $\mathbf{w}^*_{MAML}$ as follows.

**Proposition 1.** *The excess risks for* $\mathbf{w}^*_{NAL}$ *and* $\mathbf{w}^*_{MAML}$ *are:*

$$\mathcal{E}_m(\mathbf{w}^*_{NAL}) = \tfrac{1}{2}\mathbb{E}_i \left\| \mathbb{E}_{i'}\left[\boldsymbol{\Sigma}_{i'}\right]^{-1} \mathbb{E}_{i'}\left[\boldsymbol{\Sigma}_{i'}(\mathbf{w}^*_{i'} - \mathbf{w}^*_i)\right] \right\|^2_{\mathbf{Q}_i^{(m)}} \tag{6}$$

$$\mathcal{E}_m(\mathbf{w}^*_{MAML}) = \tfrac{1}{2}\mathbb{E}_i \left\| \mathbb{E}_{i'}\left[\mathbf{Q}_{i'}^{(n_2)}\right]^{-1} \mathbb{E}_{i'}\left[\mathbf{Q}_{i'}^{(n_2)}(\mathbf{w}^*_{i'} - \mathbf{w}^*_i)\right] \right\|^2_{\mathbf{Q}_i^{(m)}} \tag{7}$$

We next formally interpret these excess risks, and develop intuitions (I) and (II), by focusing on two key properties of the task environment: task hardness discrepancy and task geography.

## 3.1 Hardness discrepancy

We analyze the levels of task hardness that confer a significant advantage for MAML over NAL in this section. To do so, we compare $\mathbf{w}^*_{MAML}$ and $\mathbf{w}^*_{NAL}$ in an environment with two tasks of varying hardness. We let $n_2 = m$, $\boldsymbol{\Sigma}_1 = \rho_H\mathbf{I}_d$, and $\boldsymbol{\Sigma}_2 = \rho_E\mathbf{I}_d$, where $\rho_H < \rho_E$, thus task 1 is the hard task.[1]

In this setting, the NAL and MAML solutions defined in (5) can be simplified as

$$\mathbf{w}^*_{NAL} = \frac{\rho_H}{\rho_H + \rho_E}\mathbf{w}^*_1 + \frac{\rho_E}{\rho_H + \rho_E}\mathbf{w}^*_2, \qquad \mathbf{w}^*_{MAML} = \frac{a_H}{a_H + a_E}\mathbf{w}^*_1 + \frac{a_E}{a_H + a_E}\mathbf{w}^*_2, \tag{8}$$

where $a_E := (\rho_E(1 - \alpha\rho_E)^2 + \frac{d+1}{m}\alpha^2\rho_E^3)$ and $a_H := (\rho_H(1 - \alpha\rho_H)^2 + \frac{d+1}{m}\alpha^2\rho_H^3)$. Note that the natural choice of $\alpha$ is the inverse of the largest task smoothness parameter ($\frac{1}{\rho_E}$ in this case). Setting $\alpha = \frac{1}{\rho_E}$ yields $a_E = \frac{d+1}{m}\rho_E$ and $a_H = \rho_H(1 - \frac{\rho_H}{\rho_E})^2 + \frac{(d+1)\rho_H^3}{m\rho_E^2}$. As a result, we can easily see that for sufficiently large values of $m$, we have $a_H > a_E$. This observation shows that the solution of MAML is closer to the solution of the harder task, i.e., $\mathbf{w}^*_1$. On the other hand, $\mathbf{w}_{NAL}$ is closer to $\mathbf{w}^*_2$, the solution to the easy task, since $\rho_H < \rho_E$.

Considering these facts, we expect the performance of MAML solution after adaptation to exceed that of NAL. Using Proposition 1, the excess risks for NAL and MAML in this setting are

$$\mathcal{E}_m(\mathbf{w}_{NAL}) = \frac{a_E\rho_H^2 + a_H\rho_E^2}{(\rho_E + \rho_H)^2}, \quad \mathcal{E}_m(\mathbf{w}_{MAML}) = \frac{a_E a_H}{a_E + a_H}. \tag{9}$$

Recalling that $a_E \approx 0$ for $m \gg d$, we conclude that MAML achieves near-zero excess risk in the case of large $m$. In particular, we require that $\rho_H(1 - \frac{\rho_H}{\rho_E})^2 \gg \frac{d\rho_E}{m}$, otherwise the $O(d/m)$ terms

---

[1]The effects of task hardness could be removed by scaling the data so that it would have covariance $\alpha^{-1}\mathbf{I}_d$. However, the current setting is useful to build intuition. Further, one can imagine a similar setting in which the first dimension has variance $\alpha^{-1}$ and the rest have variance $\rho_H$, in which scaling would not be possible (as it would result in gradient descent not converging in the first coordinate).

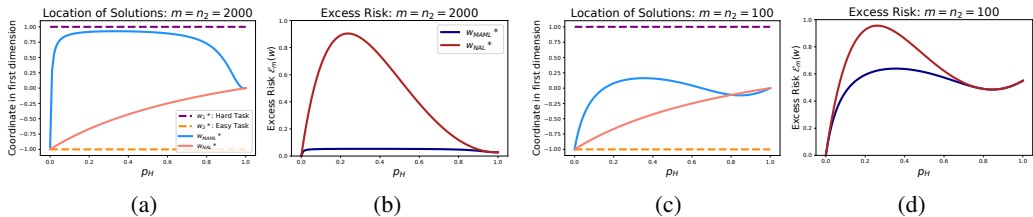

Figure 2: First coordinates of $\mathbf{w}^*_{MAML}$, $\mathbf{w}^*_{NAL}$ and their excess risks for various $\rho_H$ for large $m$ (in subplots (a)-(b) for $m=2000$) and small $m$ (in subplots (c)-(d) for $m=100$).

in $a_E$ and $a_H$ are non-negligible. Meanwhile, for NAL we have $\mathcal{E}_m(\mathbf{w}_{NAL}) = \frac{a_H \rho_E^2}{(\rho_E + \rho_H)^2}$, which may be significantly larger than zero if $\frac{\rho_H}{\rho_E} \ll 1$, i.e. the harder task is significantly harder than the easy task. Importantly, the error for NAL is dominated by poor performance on the hard task, which MAML avoids by initializing close to the hard task solution. Thus, these expressions pinpoint the level of hardness discrepancy needed for superior MAML performance: $\frac{\rho_H}{\rho_E}$ *must be much smaller than 1, but also larger than 0, such that* $\frac{\rho_H}{\rho_E}(1 - \frac{\rho_H}{\rho_E})^2 \gg \frac{d}{m}$.

Figure 2 visualizes these intuitions in the case that $\mathbf{w}_1^* = \mathbf{1}_d$, $\mathbf{w}_2^* = -\mathbf{1}_d$, $d = 10$ and $\sigma^2 = 0.01$. Subfigures (a) and (c) show the locations of the first coordinates of $\mathbf{w}^*_{NAL}$ and $\mathbf{w}^*_{MAML}$ for varying $\rho_H$, and $m = 2000$ and $m = 500$, respectively. Subfigures (b) and (d) show the corresponding excess risks. We observe that unlike NAL, MAML initializes closer to the optimal solution of the harder task as long as $\frac{\rho_H}{\rho_E}$ is not close to zero or one, which results in significantly smaller excess risk for MAML compared to NAL in such cases, especially for large $m$.

Figure 2 further shows that the MAML and NAL excess risks go to zero with $\rho_H$. This is also shown in (9) and the definition of $a_H$, and points to the fact that *too* much hardness discrepancy causes no gain for MAML. The reason for this is that $\rho_H \to 0$ corresponds to the hard task data going to zero (its mean), in which case any linear predictor has negligible excess risk. Consequently, both NAL and MAML ignore the hard task and initialize at the easy task to achieve near-zero excess risk here.

**Remark 1.** *The condition* $\frac{\rho_H}{\rho_E}(1 - \frac{\rho_H}{\rho_E})^2 \gg \frac{d}{m}$ *requires that* $m \gg d$. *However, this condition arises due to the simplification* $tr(\mathbf{\Sigma}_i) = O(d)$, *where* $tr(\mathbf{\Sigma}_i)$ *can be thought of as the effective problem dimension (Kalan et al., 2020). In realistic settings, the effective dimension may be* $o(d)$, *which would reduce the complexity of* $m$ *accordingly.*

### 3.2 TASK GEOGRAPHY

The second important property of the task environment according to Proposition 1 is the location, i.e. geography, of the task optimal solutions. In this section, we study how task geography affects the MAML and NAL excess risks by considering a task environment with many tasks. In particular, the task environment $\mu$ is a mixture over distributions of hard and easy tasks, with mixture weights $0.5$. The optimal solutions $\mathbf{w}_i^* \in \mathbb{R}^d$ for hard tasks are sampled according to $\mathbf{w}_i^* \sim \mathcal{N}(R\mathbf{1}_d, r_H\mathbf{I}_d)$ and for easy tasks are sampled as $\mathbf{w}_i^* \sim \mathcal{N}(\mathbf{0}_d, r_E\mathbf{I}_d)$. Therefore $R$ is the dimension-normalized distance between the centers of the hard and easy tasks' optimal solutions, and $r_H$ and $r_E$ capture the spread of the hard and easy tasks' optimal solutions, respectively. The data covariance is $\mathbf{\Sigma}_i = \rho_H\mathbf{I}_d$ for the hard tasks and $\mathbf{\Sigma}_i = \rho_E\mathbf{I}_d$ for the easy tasks, recalling that $\rho_H$ and $\rho_E$ parameterize hardness, with smaller $\rho_H$ meaning harder task. In this setting the following corollary follows from Proposition 1.

**Corollary 1.** *In the setting described above, the excess risks of* $\mathbf{w}^*_{NAL}$ *and* $\mathbf{w}^*_{MAML}$ *are:*

$$\mathcal{E}_m(\mathbf{w}^*_{NAL}) = \frac{d}{4(\rho_E + \rho_H)^2}\Big[(a_E\rho_E^2 + 2a_E\rho_E\rho_H)r_E + a_E\rho_H^2(r_E + R^2)$$
$$+ (a_H\rho_H^2 + 2a_H\rho_E\rho_H)r_H + a_H\rho_E^2(r_H + R^2)\Big] \quad (10)$$

$$\mathcal{E}_m(\mathbf{w}^*_{MAML}) = \frac{d}{4(a_E + a_H)^2}\Big[(a_E^3 + 2a_E^2 a_H)r_E + (a_H^3 + 2a_E a_H^2)r_H$$
$$+ a_E a_H^2(r_E + R^2) + a_E^2 a_H(r_H + R^2)\Big] \quad (11)$$

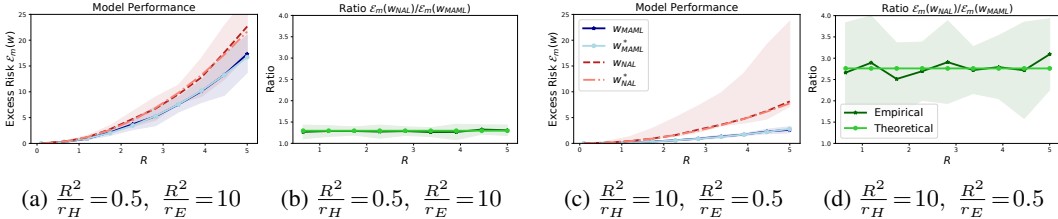

(a) $\frac{R^2}{r_H} = 0.5$, $\frac{R^2}{r_E} = 10$    (b) $\frac{R^2}{r_H} = 0.5$, $\frac{R^2}{r_E} = 10$    (c) $\frac{R^2}{r_H} = 10$, $\frac{R^2}{r_E} = 0.5$    (d) $\frac{R^2}{r_H} = 10$, $\frac{R^2}{r_E} = 0.5$

Figure 3: Theoretical and empirical excess risks for NAL and MAML and ratios of the NAL to MAML excess risk in the setting described in Section 3.2.

where $a_E := \rho_E(1 - \alpha\rho_E)^2 + \frac{d+1}{m}\alpha^2\rho_E^3$ and $a_H := \rho_H(1 - \alpha\rho_H)^2 + \frac{d+1}{m}\alpha^2\rho_H^3$.

Each excess risk is a normalized weighted sum of the quantities $r_E, r_H$ and $R^2$. So, the comparison between MAML and NAL depends on the relative weights each algorithm induces on these task environment properties. If $\frac{\rho_H}{\rho_E}(1 - \frac{\rho_H}{\rho_E})^2 \gg \frac{d}{m}$, then $a_H \gg a_E$ and the dominant weight in $\mathcal{E}_m(\mathbf{w}^*_{MAML})$ is on the $r_H$ term, while the dominant weights in $\mathcal{E}_m(\mathbf{w}^*_{NAL})$ are on the $r_H + R^2$ and $r_H$ terms. This observation leads us to obtain the following corollary of Proposition 1.

**Corollary 2.** *In the above setting, with $\alpha = 1/\rho_E$ and $\frac{\rho_H}{\rho_E}(1 - \frac{\rho_H}{\rho_E})^2 \gg \frac{d}{m}$, the relative excess risk for NAL compared to MAML satisfies*

$$\frac{\mathcal{E}_m(\mathbf{w}^*_{NAL})}{\mathcal{E}_m(\mathbf{w}^*_{MAML})} \approx 1 + \frac{R^2}{r_H}. \tag{12}$$

Corollary 2 shows that MAML achieves large gain over NAL when: *(i) the hard tasks' solutions are closely packed ($r_H$ is small) and (ii) the hard tasks' solutions are far from the center of the easy tasks' solutions ($R$ is large)*. Condition (i) allows MAML to achieve a small excess risk by initializing in the center of the hard task optimal solutions, while condition (ii) causes NAL to struggle on the hard tasks since it initializes close to the easy tasks' solutions. These conditions are reflected by the fact that the MAML excess risk weighs $r_H$ (the spread of the hard tasks) most heavily, whereas the NAL excess risk puts the most weight on $R^2$ (distance between hard and easy task solutions) as well as $r_H$.

Note that the above discussion holds under the condition that $a_H \gg a_E$. In order for $a_H \gg a_E$, we must have $\frac{\rho_H}{\rho_E}(1 - \frac{\rho_H}{\rho_E})^2 \gg \frac{d}{m}$, i.e. $m \gg d$ and $0 < \frac{\rho_H}{\rho_E} \ll 1$, as we observed in our discussion on hardness discrepancy. Now, we see that even with appropriate hardness discrepancy, the hard tasks must be both closely packed and far from the center of the easy tasks in order for MAML to achieve significant gain over NAL. This conclusion adds greater nuance to prior results (Balcan et al., 2019; Jose & Simeone, 2021), in which the authors argued that gradient-based meta-learning (e.g. MAML) is only effective when all of the tasks' optimal solutions are close to each other (in our case, all of $r_E$, $r_H$ and $R$ are small). Crucially, our analysis shows that NAL would also perform well in this scenario, and neither $r_E$ nor $R$ need to be small for MAML to achieve small excess risk.

To further explain these insights, we return to Figure 1 in which easy and hard task optimal solutions are each sampled from 10-dimensional Gaussian distributions of the form described above, and 100 random task optimal solutions are shown, along with the population-optimal MAML and NAL solutions. In subfigures (b), (c) and (d), we set $\rho_E = 0.9$, $\rho_H = 0.1$, and $m = 500$. Among these plots, the largest gain for MAML is in the case that the hard tasks' optimal solutions are closely packed (small $r_H$), and their centers are far from each other (large $R$), *demonstrating the the primary dependence of relative performance on $R^2/r_H$.*

We plot more thorough results for this setting in Figure 3. Here, we vary $R$ and compare the performance of $\mathbf{w}^*_{NAL}$ and $\mathbf{w}^*_{MAML}$ in settings with relatively large $r_H$ and small $r_E$, specifically choosing $r_H$ and $r_E$ such that $\frac{R^2}{r_H} = 0.5$ and $\frac{R^2}{r_E} = 10$ in (a)-(b), and $\frac{R^2}{r_H} = 10$ and $\frac{R^2}{r_E} = 0.5$ in (c)-(d). Here we also plot the empirical solutions $\mathbf{w}_{NAL}$ and $\mathbf{w}_{MAML}$. *Again, MAML significantly outperforms NAL when $R^2/r_H$ is large (subfigures (c)-(d)), but not otherwise.*

## 4 TWO-LAYER NEURAL NETWORK

In this section, we consider a non-linear setting in which each task is a regression problem with a two-layer neural network with a fixed second layer. The $k$-th neuron in the network maps $\mathbb{R}^d \to \mathbb{R}$

via $\sigma(\langle \mathbf{w}^k, \mathbf{x} \rangle)$, where $\sigma : \mathbb{R} \to \mathbb{R}$ is some activation function and $\mathbf{w}^k \in \mathbb{R}^d$ is the parameter vector. The network contains $M$ neurons for a total of $D = Md$ parameters, which are contained in the matrix $\mathbf{W} := [\mathbf{w}^1, \ldots, \mathbf{w}^M] \in \mathbb{R}^{d \times M}$. The predicted label for the data point $\mathbf{x}$ is the sum of the neuron outputs, namely $h_{\mathbf{W}}(\mathbf{x}) := \sum_{k=1}^{M} \sigma(\langle \mathbf{w}^k, \mathbf{x} \rangle)$. The loss function is again the squared loss, i.e. $f_i(\mathbf{W}) = \frac{1}{2} \mathbb{E}_{(\mathbf{x}_i, y_i) \sim \mu_i}[(\sigma(\sum_{k=1}^{M} \langle \mathbf{w}^k, \mathbf{x}_i \rangle) - y_i)^2]$. Ground-truth models $\mathbf{W}_{i,*}$ generate the data for task $i$. We sample $(\mathbf{x}_i, y_i) \sim \mu_i$ by first sampling $\mathbf{x}_i \sim \mathcal{N}(\mathbf{0}_d, \mathbf{\Sigma}_i)$ then computing $y_i = \sum_{k=1}^{M} \sigma(\langle \mathbf{w}_{i,*}^k, \mathbf{x}_i \rangle)$. The following result demonstrates an important property of the MAML objective function in the two-task version of this setting.

**Theorem 1.** *Suppose that in the setting described above, the task environment is the uniform distribution over two tasks, and $\sigma$ is the ReLU, Softplus, Sigmoid, or tanh function. Let $\mathbf{\Sigma}_i = \mathbf{I}_d$ and define $s_{i,k}$ as the $k$-th singular value of $\mathbf{W}_{i,*}$, $\kappa_i := \|\mathbf{W}_{i,*}\|_2 / s_{i,M}$, and $\lambda_i := \frac{\Pi_{k=1}^{M} s_{i,k}}{s_{i,M}^M}$ for $i \in \{1, 2\}$. Further, define the regions $\mathcal{S}_i := \{\mathbf{W} : \|\mathbf{W} - \mathbf{W}_{i,*}\|_2 \leq c_1/(s_{i,1}^c \lambda_i \kappa_i^2 M^2)\}$ and the parameters $\beta_i := \frac{c_2}{\lambda_i \kappa_i^2}$ and $L_i := c_3 M s_{i,1}^{2c}$ for $i \in \{1, 2\}$ and absolute constants $c, c_1, c_2,$ and $c_3$. Let $\beta_1 < \beta_2$ and $L_1 < L_2$. For any stationary point $\mathbf{W}_{MAML}^*$ of the population MAML objective (1) with full inner gradient step ($m = \infty$), such that $\mathbf{W}_{MAML}^* \in \mathcal{S}_1 \cap \mathcal{S}_2$, must satisfy for all $\alpha \leq 1/L_2$:*

$$\|\nabla f_1(\mathbf{W}_{MAML})\|_2 \leq \frac{1 - 2\alpha\beta_2}{(1 - \alpha L_1)^2} \|\nabla f_2(\mathbf{W}_{MAML})\|_2 + O(\alpha^2). \tag{13}$$

**Interpretation: MAML prioritizes hard tasks.** In the setting above, $\beta_i$ and $L_i$ are strong convexity and smoothness parameters, respectively, of the function $f_i$ on the region $\mathcal{S}_i$. Here, task 1 is the harder task since $\beta_1$ and $\beta_2$ control the rate with which gradient descent converges to the ground-truth solutions (Zhong et al., 2017), with smaller $\beta_i$ implying slower convergence, and $\beta_1 < \beta_2$. Thus, Theorem 1 shows that, any stationary point of the MAML objective in $\mathcal{S}_1 \cap \mathcal{S}_2$ has smaller gradient norm on the hard task than on the easy task as long as there is sufficient hardness discrepancy, specifically $\beta_2 > L_1$. Physically, this condition means that the curvature of the loss function for the easy task is more steep than the curvature of the hard task in all directions around their ground-truth solutions. The smaller gradient norm of MAML stationary points on the harder task suggests that *MAML solutions prioritize hard-task performance.* In contrast, we can easily see that any stationary point $\mathbf{w}_{NAL}$ of the NAL population loss must satisfy the condition $\|\nabla f_1(\mathbf{w}_{NAL})\|_2 = \|\nabla f_2(\mathbf{w}_{NAL})\|_2$, meaning that *NAL has no such preference for the hard task.*

Figure 4 demonstrates the importance of task hardness in comparing NAL and MAML in this setting. Here, for ease of visualization we consider the case in which $M = 1$ and $d = 2$, i.e. the learning model consists of one neuron (with Softplus activation) mapping from $\mathbb{R}^2 \to \mathbb{R}$. Each subfigure plots the NAL or MAML population loss landscape for different task environments with $m = 250$, as well as the ground-truth neuron for each task. In light of prior work showing that the number of gradient steps required to learn a single neuron diminishes with the variance of the data distribution (Theorem 3.2 in Yehudai & Ohad (2020)), we again control task hardness via the data variance. In all plots, $\mathbf{\Sigma}_i = 0.5\mathbf{I}_2$ for hard tasks and $\mathbf{\Sigma}_i = \mathbf{I}_2$ for easy tasks. Note that the MAML loss (evaluated after one step of adaptation) is the evaluation metric we ultimately care about.

We observe that when all tasks are equally hard, the MAML and NAL solutions are identical (subfigures (a)-(b)), whereas when one task is hard, MAML initializes closer to the hard task and achieves significantly better post-adaptation performance than the NAL solution, which is closer to the centroid of the easy tasks (c)-(d). This supports our intuition that MAML achieves significant gain over NAL in task environments in which it can leverage improved performance on hard tasks.

## 5 EXPERIMENTS

We next experimentally study whether our observations generalize to problems closer to those seen in practice. We consider image classification on the Omniglot (Lake et al., 2019) and FS-CIFAR100 (Oreshkin et al., 2018) datasets. Following convention, tasks are $N$-way, $K$-shot classification problems, i.e., classification problems among $N$ classes where $K$ labeled images from each class are available for adapting the model. For both the Omniglot and FS-CIFAR100 experiments, we use the five-layer CNN used in Finn et al. (2017); Vinyals et al. (2016). NAL is trained equivalently to MAML with no inner loop and $n = n_1 + n_2$. Further details and error bounds are in Appendix C.

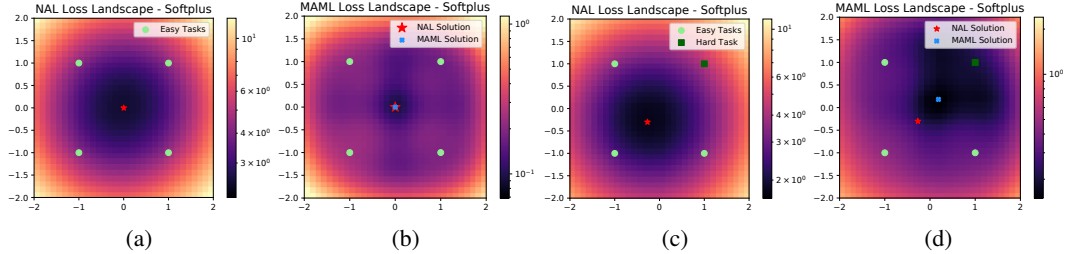

Figure 4: Loss landscapes and ground-truth neurons for NAL (a,c) and MAML (b,d) for two distinct, four-task environments and Softplus activation.

Table 1: Omniglot accuracies.

| Setting | | Train Tasks | | Test Tasks | |
|---|---|---|---|---|---|
| $r_H$ | Alg. | Easy | Hard | Easy | Hard |
| Large | MAML | 99.2 | 96.0 | 98.0 | 81.2 |
| Large | NAL-1 | 69.4 | 41.5 | 57.8 | 45.2 |
| Large | NAL-10 | 70.0 | 45.3 | 67.2 | 47.9 |
| Small | MAML | 99.2 | 99.1 | 98.1 | 95.4 |
| Small | NAL-1 | 69.2 | 46.0 | 55.8 | 45.8 |
| Small | NAL-10 | 70.2 | 44.0 | 67.8 | 48.9 |

Table 2: FS-CIFAR100 accuracies.

| Setting | | Train Tasks | | Test Tasks | |
|---|---|---|---|---|---|
| $p$ | Alg. | Easy | Hard | Easy | Hard |
| 0.99 | NAL | 78.2 | 9.9 | 74.8 | 9.5 |
| 0.99 | MAML | 92.9 | 50.1 | 84.6 | 21.7 |
| 0.5 | NAL | 81.9 | 9.5 | 76.5 | 8.6 |
| 0.5 | MAML | 93.6 | 41.1 | 84.3 | 17.9 |
| 0.01 | NAL | 82.4 | 7.6 | 76.9 | 8.2 |
| 0.01 | MAML | 94.0 | 15.7 | 85.0 | 11.8 |

**Omniglot.** Omniglot contains images of 1623 handwritten characters from 50 different alphabets. Characters from the same alphabet typically share similar features, so are harder to distinguish compared to characters from differing alphabets. We thus define easy tasks as classification problems among characters from *distinct* alphabets, and hard tasks as classification problems among characters from the *same* alphabet, consistent with prior definitions of *semantic hardness* (Zhou et al., 2020). Here we use $N = 5$ and $K = 1$. For NAL, we include results for $K = 10$ (NAL-10) in addition to $K = 1$ (NAL-1). We split the 50 alphabets into four disjoint sets: easy train (25 alphabets), easy test (15), hard train (5), and hard test (5). During training, tasks are drawn by first choosing 'easy' or 'hard' with equal probability. If 'easy' is chosen, 5 characters from 5 distinct alphabets among the easy train alphabets are selected. If 'hard' is chosen, a hard alphabet is selected from the hard train alphabets, then $N = 5$ characters are drawn from that alphabet. After training, we evaluate the models on new tasks drawn analogously from the test (unseen) alphabets as well as the train alphabets.

Table 1 gives the average errors after completing training for two experiments: the first (Large) when the algorithms use all 5 hard train and test alphabets, and the second (Small) when the algorithms use only one hard train alphabet (Sanskrit) and one hard test (Bengali). The terms 'Large' and 'Small' describe the hypothesized size of $r_H$ in each experiment: the optimal solutions of hard tasks drawn from ten (train and test) different alphabets are presumably more dispersed than hard tasks drawn from only two (train and test) similar alphabets. By our previous analysis, we expect MAML to achieve more gain over NAL in the Small $r_H$ setting. Table 1 supports this intuition, as MAML's performance improves significantly with more closely-packed hard tasks (Small), while NAL's does not change substantially. Hence, *MAML's relative gain over NAL increases with smaller $r_H$*.

**FS-CIFAR100.** FS-CIFAR100 has 100 total classes that are split into 80 training classes and 20 testing classes. We further split the 600 images in each of the training classes into 450 training images and 150 testing images in order to measure test accuracy on the training classes in Table 2. Here we use $N$ and $K$ as proxies for hardness, with larger $N$ and smaller $K$ being more hard as the agent must distinguish more classes with fewer training samples/class. Specifically, easy tasks have $(N, K) = (2, 10)$, and the hard tasks have $(N, K) = (20, 1)$. During training, hard tasks are sampled with probability $p$ and easy tasks with probability $1 - p$. Observe that the largest performance gains for MAML in Table 2 are on the hard tasks, consistent with our conclusion that MAML outperforms NAL primarily by achieving relatively high accuracy on the hard tasks. However, the improvement by MAML on the hard tasks disappears when the hard tasks are scarce ($p = 0.01$), supporting the idea of oversampling hard tasks during training in order to improve MAML performance (Zhou et al., 2020; Sun et al., 2020).

# 6 ETHICS STATEMENT

We believe that our paper does not have any potential ethical concerns.

# 7 REPRODUCIBILITY STATEMENT

The proofs for all theoretical results are in the Appendix. Experimental details can also be found in the Appendix, and we have provided our code, as well as instructions on how to use it, in a zip file as part of the supplementary material.

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

APPENDIX

## A PROOFS FOR SECTION 3

### A.1 PROOF OF PROPOSITION 1

For all $i$, let $\mathbf{y}_i^{in} = \mathbf{X}_i^{in}\mathbf{w}_{i,*} + \mathbf{z}_i^{in}$ and $\mathbf{y}_i^{out} = \mathbf{X}_i^{out}\mathbf{w}_{i,*} + \mathbf{z}_i^{out}$. In other words, $\mathbf{z}_i^{in} \in \mathbb{R}^{n_2}$ is the vector containing the additive noise for the inner samples and $\mathbf{z}_i^{out} \in \mathbb{R}^{n_1}$ is the vector containing the additive noise for the outer samples.

From (41) and (43), we have that

$$\mathbf{w}_{NAL}^* = \mathbb{E}_i[\boldsymbol{\Sigma}_i]^{-1}\mathbb{E}_i[\boldsymbol{\Sigma}_i\mathbf{w}_i^*] \tag{14}$$

$$\mathbf{w}_{MAML}^* = \mathbb{E}_i[\mathbf{Q}_i^{(n_2)}]^{-1}\mathbb{E}_i[\mathbf{Q}_i^{(n_2)}\mathbf{w}_i^*], \tag{15}$$

Next, we compute a closed-form expression for $F_m(\mathbf{w})$ (defined in (1)). For all $i$, let $\hat{\mathbf{y}}_i = \hat{\mathbf{X}}_i\mathbf{w}_{i,*} + \hat{\mathbf{z}}_i$ and $y_i = \mathbf{x}_i^\top\mathbf{w}_{i,*} + z_i$. In other words, $\mathbf{z}_i^{in} \in \mathbb{R}^m$ is the vector containing the additive noise for the inner samples and $z_i \in \mathbb{R}$ is the scalar additive noise containing the outer (evaluation) sample. Then we have

$$F(\mathbf{w}) = \frac{1}{2}\mathbb{E}_i\mathbb{E}_{(\mathbf{x}_i,y_i)}\mathbb{E}_{(\hat{\mathbf{X}}_i,\hat{\mathbf{y}}_i)}\left[\left(\left\langle\mathbf{x}_i, \mathbf{w} - \frac{\alpha}{m}\hat{\mathbf{X}}_i^\top(\hat{\mathbf{X}}_i\mathbf{w} - \hat{\mathbf{y}}_i)\right\rangle - y_i\right)^2\right]$$

$$= \frac{1}{2}\mathbb{E}_i\mathbb{E}_{(\mathbf{x}_i,y_i)}\mathbb{E}_{(\hat{\mathbf{X}}_i,\hat{\mathbf{y}}_i)}\left[\left(\langle\mathbf{x}_i, \mathbf{P}_i(\mathbf{w} - \mathbf{w}_i^*)\rangle + (z_i + \frac{\alpha}{m}\mathbf{x}_i^\top\hat{\mathbf{X}}_i^\top\hat{\mathbf{z}}_i)\right)^2\right]$$

$$= \frac{1}{2}\mathbb{E}_i\mathbb{E}_{(\mathbf{x}_i,y_i)}\mathbb{E}_{(\hat{\mathbf{X}}_i,\hat{\mathbf{y}}_i)}\left[\langle\mathbf{x}_i, \mathbf{P}_i(\mathbf{w} - \mathbf{w}_i^*)\rangle^2 + \left(z_i + \frac{\alpha}{m}\mathbf{x}_i^\top\hat{\mathbf{X}}_i^\top\hat{\mathbf{z}}_i\right)^2\right]$$

$$= \frac{1}{2}\mathbb{E}_i\mathbb{E}_{(\mathbf{x}_i,y_i)}\mathbb{E}_{(\hat{\mathbf{X}}_i,\hat{\mathbf{y}}_i)}\left[\langle\mathbf{x}_i, \mathbf{P}_i(\mathbf{w} - \mathbf{w}_i^*)\rangle^2 + \sigma^2 + \frac{\alpha^2}{m^2}\mathrm{tr}(\mathbf{x}_i^\top\hat{\mathbf{X}}_i^\top\hat{\mathbf{z}}_i\hat{\mathbf{z}}_i^\top\hat{\mathbf{X}}_i x_i)\right]$$

$$= \frac{1}{2}\mathbb{E}_i\left[\mathbb{E}_{\hat{\mathbf{X}}_i}\|\mathbf{P}_i(\mathbf{w} - \mathbf{w}_i^*)\|_{\boldsymbol{\Sigma}_i}^2 + \sigma^2 + \frac{\alpha^2\sigma^2}{m}\mathrm{tr}(\boldsymbol{\Sigma}_i^2)\right]$$

where $\mathbf{P}_i := \mathbf{I}_d - \frac{\alpha}{m}\hat{\mathbf{X}}_i^\top\hat{\mathbf{X}}_i$. Hence,

$$\inf_{\mathbf{w}\in\mathbb{R}^d} F_m(\mathbf{w}) = \sigma^2 + \frac{\alpha^2\sigma^2}{m}\mathrm{tr}(\boldsymbol{\Sigma}_i^2) \tag{16}$$

Thus we have

$$\mathcal{E}_m(\mathbf{w}_{NAL}) = F_m(\mathbf{w}_{NAL}) - \frac{1}{2}\mathbb{E}_i\left[\sigma^2 + \frac{\alpha^2\sigma^2}{m}\mathrm{tr}(\boldsymbol{\Sigma}_i^2)\right]$$

$$= \mathbb{E}_i\mathbb{E}_{\hat{\mathbf{X}}_i}\|\mathbf{P}_i(\mathbf{w}_{NAL} - \mathbf{w}_i^*)\|_{\boldsymbol{\Sigma}_i}^2$$

$$= \mathbb{E}_i\mathbb{E}_{\hat{\mathbf{X}}_i}\left[(\mathbf{w}_{NAL} - \mathbf{w}_i^*)^\top\mathbf{P}_i^\top\boldsymbol{\Sigma}_i\mathbf{P}_i(\mathbf{w}_{NAL} - \mathbf{w}_i^*)\right]$$

$$= \mathbb{E}_i\mathbb{E}_{\hat{\mathbf{X}}_i}\left[\left(\mathbb{E}_{i'}[\boldsymbol{\Sigma}_{i'}]^{-1}\mathbb{E}_{i'}[\boldsymbol{\Sigma}_{i'}(\mathbf{w}_{i'}^* - \mathbf{w}_i^*)]\right)^\top\right.$$
$$\left.\mathbf{P}_i^\top\boldsymbol{\Sigma}_i\mathbf{P}_i\mathbb{E}_{i'}[\boldsymbol{\Sigma}_{i'}]^{-1}\mathbb{E}_{i'}[\boldsymbol{\Sigma}_{i'}(\mathbf{w}_{i'}^* - \mathbf{w}_i^*)]\right]$$

$$= \mathbb{E}_i\left[\left(\mathbb{E}_{i'}[\boldsymbol{\Sigma}_{i'}]^{-1}\mathbb{E}_{i'}[\boldsymbol{\Sigma}_{i'}(\mathbf{w}_{i'}^* - \mathbf{w}_i^*)]\right)^\top\right.$$
$$\left.\mathbb{E}_{\hat{\mathbf{X}}_i}\left[\mathbf{P}_i^\top\boldsymbol{\Sigma}_i\mathbf{P}_i\right]\left(\mathbb{E}_{i'}[\boldsymbol{\Sigma}_{i'}]^{-1}\mathbb{E}_{i'}[\boldsymbol{\Sigma}_{i'}(\mathbf{w}_{i'}^* - \mathbf{w}_i^*)]\right)\right]$$

$$= \mathbb{E}_i\left[\left(\mathbb{E}_{i'}[\boldsymbol{\Sigma}_{i'}]^{-1}\mathbb{E}_{i'}[\boldsymbol{\Sigma}_{i'}(\mathbf{w}_{i'}^* - \mathbf{w}_i^*)]\right)^\top\right.$$
$$\left.\mathbf{Q}_i^{(m)}\left(\mathbb{E}_{i'}[\boldsymbol{\Sigma}_{i'}]^{-1}\mathbb{E}_{i'}[\boldsymbol{\Sigma}_{i'}(\mathbf{w}_{i'}^* - \mathbf{w}_i^*)]\right)\right] \tag{17}$$

$$= \mathbb{E}_i\left\|\left(\mathbb{E}_{i'}[\boldsymbol{\Sigma}_{i'}]^{-1}\mathbb{E}_{i'}[\boldsymbol{\Sigma}_{i'}(\mathbf{w}_{i'}^* - \mathbf{w}_i^*)]\right)\right\|_{\mathbf{Q}_i^{(m)}}^2 \tag{18}$$

where (17) follows from Lemma 2. Similarly, for MAML we have the following chain of equalities for $\mathcal{E}_m(\mathbf{w}_{MAML})$ :

$$\mathcal{E}_m(\mathbf{w}_{MAML})$$
$$= F_m(\mathbf{w}_{MAML}) - \frac{1}{2}\mathbb{E}_i\left[\sigma^2 + \frac{\alpha^2\sigma^2}{m}\mathrm{tr}(\mathbf{\Sigma}_i^2)\right]$$
$$= \mathbb{E}_i\mathbb{E}_{\hat{\mathbf{X}}_i}\|\mathbf{P}_i(\mathbf{w}_{MAML} - \mathbf{w}_i^*)\|_{\mathbf{\Sigma}_i}^2$$
$$= \mathbb{E}_i\mathbb{E}_{\hat{\mathbf{X}}_i}\left[(\mathbf{w}_{MAML} - \mathbf{w}_i^*)^\top\mathbf{P}_i^\top\mathbf{\Sigma}_i\mathbf{P}_i(\mathbf{w}_{MAML} - \mathbf{w}_i^*)\right]$$
$$= \mathbb{E}_i\mathbb{E}_{\hat{\mathbf{X}}_i}\left[\left(\mathbb{E}_{i'}[\mathbf{Q}_{i'}^{(n_2)}]^{-1}\mathbb{E}_{i'}[\mathbf{Q}_{i'}^{(n_2)}\mathbf{w}_{i'}^*] - \mathbf{w}_i^*\right)^\top\mathbf{P}_i^\top\mathbf{\Sigma}_i\mathbf{P}_i\left(\mathbb{E}_{i'}[\mathbf{Q}_{i'}^{(n_2)}]^{-1}\mathbb{E}_{i'}[\mathbf{Q}_{i'}^{(n_2)}\mathbf{w}_{i'}^*] - \mathbf{w}_i^*\right)\right]$$
$$= \mathbb{E}_i\mathbb{E}_{\hat{\mathbf{X}}_i}\left[\left(\mathbb{E}_{i'}[\mathbf{Q}_{i'}^{(n_2)}]^{-1}\mathbb{E}_{i'}[\mathbf{Q}_{i'}^{(n_2)}(\mathbf{w}_{i'}^* - \mathbf{w}_i^*)]\right)^\top\right.$$
$$\left.\mathbf{P}_i^\top\mathbf{\Sigma}_i\mathbf{P}_i\left(\mathbb{E}_{i'}[\mathbf{Q}_{i'}^{(n_2)}]^{-1}\mathbb{E}_{i'}[\mathbf{Q}_{i'}^{(n_2)}(\mathbf{w}_{i'}^* - \mathbf{w}_i^*)]\right)\right]$$
$$= \mathbb{E}_i\left[\left(\mathbb{E}_{i'}[\mathbf{Q}_{i'}^{(n_2)}]^{-1}\mathbb{E}_{i'}[\mathbf{Q}_{i'}^{(n_2)}(\mathbf{w}_{i'}^* - \mathbf{w}_i^*)]\right)^\top\right.$$
$$\left.\mathbb{E}_{\hat{\mathbf{X}}_i}\left[\mathbf{P}_i^\top\mathbf{\Sigma}_i\mathbf{P}_i\right]\left(\mathbb{E}_{i'}[\mathbf{Q}_{i'}^{(n_2)}]^{-1}\mathbb{E}_{i'}[\mathbf{Q}_{i'}^{(n_2)}(\mathbf{w}_{i'}^* - \mathbf{w}_i^*)]\right)\right]$$
$$= \mathbb{E}_i\left[\left(\mathbb{E}_{i'}[\mathbf{Q}_{i'}^{(n_2)}]^{-1}\mathbb{E}_{i'}[\mathbf{Q}_{i'}^{(n_2)}(\mathbf{w}_{i'}^* - \mathbf{w}_i^*)]\right)^\top\mathbf{Q}_i^{(m)}\left(\mathbb{E}_{i'}[\mathbf{Q}_{i'}^{(n_2)}]^{-1}\mathbb{E}_{i'}[\mathbf{Q}_{i'}^{(n_2)}(\mathbf{w}_{i'}^* - \mathbf{w}_i^*)]\right)\right]$$
(19)
$$= \mathbb{E}_i\left\|\left(\mathbb{E}_{i'}[\mathbf{Q}_{i'}^{(n_2)}]^{-1}\mathbb{E}_{i'}[\mathbf{Q}_{i'}^{(n_2)}(\mathbf{w}_{i'}^* - \mathbf{w}_i^*)]\right)\right\|_{\mathbf{Q}_i^{(m)}}^2$$
(20)

where (19) follows from Lemma 2.

### A.2 PROOF OF COROLLARY 1

We first write out the dimension-wise excess risks for NAL and MAML, using the results in Proposition 1. Let $w_{i,l}^*$ denote the $l$-th element of the vector $\mathbf{w}_i^*$. Since each $\mathbf{\Sigma}_i$ is diagonal, we have that $\mathbf{Q}_i^{(m)}$ is diagonal. Let $\lambda_{i,l}$ denote the $l$-th diagonal element of the matrix $\mathbf{\Sigma}_i$ and let $q_{i,l}$ denote the $l$-th diagonal element of the matrix $\mathbf{Q}_i^{(m)}$, thus $q_{i,l} = \lambda_{i,l}(1 - \alpha\lambda_{i,l})^2 + \frac{\alpha^2}{m}(\mathrm{tr}(\mathbf{\Sigma}_i^2)\lambda_{i,l} + \lambda_{i,l}^3)$. Recall that we assume $n_2 = m$. The error for NAL in the $l$-th dimension is:

$$\mathcal{E}_m^{(l)}(\mathbf{w}_{NAL}^*) = \frac{1}{2}\mathbb{E}_i\left[q_{i,l}\left(\frac{\mathbb{E}_{i'}\left[\lambda_{i',l}\left(w_{i',l}^* - w_{i,l}^*\right)\right]}{\mathbb{E}_{i'}\left[\lambda_{i',l}\right]}\right)^2\right]$$
$$= \frac{1}{2\mathbb{E}_i\left[\lambda_{i,l}\right]^2}\mathbb{E}_i\left[\mathbb{E}_{i'}\left[\mathbb{E}_{i''}\left[q_{i,l}\lambda_{i',l}\lambda_{i'',l}\left(w_{i',l}^* - w_{i,l}^*\right)\left(w_{i'',l}^* - w_{i,l}^*\right)\right]\right]\right]$$

Likewise, the error for MAML in the $l$-th dimension is:

$$\mathcal{E}_m^{(l)}(\mathbf{w}_{MAML}^*) = \frac{1}{2}\mathbb{E}_i\left[q_{i,l}\left(\frac{\mathbb{E}_{i'}\left[q_{i',l}\left(w_{i',l}^* - w_{i,l}^*\right)\right]}{\mathbb{E}_{i'}\left[q_{i',l}\right]}\right)^2\right]$$
$$= \frac{1}{2\mathbb{E}_i\left[q_{i,l}\right]^2}\mathbb{E}_i\left[\mathbb{E}_{i'}\left[\mathbb{E}_{i''}\left[q_{i,l}q_{i',l}q_{i'',l}\left(w_{i',l}^* - w_{i,l}^*\right)\left(w_{i'',l}^* - w_{i,l}^*\right)\right]\right]\right]$$

Next we use the definitions of $\lambda_{i,l}$ for the easy and hard tasks. For the easy tasks, note that

$$
\begin{aligned}
q_{i,l}^{\text{easy}} &= \lambda_{i,l} \left(1 - \alpha\lambda_{i,l}\right)^2 + \frac{\alpha^2}{m}(\text{tr}(\boldsymbol{\Sigma}_i^2)\lambda_{i,l} + \lambda_{i,l}^3) \\
&= \rho_E(1 - \alpha\rho_E)^2 + \frac{(d+1)\alpha^2\rho_E^3}{m} \\
&=: a_E
\end{aligned}
\tag{21}
$$

and similarly for the hard tasks, namely

$$
q_{i,l}^{\text{hard}} = \rho_H(1 - \alpha\rho_H)^2 + \frac{(d+1)\alpha^2\rho_H^3}{m} =: a_H
\tag{22}
$$

Next, recall that the task distribution is a mixture of the distribution of the hard tasks and the distribution of the easy tasks, with mixture weights $(0.5, 0.5)$. Let $i_H$ be the index of a task drawn from the distribution of hard tasks, and $i_E$ be the index of a task drawn from the distribution of easy tasks. Then using the law of total expectation, the excess risk for MAML in the $l$-th dimension can then be written as:

$$
\begin{aligned}
&\mathcal{E}_m^{(l)}(\mathbf{w}_{MAML}) \\
&= \frac{1}{2}\mathbb{E}_i\left[q_{i,l}\left(\frac{\mathbb{E}_{i'}\left[q_{i',l}\left(w_{i',l}^* - w_{i,l}^*\right)\right]}{\mathbb{E}_{i'}\left[q_{i',l}\right]}\right)^2\right] \\
&= \frac{1}{8\mathbb{E}_i\left[q_{i,l}\right]^2}\mathbb{E}_i\left[q_{i,l}\mathbb{E}_{i'_H}\left[a_H\left(w_{i'_H,l}^* - w_{i,l}^*\right)\right]^2 + q_{i,l}\mathbb{E}_{i_E}\left[a_E\left(w_{i'_E,l}^* - w_{i,l}^*\right)\right]^2\right. \\
&\qquad\qquad\qquad \left. + 2q_{i,l}\mathbb{E}_{i_H}\left[a_H\left(w_{i'_H,l}^* - w_{i,l}^*\right)\right]\mathbb{E}_{i_E}\left[a_E\left(w_{i'_E,l}^* - w_{i,l}^*\right)\right]\right] \\
&= \frac{1}{16\mathbb{E}_i\left[q_{i,l}\right]^2}\mathbb{E}_{i_H}\left[a_H\mathbb{E}_{i'_H}\left[a_H\left(w_{i'_H,l}^* - w_{i_H,l}^*\right)\right]^2 + a_H\mathbb{E}_{i'_E}\left[a_E\left(w_{i'_E,l}^* - w_{i_H,l}^*\right)\right]^2\right. \\
&\qquad\qquad\qquad \left. + 2a_H\mathbb{E}_{i'_H}\left[a_H\left(w_{i'_H,l}^* - w_{i_H,l}^*\right)\right]\mathbb{E}_{i'_E}\left[a_E\left(w_{i'_E,l}^* - w_{i_H,l}^*\right)\right]\right] \\
&\quad + \frac{1}{16\mathbb{E}_i\left[q_{i,l}\right]^2}\mathbb{E}_{i_E}\left[a_E\mathbb{E}_{i'_H}\left[a_E\left(w_{i'_H,l}^* - w_{i_E,l}^*\right)\right]^2 + a_E\mathbb{E}_{i'_E}\left[a_E\left(w_{i'_E,l}^* - w_{i_E,l}^*\right)\right]^2\right. \\
&\qquad\qquad\qquad \left. + 2a_E\mathbb{E}_{i'_H}\left[a_H\left(w_{i'_H,l}^* - w_{i_E,l}^*\right)\right]\mathbb{E}_{i'_E}\left[a_E\left(w_{i'_E,l}^* - w_{i_E,l}^*\right)\right]\right] \\
&= \frac{1}{16\mathbb{E}_i\left[q_{i,l}\right]^2}\left(a_H^3 r_H + a_H a_E^2(r_H + R^2) + 2a_H^2 a_E r_H\right. \\
&\qquad\qquad\qquad \left. + a_E a_H^2(r_E + R^2) + a_E^3 r_E + 2a_E^2 a_H r_E\right) \\
&= \frac{1}{4(a_E + a_H)^2}\left(a_H^3 r_H + a_H a_E^2(r_H + R^2) + 2a_H^2 a_E r_H\right. \\
&\qquad\qquad\qquad \left. + a_E a_H^2(r_E + R^2) + a_E^3 r_E + 2a_E^2 a_H r_E\right)
\end{aligned}
\tag{23}
$$

where (23) follows by the properties of the distributions of the hard and easy tasks. Likewise, for NAL we have

$$
\mathcal{E}_m^{(l)}(\mathbf{w}_{NAL})
$$

$$
= \frac{1}{2}\mathbb{E}_i\left[ q_{i,l}\left( \frac{\mathbb{E}_{i'}\left[\lambda_{i',l}\left(w_{i',l}^* - w_{i,l}^*\right)\right]}{\mathbb{E}_{i'}\left[\lambda_{i',l}\right]} \right)^2 \right]
$$

$$
= \frac{1}{16\mathbb{E}_i\left[\lambda_{i,l}\right]^2}\mathbb{E}_{i_H}\left[ a_H\mathbb{E}_{i_H'}\left[\rho_H\left(w_{i_H',l}^* - w_{i_H,l}^*\right)\right]^2 + a_H\mathbb{E}_{i_E'}\left[\rho_E\left(w_{i_E',l}^* - w_{i_H,l}^*\right)\right]^2 \right.
$$

$$
\left. + 2a_H\mathbb{E}_{i_H'}\left[\rho_H\left(w_{i_H',l}^* - w_{i_H,l}^*\right)\right]\mathbb{E}_{i_E'}\left[\rho_E\left(w_{i_E',l}^* - w_{i_H,l}^*\right)\right] \right]
$$

$$
+ \frac{1}{16\mathbb{E}_i\left[\lambda_{i,l}\right]^2}\mathbb{E}_{i_E}\left[ a_E\mathbb{E}_{i_H'}\left[\rho_E\left(w_{i_H',l}^* - w_{i_E,l}^*\right)\right]^2 + a_E\mathbb{E}_{i_E'}\left[\rho_E\left(w_{i_E',l}^* - w_{i_E,l}^*\right)\right]^2 \right.
$$

$$
\left. + 2a_E\mathbb{E}_{i_H'}\left[a_H\left(w_{i_H',l}^* - w_{i_E,l}^*\right)\right]\mathbb{E}_{i_E'}\left[a_E\left(w_{i_E',l}^* - w_{i_E,l}^*\right)\right] \right]
$$

$$
= \frac{1}{4(a_E + a_H)^2}\Big( a_H\rho_H^2 r_H + a_H\rho_E^2(r_H + R^2) + 2a_H\rho_H\rho_E r_H +
$$

$$
a_E\rho_H^2(r_E + R^2) + a_E\rho_E^2 r_E + 2a_E\rho_H\rho_E \Big)
$$

Note that in the this setting the excess risks are symmetric across dimension. Thus multiplying by $d$ completes the proof.

### A.3 PROOF OF COROLLARY 2

For the case that $\alpha = 1/\rho_E$ and $\frac{\rho_H}{\rho_E}(1 - \frac{\rho_H}{\rho_E})^2 \gg \frac{d}{m}$, we can compare MAML and NAL by the weights that are placed on $r_H$, $r_E$, $r_H + R^2$ and $r_E + R^2$ in their excess risks. Using the fact that $\alpha = 1/\rho_E$, the expressions in (21) and (22) can be simplified as

$$
a_E = \frac{(d+1)\rho_E}{m}
$$

$$
a_H = \rho_H(1 - \frac{\rho_H}{\rho_E})^2 + \frac{(d+1)\rho_H^3}{m\rho_E^2} \tag{24}
$$

Under the assumption that $\frac{\rho_H}{\rho_E}(1 - \frac{\rho_H}{\rho_E})^2 \gg \frac{d}{m}$, the bias term $\rho_H(1 - \frac{\rho_H}{\rho_E})^2$ dominates the gradient variance term $\frac{(d+1)\rho_E}{m}$, thus $a_H$ dominates $a_E$. As a result, we can ignore $a_E$ in the expressions. Furthermore, $\rho_H(1 - \frac{\rho_H}{\rho_E})^2$ dominates $\frac{(d+1)\rho_H^3}{m\rho_E^2}$ within the expression for $a_H$, meaning we can also drop the latter term. Thus we have

$$
\mathcal{E}_m(\mathbf{w}_{MAML}^*) \approx \frac{da_H^3}{4a_H^2}r_H = \frac{d\rho_H(1 - \frac{\rho_H}{\rho_E})^2}{4}r_H \tag{25}
$$

and

$$
\mathcal{E}_m(\mathbf{w}_{NAL}^*) \approx \frac{da_H\rho_H^2 + 2da_H\rho_E\rho_H}{4(\rho_H + \rho_E)^2}r_H + \frac{da_H\rho_E^2}{4(\rho_H + \rho_E)^2}(r_H + R^2)
$$

$$
= \frac{d\rho_H(1 - \frac{\rho_H}{\rho_E})^2}{4}r_H + \frac{d\rho_H\rho_E^2(1 - \frac{\rho_H}{\rho_E})^2}{4(\rho_H + \rho_E)^2}R^2 \tag{26}
$$

Dividing (26) by (25) yields

$$
\frac{\mathcal{E}_m(\mathbf{w}^*_{NAL})}{\mathcal{E}_m(\mathbf{w}^*_{MAML})} \approx \left( \frac{d\rho_H(1 - \frac{\rho_H}{\rho_E})^2}{4} r_H + \frac{d\rho_H \rho_E^2 (1 - \frac{\rho_H}{\rho_E})^2}{4(\rho_H + \rho_E)^2} R^2 \right) \bigg/ \left( \frac{d\rho_H(1 - \frac{\rho_H}{\rho_E})^2}{4} r_H \right)
$$

$$
= \left( r_H + \frac{\rho_E^2}{(\rho_H + \rho_E)^2} R^2 \right) \bigg/ r_H
$$

$$
= 1 + \frac{\rho_E^2}{(\rho_H + \rho_E)^2} \frac{R^2}{r_H^2}
$$

$$
\approx 1 + \frac{R^2}{r_H^2} \tag{27}
$$

where the last approximation follows since $\frac{1}{4} \le \frac{\rho_E^2}{(\rho_E + \rho_H)^2} \le 1$.

## B  PROOF OF THEOREM 1

In this section we prove Theorem 1. We first show a general result.

**Lemma 1.** *Let $\mathbf{w}_{MAML} \in \mathbb{R}^D$ be a stationary point of the MAML objective 1 with $m = \infty$ whose Hessians of the task loss functions are bounded and positive definite, i.e. $\mathbf{w}_{MAML}$ satisfies*

$$
\beta_1 \mathbf{I}_D \preceq \nabla^2 f_1(\mathbf{w}_{MAML}) \preceq L_1 \mathbf{I}_D, \quad \beta_2 \mathbf{I}_D \preceq \nabla^2 f_2(\mathbf{w}_{MAML}) \preceq L_2 \mathbf{I}_D \tag{28}
$$

*for some $L_1 \ge \beta_1 > 0$ and $L_2 \ge \beta_2 > 0$. Suppose that $\beta_2 > \beta_1$. Define $\mathbf{w}_1 \coloneqq \mathbf{w}_{MAML} - \alpha \nabla f_1(\mathbf{w}_{MAML})$ and $\mathbf{w}_2 \coloneqq \mathbf{w}_{MAML} - \alpha \nabla f_2(\mathbf{w}_{MAML})$. Then the following holds for any $\alpha < 1/\max(L_1, L_2)$:*

$$
\|\nabla f_1(\mathbf{w}_1)\|_2 \lesssim \frac{1 - \alpha\beta_2}{1 - \alpha L_1} \|\nabla f_2(\mathbf{w}_2)\| + O(\alpha^2). \tag{29}
$$

*Proof.* For some vector $\mathbf{w}_{MAML} \in \mathbb{R}^D$, let $\mathbf{H}_1(\mathbf{w}_{MAML}) \coloneqq \nabla^2 f_1(\mathbf{w}_{MAML})$ and $\mathbf{H}_2(\mathbf{w}_{MAML}) = \nabla^2 f_2(\mathbf{w}_{MAML})$ and $\mathbf{g}_1(\mathbf{w}_{MAML}) = \nabla f_1(\mathbf{w}_1)$ and $\mathbf{g}_2(\mathbf{w}_{MAML}) = \nabla f_2(\mathbf{w}_2)$, recalling that $\mathbf{w}_1 = \mathbf{w}_{MAML} - \alpha \nabla f_1(\mathbf{w}_{MAML})$ and $\mathbf{w}_2 = \mathbf{w}_{MAML} - \alpha \nabla f_2(\mathbf{w}_{MAML})$.

Recall that the MAML objective in this case is given by

$$
F_\infty(\mathbf{w}) = \frac{1}{2} f_1(\mathbf{w} - \alpha \nabla f_1(\mathbf{w})) + \frac{1}{2} f_2(\mathbf{w} - \alpha \nabla f_2(\mathbf{w})). \tag{30}
$$

After setting the gradient of $F_\infty(.)$ equal to zero at $\mathbf{w}_{MAML}$, we find that any stationary point $\mathbf{w}_{MAML}$ of the MAML objective must satisfy

$$
(\mathbf{I}_D - \alpha \mathbf{H}_1(\mathbf{w}_{MAML}))\mathbf{g}_1(\mathbf{w}_1) = -(\mathbf{I}_D - \alpha \mathbf{H}_2(\mathbf{w}_{MAML}))\mathbf{g}_2(\mathbf{w}_2) \tag{31}
$$

Since $\mathbf{H}_2(\mathbf{w}_{MAML}) \preceq L_2 \mathbf{I}_d$ and $\alpha \le 1/\max(L_1, L_2)$, $\mathbf{I}_d - \alpha \mathbf{H}_2(\mathbf{w}_{MAML})$ is positive definite with minimum eigenvalue at least $1 - \alpha L_2 > 0$. Thus we have

$$
(\mathbf{I}_D - \alpha \mathbf{H}_2(\mathbf{w}_{MAML}))^{-1}(\mathbf{I}_D - \alpha \mathbf{H}_1(\mathbf{w}_{MAML}))\mathbf{g}_1(\mathbf{w}_1) = -\mathbf{g}_2(\mathbf{w}_2) \tag{32}
$$

Taking the norm of both sides and using the fact that $\|\mathbf{A}\mathbf{v}\|_2 \ge \sigma_{\min}(\mathbf{A})\|\mathbf{v}\|_2$ for any matrix $\mathbf{A}$ and vector $\mathbf{v}$, along with the facts that $\beta_1 \mathbf{I}_D \preceq \mathbf{H}_1(\mathbf{w}_{MAML}) \preceq L_1 \mathbf{I}_D$ and $\beta_2 \mathbf{I}_D \preceq \mathbf{H}_2(\mathbf{w}_{MAML}) \preceq L_2 \mathbf{I}_D$ by assumption, yields

$$
\begin{aligned}
\|\mathbf{g}_2(\mathbf{w}_2)\|_2 &= \|(\mathbf{I}_D - \alpha \mathbf{H}_2(\mathbf{w}_{MAML}))^{-1}(\mathbf{I}_D - \alpha \mathbf{H}_1(\mathbf{w}_{MAML}))\mathbf{g}_1(\mathbf{w}_1)\|_2 \\
&\ge \sigma_{\max}^{-1}(\mathbf{I}_D - \alpha \mathbf{H}_2(\mathbf{w}_{MAML}))\sigma_{\min}(\mathbf{I}_D - \alpha \mathbf{H}_1(\mathbf{w}_{MAML}))\|\mathbf{g}_1(\mathbf{w}_1)\|_2 \\
&\ge \frac{1 - \alpha L_1}{1 - \alpha\beta_2} \|\mathbf{g}_1(\mathbf{w}_1)\|_2 \tag{33}
\end{aligned}
$$

Next, note that using Taylor expansion,

$$
\begin{aligned}
\mathbf{g}_1(\mathbf{w}_1) &= \nabla f_1(\mathbf{w}_{MAML}) - \alpha \nabla^2 f_1(\mathbf{w}_{MAML})\nabla f_1(\mathbf{w}_{MAML}) + O(\alpha^2) \\
&= (\mathbf{I}_D - \alpha \mathbf{H}_1(\mathbf{w}_{MAML}))\mathbf{g}_1(\mathbf{w}_{MAML}) \tag{34}
\end{aligned}
$$

$$
\begin{aligned}
\mathbf{g}_2(\mathbf{w}_2) &= \nabla f_2(\mathbf{w}_{MAML}) - \alpha \nabla^2 f_2(\mathbf{w}_{MAML})\nabla f_2(\mathbf{w}_{MAML}) + O(\alpha^2) \\
&= (\mathbf{I}_D - \alpha \mathbf{H}_2(\mathbf{w}_{MAML}))\mathbf{g}_2(\mathbf{w}_{MAML}) \tag{35}
\end{aligned}
$$

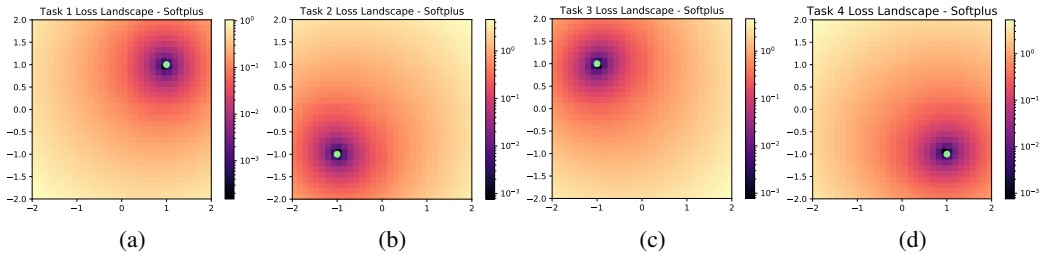

Figure 5: Task loss functions $f_1, f_2, f_3, f_4$ for task environment in Figure 4 (a-b).

Therefore

$$\|\mathbf{g}_2(\mathbf{w}_{MAML})\|_2 + O(\alpha^2)$$

$$\gtrsim \sigma_{\max}^{-1}(\mathbf{I}_D - \alpha\mathbf{H}_2(\mathbf{w}_{MAML}))\sigma_{\min}(\mathbf{I}_D - \alpha\mathbf{H}_1(\mathbf{w}_{MAML}))\frac{1-\alpha L_1}{1-\alpha\beta_2}\|\mathbf{g}_1(\mathbf{w}_1)\|_2$$

$$\gtrsim \left(\frac{1-\alpha L_1}{1-\alpha\beta_2}\right)^2 \|\mathbf{g}_1(\mathbf{w}_{MAML})\|_2, \tag{36}$$

i.e., $\|\nabla f_1(\mathbf{w}_{MAML})\|_2 \lesssim \left(\frac{1-\alpha\beta_2}{1-\alpha L_1}\right)^2 \|\nabla f_2(\mathbf{w}_{MAML})\|_2 + O(\alpha^2)$. $\qquad\square$

Now we are ready to prove Theorem 1.

*Proof.* We first show that $\beta_i \mathbf{I}_{Nd} \preceq \nabla^2 f_i(\mathbf{W}) \preceq L_i \mathbf{I}_{Nd}$ for all $\mathbf{W} \in \mathcal{S}_i$ for $i \in \{1, 2\}$, where $\nabla^2 f_i(\mathbf{W}) \in \mathbb{R}^{Nd \times Nd}$ is the Hessian of the loss of the vectorized $\mathbf{W}$ and each $\beta_i$ and $L_i$ is defined in Theorem 1. In other words, we will show that each $f_i$ is $\beta_i$-strongly convex and $L_i$-smooth within $\mathcal{S}_i$. This will show that any stationary point $\mathbf{W}_{MAML}$ that lies within $\mathcal{S}_1 \cap \mathcal{S}_2$ satisfies the conditions of Lemma 1.

For any $i \in \{1, 2\}$ and for any $\mathbf{W} \in \mathbb{R}^{d \times N}$ we have by Weyl's Inequality

$$\nabla^2 f_i(\mathbf{W}_{i,*}) - \|\nabla^2 f_i(\mathbf{W}) - \nabla^2 f_i(\mathbf{W}_{i,*})\|_2 \mathbf{I}_{Nd}$$
$$\preceq \nabla^2 f_i(\mathbf{W}) \preceq \nabla^2 f_1(\mathbf{W}_{i,*}) + \|\nabla^2 f_i(\mathbf{W}) - \nabla^2 f_i(\mathbf{W}_{i,*})\|_2 \mathbf{I}_{Nd} \tag{37}$$

Thus, we will control the spectrum of $\nabla^2 f_i(\mathbf{W})$ by controlling the spectrum of $\nabla^2 f_i(\mathbf{W}_{i,*})$ and by upper bounding $\|\nabla^2 f_i(\mathbf{W}) - \nabla^2 f_i(\mathbf{W}_{i,*})\|_2$.

To control the spectrum of $\nabla^2 f_i(\mathbf{W}_{i,*})$ we use Lemma D.3 from Zhong et al. (2017), which shows that for absolute constants $c_1$ and $c_2$ and each of the possible $\sigma$ functions listed in Theorem 1,

$$(c_1/(\kappa^2 \lambda))\mathbf{I}_{Nd} \preceq \nabla^2 f_1(\mathbf{W}_{i,*}) \preceq (c_2 N s_{i,1}^{2c})\mathbf{I}_{Nd}. \tag{38}$$

Next, we apply Lemma D.10 from Zhong et al. (2017), which likewise applies for all the mentioned $\sigma$, to obtain

$$\|\nabla^2 f_i(\mathbf{W}) - \nabla^2 f_i(\mathbf{W}_{i,*})\|_2 \le c_3 N^2 s_{i,1}^c \|\mathbf{W} - \mathbf{W}_{i,*}\|_2 \tag{39}$$

for a constant $c_3$. Next, for any $\mathbf{W} \in \mathcal{S}_i$, we have $\|\mathbf{W} - \mathbf{W}_{i,*}\|_2 = O(1/(s_{i,1}^c \lambda_i \kappa_i^2 N^2))$. Therefore, by combining (37), (38) and (39), we obtain that $\beta_i \mathbf{I}_{Nd} \preceq \nabla^2 f_i(\mathbf{W}) \preceq L_i \mathbf{I}_{Nd}$ for all $\mathbf{W} \in \mathcal{S}_i$.

Finally we apply Lemma 1 to complete the proof. $\qquad\square$

Note that the constant $c$ used in Theorem 1 is the homogeneity constant for which $\sigma$ satisfies Property 3.1 from Zhong et al. (2017), namely, $0 \le \sigma'(z) \le a|z|^c \ \forall z \in \mathbb{R}$.

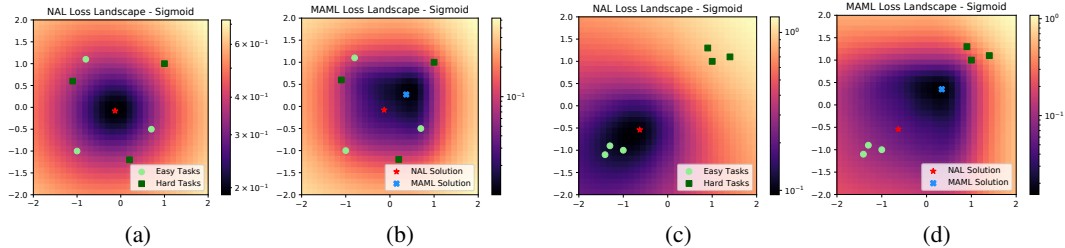

Figure 6: Loss landscapes for NAL (a,c) and MAML (b,d) for two distinct task environments and Sigmoid activation.

Table 3: Effect of up-weighting hard tasks for NAL.

|  | NAL | NAL, $\nu = 2$ | NAL, $\nu = 5$ | NAL, $\nu = 10$ | MAML |
|---|---|---|---|---|---|
| Avg. coord. | $0.18 \pm .02$ | $0.33 \pm .02$ | $0.61 \pm .03$ | $0.87 \pm .03$ | $1.58 \pm 0.01$ |
| Test error | $0.78 \pm .11$ | $0.64 \pm .08$ | $0.51 \pm 04$ | $0.39 \pm .05$ | $0.26 \pm 0.10$ |

## C  ADDITIONAL EXPERIMENTS AND DETAILS

### C.1  LINEAR REGRESSION

In all of the linear regression experiments, to run SGD on the MAML and NAL objectives, we sample one task from the corresponding environment on each iteration for 5,000 iterations. Each task has $n_1 = 25$ outer loop samples and varying $n_2$ inner loop samples for MAML, and $n = n_1 + n_2$ samples for NAL. We appropriately tuned the 'meta-learning rates', i.e. the learning rate with which $\mathbf{w}_{MAML}$ and $\mathbf{w}_{NAL}$ are updated after each full iteration, and used $n_2/10000$ for MAML and $0.025$ for NAL. After 5,000 iterations, the excess risks of the final iterates were estimated using 3,000 randomly samples from the environment. We repeated this procedure ten times to obtain standard deviations.

We also ran an experiment to test whether up-weighting the hard tasks improves NAL performance, in light of our observation that MAML achieves performance gain by initializing closer to the hard task solutions. We use a similar environment as in Section 3.2. Tasks are 10-dimensional linear regression problems with $n_2 = 25$ inner loop samples and $n_1 = 500$ outer loop samples, and noise variance 0.01. To implement up-weighting for NAL, we introduce a parameter $\nu$ which is the ratio of the weight placed on the hard tasks to the weight placed on the easy tasks within each batch of tasks. We normalize the weights to sum to 1. For example, if a task batch consists of 6 hard tasks and 4 easy tasks, then NAL with $\nu = 2$ places a weight of $\frac{\nu}{6\nu+4} = \frac{1}{8}$ on the hard task loss functions and $\frac{1}{6\nu+4} = \frac{1}{16}$ on the easy task loss functions (as opposed to $\frac{1}{10}$ on all tasks for standard NAL).

Easy tasks have hardness parameter $\rho_E = 1$ and optimal solution drawn from $\mathcal{N}(\mathbf{0}_d, \mathbf{I}_d)$, and hard tasks have hardness parameter $\rho_H = 0.1$ and optimal solution drawn from $\mathcal{N}(2\mathbf{1}_d, \mathbf{I}_d)$. We run NAL and MAML for 4000 iterations and use a task-batch size of 10 tasks per iteration, sampling easy and hard tasks with equal probability. We report the average coordinate value for the final solutions $\mathbf{w}_{NAL}$ and $\mathbf{w}_{MAML}$ and their test error (averaged across randomly sampled hard and easy tasks), plus or minus standard deviation over 5 independent random trials.

Note that average coordinate value closer to 2 means the solution is closer to optimal solutions of the hard tasks, while closer to 0 means it is closer to the optimal solutions of the easy tasks. Indeed we see that when NAL places more emphasis on the hard tasks, i.e. $\nu$ is large, its performance correspondingly increases and approaches that of MAML. Indeed, it illustrates that MAML can be interpreted as a reweighing of tasks based on their level of hardness for GD.

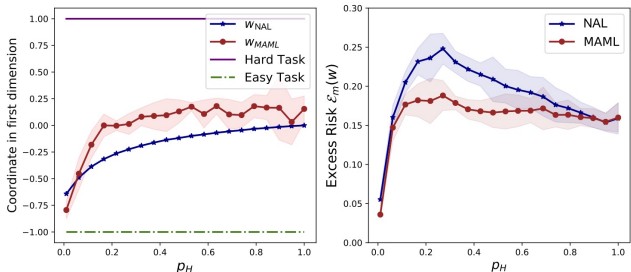

Figure 7: **Logistic regression** results in analogous setting to $m = 500$ column in Figure 2 ($T = 2$ tasks, $d = 10$ dimensions). Recall that $\rho_H$ is the strong convexity parameter (data variance) for the hard task, which determines its hardness, while the strong convexity parameter of the easy task is 1. MAML again initializes closer to the harder task, and has smaller excess risk for appropriate $\rho_H$.

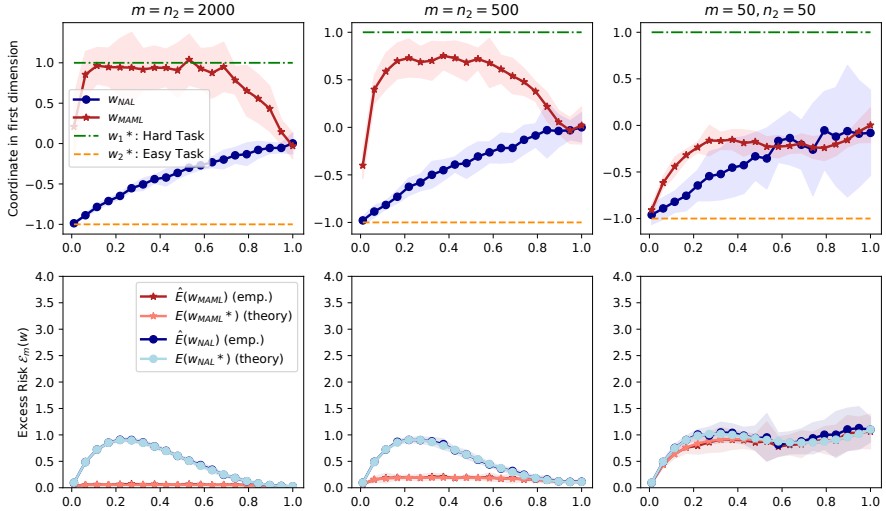

Figure 8: Version of Figure 2 with corresponding empirical results, including 95% confidence intervals. The hardness parameter $\rho_H$ varies along the $x$-axis.

## C.2 LOGISTIC REGRESSION

We also experimented with logistic regression, please see Figure 7 for details.

## C.3 ONE-LAYER NEURAL NETWORKS

We approximate loss landscapes for two types of activations: Softplus (Figures 4 and 5) and Sigmoid (Figure 6). To approximate each landscape, we sample Gaussian data as specified in Section 4 and compute the corresponding empirical losses as in equation (3) for NAL and (4) for MAML. We use $n = 500$ for NAL and $n_1 = 20$ and $\tau = 25$ for MAML in all cases. We use $n_2 = m = 250$ for Softplus and $n_2 = m = 80$ for Sigmoid. *Figure 6 shows that when the hard and easy task solutions have similar centroids ($R$ is small, as in subfigures (a)-(b)), then the NAL and MAML solutions are close and achieve similar post-adaptation loss (b). On the other hand, if the centroids are spread and the hard tasks are close (large $R$, small $r_H$ as in subfigures (c)-(d)), then the NAL and MAML solutions are far apart and MAML obtains significantly smaller post-adaptation loss (d).*

| Setting | | Train Tasks | | Test Tasks | |
|---|---|---|---|---|---|
| $r_H$ | ALG | EASY | HARD | EASY | HARD |
| LARGE | MAML | $99.2\pm.2$ | $96.0\pm.6$ | $98.0\pm.2$ | $81.2\pm.3$ |
| LARGE | NAL-1 | $69.4\pm.4$ | $41.5\pm.3$ | $57.8\pm.4$ | $45.2\pm.8$ |
| LARGE | NAL-10 | $70.0\pm.8$ | $45.3\pm.9$ | $67.2\pm.3$ | $47.9\pm.7$ |
| SMALL | MAML | $99.2\pm.5$ | $99.1\pm.2$ | $98.1\pm.3$ | $95.4\pm.3$ |
| SMALL | NAL-1 | $69.2\pm.5$ | $46.0\pm.6$ | $55.8\pm.5$ | $45.8\pm1.0$ |
| SMALL | NAL-10 | $70.2\pm.6$ | $44.0\pm.8$ | $67.8\pm.8$ | $48.9\pm.8$ |

Table 4: Omniglot accuracies with 95% confidence intervals.

| Setting | | Train Tasks | | Test Tasks | |
|---|---|---|---|---|---|
| $p$ | Alg. | Easy | Hard | Easy | Hard |
| 0.99 | NAL | $78.2\pm.4$ | $9.9\pm1.0$ | $74.8\pm.4$ | $9.5\pm.6$ |
| | MAML | $92.9\pm.3$ | $50.1\pm1.0$ | $84.6\pm.4$ | $21.7\pm.3$ |
| 0.5 | NAL | $81.9\pm.3$ | $9.5\pm1.0$ | $76.5\pm.4$ | $8.6\pm.3$ |
| | MAML | $93.6\pm.5$ | $41.1\pm.3$ | $84.3\pm.6$ | $17.9\pm.4$ |
| 0.01 | NAL | $82.4\pm1.4$ | $7.6\pm.2$ | $76.9\pm.9$ | $8.2\pm.4$ |
| | MAML | $94.0\pm1.2$ | $15.7\pm.3$ | $85.0\pm.2$ | $11.8\pm.3$ |

Table 5: FS-CIFAR100 accuracies with 95% confidence intervals.

## C.4 OMNIGLOT

The full version of Table 1, with error bounds, is given in Table 4.

We use the same 4-layer convolutional neural network architecture as in (Finn et al., 2017), using code adapted from the code that implements in PyTorch the experiments in the paper (Antoniou et al., 2018). We ran SGD on the MAML and NAL objectives using 10 target samples per class for MAML, i.e. $n_1 = 5 \times 10$. Likewise, $n = n_1 + 5 \times n_2$ samples were used in each task to update $\mathbf{w}_{NAL}$ on each iteration, for $n_2 = 5 \times K$. Eight tasks were drawn on each iteration for a total of 20,000 iterations. The outer-loop learning rate for MAML was tuned in $\{10^{-2}, 10^{-3}, 10^{-4}, 10^{-5}, 10^{-6}\}$ and selected as $10^{-3}$. Similarly, the learning rate for NAL was tuned in $\{10^{-2}, 10^{-3}, 10^{-4}, 10^{-5}, 10^{-6}\}$ and selected as $10^{-6}$. Both MAML and NAL used a task-specific adaptation (inner) learning rate of $10^{-1}$ as in Antoniou et al. (2018). To select the alphabets corresponding to hard tasks, we ran MAML on tasks drawn from all 50 Omniglot alphabets, and took the 10 alphabets with the lowest accuracies. The train/test split for the other (easy) alphabets was random among the remaining alphabets.

To compute the accuracies in Table 1, we randomly sample 500 tasks from the training classes and 500 from the testing classes, with easy tasks and hard tasks being chosen with equal probability, and take the average accuracies after task-specific adaption from the fixed, fully trained model for each set of sampled tasks. MAML uses 1 step of SGD for task-specific adaptation during both training and testing, and NAL uses 1 for testing. The entire procedure was repeated 5 times with different random seeds to compute the average accuracies in Table 1 and the confidence bounds in Figure 4.

## C.5 FS-CIFAR100

First, we note a typo from the main body: we used 5x as many samples for task-specific adaptation as stated in the main body for both easy and hard tasks, that is, $(N, K) = (2, 50)$ for easy tasks and $(N, K) = (20, 5)$ for hard tasks.

The full version of Table 2 with error bounds is given in Table 5.

We use the same CNN as for Omniglot but with a different number of input nodes and channels to account for the larger-sized CIFAR images, which are 32-by-32 RGB images (Omniglot images are 28-by-28 grayscale). We ran MAML and NAL (SGD on the MAML and NAL objectives, respectively) using 10 target (outer loop) samples per class per task for MAML, and 15 samples per class per task for NAL (recalling that NAL has no inner loop samples). Eight tasks were drawn on

each iteration for a total of 20,000 iterations. The outer-loop learning rate for MAML was tuned in $\{10^{-2}, 10^{-3}, 10^{-4}, 10^{-5}, 10^{-6}\}$ and selected as $10^{-3}$. Similarly, the learning rate for NAL was tuned in $\{10^{-2}, 10^{-3}, 10^{-4}, 10^{-5}, 10^{-6}\}$ and selected as $10^{-6}$. Both MAML and NAL used a task-specific adaptation (inner loop) learning rate of $10^{-1}$ as in Antoniou et al. (2018).

We trained for 10,000 iterations with a task batch size of 2, with hard tasks being chosen with probability $p$ and easy tasks with probability $1-p$. As in the Omniglot experiment, after completing training we randomly sample 500 tasks from the training classes and 500 from the testing classes, with easy tasks and hard tasks being chosen with equal probability. We then take the average accuracies after task-specific adaption from the fixed, fully trained model for each set of sampled tasks. NAL uses 5 steps of task-specific adaptation for testing and MAML uses 1 for both training and testing. The entire procedure was repeated 5 times with different random seeds to compute average accuracies in Table 2 and the confidence bounds in Figure 5.

## D    MULTI-TASK LINEAR REGRESSION CONVERGENCE RESULTS

We motivate our analysis of the NAL and MAML population-optimal solutions for multi-task linear regression by showing that their respective empirical training solutions indeed converge to their population-optimal values.

First note that the empirical training problem for NAL can be written as

$$\min_{\mathbf{w} \in \mathbf{R}^d} \frac{1}{T} \sum_{i=1}^{T} \|\mathbf{X}_i(\mathbf{w} - \mathbf{w}_{i,*}) - \mathbf{z}_i\|_2^2, \tag{40}$$

where the $j$-element of $\mathbf{z}_i \in \mathbb{R}^n$ contains the noise for the $j$-th sample for task $\mathcal{T}_i$. Taking the derivative with respect to $\mathbf{w}$ and setting it equal to zero yields that the NAL training solution is:

$$\mathbf{w}_{NAL} = \left( \sum_{i=1}^{T} \mathbf{X}_i^\top \mathbf{X}_i \right)^{-1} \sum_{i=1}^{T} \left( \mathbf{X}_i^\top \mathbf{X}_i \mathbf{w}_i^* + \mathbf{X}_i^\top \mathbf{z}_i \right), \tag{41}$$

assuming $\sum_{i=1}^{T} \mathbf{X}_i^\top \mathbf{X}_i$ is invertible. Similarly, using $\mathbf{z}_i^{out}$ and $\mathbf{z}_i^{in}$ to denote noise vectors, as well as $\hat{\mathbf{Q}}_{i,j} = \hat{\mathbf{P}}_{i,j}(\mathbf{X}_{i,j}^{out})^\top \mathbf{X}_{i,j}^{out} \hat{\mathbf{P}}_{i,j}$ where $\hat{\mathbf{P}}_{i,j} = \mathbf{I}_d - \frac{\alpha}{n_2}(\mathbf{X}_{i,j}^{in})^\top \mathbf{X}_{i,j}^{in}$, we have that the empirical training problem for MAML is

$$\min_{\mathbf{w} \in \mathbf{R}^d} \frac{1}{T\tau} \sum_{i=1}^{T} \sum_{j=1}^{\tau} \|\mathbf{X}_i^{out} \hat{\mathbf{P}}_{i,j}(\mathbf{w} - \mathbf{w}_{i,*}) - \mathbf{z}_{i,j}^{out} - \frac{\alpha}{n_2} \mathbf{X}_{i,j}^{out}(\mathbf{X}_{i,j}^{in})^\top \mathbf{z}_{i,j}^{in}\|_2^2, \tag{42}$$

therefore

$$\mathbf{w}_{MAML}$$
$$= \left( \sum_{i=1}^{T} \sum_{j=1}^{\tau} \hat{\mathbf{Q}}_{i,j} \right)^{-1} \sum_{i=1}^{T} \sum_{j=1}^{\tau} \left( \hat{\mathbf{Q}}_{i,j} \mathbf{w}_i^* + \hat{\mathbf{P}}_{i,j}(\mathbf{X}_{i,j}^{out})^\top \mathbf{z}_{i,j}^{out} - \frac{\alpha}{n_2} \hat{\mathbf{P}}_{i,j}(\mathbf{X}_{i,j}^{out})^\top \mathbf{X}_{i,j}^{out}(\mathbf{X}_{i,j}^{in})^\top \mathbf{z}_{i,j}^{in} \right) \tag{43}$$

To show that $\mathbf{w}_{NAL}$ and $\mathbf{w}_{MAML}$ indeed converge to their population-optimal values as $T, n, n_1, \tau \to \infty$, we first make the following regularity assumptions.

**Assumption 1.** *There exists $B > 0$ s.t. $\|\mathbf{w}_i^*\| \leq B \ \forall i$.*

**Assumption 2.** *There exists $\beta, L > 0$ s.t. $\beta \mathbf{I}_d \preceq \mathbf{\Sigma}_i \preceq L \mathbf{I}_d \ \forall i$.*

Assumption 1 ensures that the task optimal solutions have bounded norm and Assumption 2 ensures that the data covariances are positive definite with bounded spectral norm.

**Remark 2.** *We would like to note that Gao & Sener (2020) achieve a similar results as our Theorem 2 and 3. However, we arrive at our results using distinct techniques from theirs. Moreover, our MAML convergence result (Theorem 3) accounts for convergence over task instances to the population-optimal solution for MAML when a finite number of samples are allowed for task-specific adaptation (a stochastic gradient step), whereas the analogous result in Gao & Sener (2020) (Theorem 2) does*

*not: it assumes $\tau = 1$ and shows convergence as $n_1 = n_2 \to \infty$. Since their result relies on $n_2 \to \infty$, it shows convergence to the population-optimal MAML solution when an* infinite *amount of samples are allowed for the inner task-specific update, i.e. a full gradient step. Our dimension-dependence is significantly worse, than theirs, which suggests the extra complexity of the MAML objective with finite samples allowed for task-specific adaptation.*

### D.1 NAL CONVERGENCE

Define $\mathrm{Var}(\boldsymbol{\Sigma}_i) := \|\mathbb{E}_i[(\boldsymbol{\Sigma}_i - \mathbb{E}_{i'}[\boldsymbol{\Sigma}_{i'}])^2]\|$ and $\mathrm{Var}(\boldsymbol{\Sigma}_i \mathbf{w}_{i,*}) := \|\mathbb{E}_i[(\boldsymbol{\Sigma}_i \mathbf{w}_{i,*} - \mathbb{E}_{i'}[\boldsymbol{\Sigma}_{i'} \mathbf{w}_{i,*}])^2]\|$.

**Theorem 2.** *(NAL Convergence) Under Assumptions 1 and 2, the distance of the NAL training solution (41) to its population-optimal value (5) is bounded as*

$$\|\mathbf{w}_{NAL} - \mathbf{w}^*_{NAL}\|_2 \leq \left( \frac{c'\sqrt{d} + \beta\sqrt{\frac{dK}{c''}\log(200n)} + \frac{K}{c''}\log(200n)}{\beta^2\sqrt{n}} + \frac{\sqrt{\log(200T)}L^2 B}{\beta^2 n} \right)$$
$$+ \frac{\sqrt{Var(\boldsymbol{\Sigma}_i \mathbf{w}_{i,*})\log(200d)}}{\beta\sqrt{T}} + \frac{LB\sqrt{Var(\boldsymbol{\Sigma}_i)\log(200d)}}{\beta^2\sqrt{T}} \tag{44}$$

*with probability at least 0.96, where $K$ is the maximum sub-exponential norm of pairwise products of data and noise samples, for some absolute constants $c, c'$, and $n \geq 4\left( \frac{cL\sqrt{d} + \sqrt{c^2 L^2 d + 4\beta cL\sqrt{\log(200T)}}}{2\beta} \right)^2$ and $T > L^2 B^2 (Var(\boldsymbol{\Sigma}_i) + Var(\boldsymbol{\Sigma}_i \mathbf{w}_i^*))/9$. Informally,*

$$\|\mathbf{w}_{NAL} - \mathbf{w}^*_{NAL}\|_2 \leq \tilde{\mathcal{O}}\left( \frac{\sqrt{d}}{\sqrt{n}} + \frac{1}{\sqrt{T}} \right) \tag{45}$$

*with probability at least $1 - o(1)$, as long as $n = \tilde{\Omega}(d)$ and $T = \tilde{\Omega}(Var(\boldsymbol{\Sigma}_i) + Var(\boldsymbol{\Sigma}_i \mathbf{w}_i^*))$, where $\tilde{\mathcal{O}}$ and $\tilde{\Omega}$ exclude log factors.*

*Proof.* We first introduce notation to capture the dependency of the empirical training solution on $T$ and $n$. In particular, for any $T, n \geq 1$, we define

$$\mathbf{w}_{NAL}^{(T,n)} := (\frac{1}{Tn}\sum_{i=1}^{T}\mathbf{X}_i^\top \mathbf{X}_i)^{-1}\frac{1}{Tn}\sum_{i=1}^{T}\mathbf{X}_i^\top \mathbf{X}_i \mathbf{w}^* + \left( \frac{1}{Tn}\sum_{i=1}^{T}\mathbf{X}_i^\top \mathbf{X}_i \right)^{-1} \frac{1}{Tn}\sum_{i=1}^{T}\mathbf{X}_i^\top \mathbf{z}_i.$$

We next fix the number of tasks $T$ and define the asymptotic solution over $T$ tasks as the number of samples approaches infinity, i.e., $n \to \infty$, namely we define

$$\mathbf{w}_{NAL}^{(T)*} := (\sum_{i=1}^{T}\boldsymbol{\Sigma}_i)^{-1}\sum_{i=1}^{T}\boldsymbol{\Sigma}_i \mathbf{w}_i^*.$$

Using the triangle inequality we have

$$\|\mathbf{w}_{NAL}^{(T,n)} - \mathbf{w}^*_{NAL}\| = \|\mathbf{w}_{NAL}^{(T,n)} - \mathbf{w}_{NAL}^{(T)*} + \mathbf{w}_{NAL}^{(T)*} - \mathbf{w}^*_{NAL}\|$$
$$\leq \|\mathbf{w}_{NAL}^{(T,n)} - \mathbf{w}_{NAL}^{(T)*}\| + \|\mathbf{w}_{NAL}^{(T)*} - \mathbf{w}^*_{NAL}\| \tag{46}$$

We will first bound the first term in (46), which we denote as $\theta = \|\mathbf{w}_{NAL}^{(T,n)} - \mathbf{w}_{NAL}^{(T)*}\|$ for convenience. For this part we implicitly condition on the choice of $T$ training tasks to obtain a bound of the form $\mathbb{P}(\|\theta\| \geq \epsilon | \{\mathcal{T}_i\}_i) \leq 1 - \delta$. Since this holds for all $\{\mathcal{T}_i\}_i$, we will obtain the final result $\mathbb{P}(\|\theta\| \geq \epsilon) \leq 1 - \delta$ by the Law of Total Probability. We thus make the conditioning on $\{\mathcal{T}_i\}_i$ implicit for the rest of the analysis dealing with $\theta$.

We use the triangle inequality to separate $\theta$ into a term dependent on the data variance and a term dependent on the noise variance as follows:

$$
\begin{aligned}
\theta = {} & \left\| \sum_{i=1}^{T} \frac{1}{Tn} \left[ \left( \sum_{i'=1}^{T} \frac{1}{Tn} \mathbf{X}_{i'}^{\top} \mathbf{X}_{i'} \right)^{-1} \mathbf{X}_i^{\top} \mathbf{X}_i - \left( \frac{1}{T} \sum_{i'=1}^{T} \mathbf{\Sigma}_{i'} \right)^{-1} \frac{1}{T} \mathbf{\Sigma}_i \right] \mathbf{w}_i^* \right. \\
& \left. + \left( \frac{1}{Tn} \sum_{i=1}^{T} \mathbf{X}_i^{\top} \mathbf{X}_i \right)^{-1} \left( \frac{1}{Tn} \sum_{i=1}^{T} \mathbf{X}_i^{\top} \mathbf{z}_i \right) \right\| \\
\leq {} & \left\| \sum_{i=1}^{T} \frac{1}{Tn} \left[ \left( \sum_{i'=1}^{T} \frac{1}{Tn} \mathbf{X}_{i'}^{\top} \mathbf{X}_{i'} \right)^{-1} \mathbf{X}_i^{\top} \mathbf{X}_i - \left( \frac{1}{T} \sum_{i'=1}^{T} \mathbf{\Sigma}_{i'} \right)^{-1} \frac{1}{T} \mathbf{\Sigma}_i \right] \mathbf{w}_i^* \right\| \\
& + \left\| \left( \frac{1}{Tn} \sum_{i=1}^{T} \mathbf{X}_i^{\top} \mathbf{X}_i \right)^{-1} \left( \sum_{i=1}^{T} \frac{1}{Tn} \mathbf{X}_i^{\top} \mathbf{z}_i \right) \right\| 
\end{aligned}
\tag{47}
$$

where (47) follows from the triangle inequality. We analyze the two terms in (47) separately, starting with the first term, which we denote by $\vartheta$, namely

$$
\vartheta := \left\| \sum_{i=1}^{T} \frac{1}{Tn} \left[ \left( \sum_{i'=1}^{T} \frac{1}{Tn} \mathbf{X}_{i'}^{\top} \mathbf{X}_{i'} \right)^{-1} \mathbf{X}_i^{\top} \mathbf{X}_i - \left( \frac{1}{T} \sum_{i'=1}^{T} \mathbf{\Sigma}_{i'} \right)^{-1} \frac{1}{T} \mathbf{\Sigma}_i \right] \mathbf{w}_i^* \right\|
\tag{48}
$$

We define the matrix $\mathbf{A}_i$ for each $i$:

$$
\mathbf{A}_i = \frac{1}{Tn} \left( \sum_{i'=1}^{T} \frac{1}{Tn} \mathbf{X}_{i'}^{\top} \mathbf{X}_{i'} \right)^{-1} \mathbf{X}_i^{\top} \mathbf{X}_i - \left( \frac{1}{T} \sum_{i'=1}^{T} \mathbf{\Sigma}_{i'} \right)^{-1} \frac{1}{T} \mathbf{\Sigma}_i,
\tag{49}
$$

which implies that $\vartheta := \| \sum_{i=1}^{T} \mathbf{A}_i \mathbf{w}_i^* \|$. We proceed to bound the maximum singular value of $\mathbf{A}_i$ with high probability. Note that if we define $\mathbf{C}$ as

$$
\mathbf{C} := \frac{1}{Tn} \sum_{i'=1}^{T} \mathbf{X}_{i'}^{\top} \mathbf{X}_{i'}
$$

then $\mathbf{A}_i$ can be written as

$$
\mathbf{A}_i = \mathbf{C}^{-1} \left( \frac{1}{Tn} \mathbf{X}_i^{\top} \mathbf{X}_i - \left( \frac{1}{Tn} \sum_{i'=1}^{T} \mathbf{X}_{i'}^{\top} \mathbf{X}_{i'} \right) \left( \frac{1}{T} \sum_{i'=1}^{T} \mathbf{\Sigma}_{i'} \right)^{-1} \frac{1}{T} \mathbf{\Sigma}_i \right) = \mathbf{C}^{-1} \mathbf{B}_i
\tag{50}
$$

where, defining $\mathbf{\Lambda} := \frac{1}{T} \sum_{i''=1}^{T} \mathbf{\Sigma}_{i''}$ for notational convenience,

$$
\begin{aligned}
\mathbf{B}_i &= \left( \frac{1}{Tn} \mathbf{X}_i^{\top} \mathbf{X}_i - \left( \frac{1}{Tn} \sum_{i'=1}^{T} \mathbf{X}_{i'}^{\top} \mathbf{X}_{i',j} \right) \mathbf{\Lambda}^{-1} \frac{1}{T} \mathbf{\Sigma}_i \right) \\
&= \left( \frac{1}{Tn} \mathbf{X}_i^{\top} \mathbf{X}_i \mathbf{\Lambda}^{-1} \mathbf{\Lambda} - \left( \frac{1}{Tn} \sum_{i'=1}^{T} \mathbf{X}_{i'}^{\top} \mathbf{X}_{i'} \right) \mathbf{\Lambda}^{-1} \frac{1}{T} \mathbf{\Sigma}_i \right) \\
&= \left( \frac{1}{T^2 n} \sum_{i'=1}^{T} \mathbf{X}_i^{\top} \mathbf{X}_i \mathbf{\Lambda}^{-1} \mathbf{\Sigma}_{i'} - \left( \frac{1}{T^2 n} \sum_{i'=1}^{T} \mathbf{X}_{i'}^{\top} \mathbf{X}_{i'} \right) \mathbf{\Lambda}^{-1} \mathbf{\Sigma}_i \right) \\
&= \frac{1}{T^2 n} \sum_{i'=1}^{T} \left( \mathbf{X}_i^{\top} \mathbf{X}_i \mathbf{\Lambda}^{-1} \mathbf{\Sigma}_{i'} - \mathbf{X}_{i'}^{\top} \mathbf{X}_{i'} \mathbf{\Lambda}^{-1} \mathbf{\Sigma}_i \right)
\end{aligned}
\tag{51}
$$

Adding and subtracting terms, we have

$$
\begin{aligned}
\mathbf{B}_i &= \frac{1}{T^2 n} \sum_{i'=1}^{T} \left( \mathbf{X}_i^\top \mathbf{X}_i \mathbf{\Lambda}^{-1} \mathbf{\Sigma}_{i'} - n\mathbf{\Sigma}_i \mathbf{\Lambda}^{-1} \mathbf{\Sigma}_{i'} + n\mathbf{\Sigma}_{i'} \mathbf{\Lambda}^{-1} \mathbf{\Sigma}_i - \mathbf{X}_{i'}^\top \mathbf{X}_{i'} \mathbf{\Lambda}^{-1} \mathbf{\Sigma}_i \right) \\
&\quad + \frac{1}{T^2} \sum_{i'=1}^{T} \left( \mathbf{\Sigma}_i \mathbf{\Lambda}^{-1} \mathbf{\Sigma}_{i'} - \mathbf{\Sigma}_{i'} \mathbf{\Lambda}^{-1} \mathbf{\Sigma}_i \right) \\
&= \frac{1}{T} \left( \frac{1}{n} \mathbf{X}_i^\top \mathbf{X}_i - \mathbf{\Sigma}_i \right) \mathbf{\Lambda}^{-1} \left( \frac{1}{T} \sum_{i'=1}^{T} \mathbf{\Sigma}_{i'} \right) + \frac{1}{T^2} \sum_{i'=1}^{T} \left( \mathbf{\Sigma}_{i'} - \frac{1}{n} \mathbf{X}_{i'}^\top \mathbf{X}_{i'} \right) \mathbf{\Lambda}^{-1} \mathbf{\Sigma}_i \\
&\quad + \frac{1}{T} \mathbf{\Sigma}_i \mathbf{\Lambda}^{-1} \left( \frac{1}{T} \sum_{i'=1}^{T} \mathbf{\Sigma}_{i'} \right) - \frac{1}{T} \left( \frac{1}{T} \sum_{i'=1}^{T} \mathbf{\Sigma}_{i'} \right) \mathbf{\Lambda}^{-1} \mathbf{\Sigma}_i \\
&= \frac{1}{T} \left( \frac{1}{n} \mathbf{X}_i^\top \mathbf{X}_i - \mathbf{\Sigma}_i \right) + \frac{1}{T^2} \sum_{i'=1}^{T} \left( \mathbf{\Sigma}_{i'} - \frac{1}{n} \mathbf{X}_{i'}^\top \mathbf{X}_{i'} \right) \mathbf{\Lambda}^{-1} \mathbf{\Sigma}_i
\end{aligned}
$$

where the last line follows by the definition of $\mathbf{\Lambda}$.

Next, define

$$
\mathbf{Z}_i = \left( \mathbf{\Sigma}_i - \frac{1}{n} \mathbf{X}_i^\top \mathbf{X}_i \right) = \frac{1}{n} \sum_{h=1}^{n} \left( \mathbf{\Sigma}_i - \mathbf{x}_i^{(h)} (\mathbf{x}_i^{(h)})^\top \right)
$$

for all $i = 1, ..., T$, where $\mathbf{x}_i^{(h)} \in \mathbb{R}^d$ is the $h$-th sample for the $i$-th task (the $h$-th row of the matrix $\mathbf{X}_i$). Using this expression we can write

$$
\mathbf{B}_i = \frac{1}{T} \left( -\mathbf{Z}_i + \frac{1}{T} \sum_{i'=1}^{T} \mathbf{Z}_{i'} \mathbf{\Lambda}^{-1} \mathbf{\Sigma}_i \right)
$$

Note that each $\mathbf{Z}_i$ is the sum of $n$ independent random matrices with mean zero.

Using Lemma 27 from Tripuraneni et al. (2020) with each $a_i = 1$, we have that

$$
\mathbb{P}\left( \|\mathbf{Z}_i\| \le c_1 \lambda_{i,\max} \max\left( c_2(\sqrt{d/n} + t_i/n), c_2^2(\sqrt{d/n} + t_i/n)^2 \right) \right) \ge 1 - 2\exp(-t_i^2) \quad (52)
$$

for any $t_i > 0$, and for each $i \in [n]$.

Next, considering the expressions for $\vartheta$, $\mathbf{A}_i$, $\mathbf{B}_i$, $\mathbf{C}$, and $\mathbf{Z}_i$, we can write that

$$
\begin{aligned}
\vartheta &= \left\| \sum_{i=1}^{T} \mathbf{A}_i \mathbf{w}_i^* \right\| = \left\| \sum_{i=1}^{T} \mathbf{C}^{-1} \mathbf{B}_i \mathbf{w}_i^* \right\| = \left\| \mathbf{C}^{-1} \sum_{i=1}^{T} \mathbf{B}_i \mathbf{w}_i^* \right\| \\
&= \left\| \mathbf{C}^{-1} \sum_{i=1}^{T} \frac{1}{T} \left( -\mathbf{Z}_i + \frac{1}{T} \sum_{i'=1}^{T} \mathbf{Z}_{i'} \mathbf{\Lambda}^{-1} \mathbf{\Sigma}_i \right) \mathbf{w}_i^* \right\| \quad (53)
\end{aligned}
$$

Next replace $\mathbf{\Lambda}$ by its definition $\frac{1}{T} \sum_{i''=1}^{T} \mathbf{\Sigma}_{i''}$ to obtain

$$
\begin{aligned}
\vartheta &= \left\| \mathbf{C}^{-1} \frac{1}{T} \sum_{i=1}^{T} \left[ -\mathbf{Z}_i + \sum_{i'=1}^{T} \mathbf{Z}_{i'} \left( \sum_{i''=1}^{T} \mathbf{\Sigma}_{i''} \right)^{-1} \mathbf{\Sigma}_i \right] \mathbf{w}_i^* \right\| \\
&= \left\| \mathbf{C}^{-1} \frac{1}{T} \sum_{i'=1}^{T} \left[ \mathbf{Z}_{i'} \left( -\mathbf{w}_{i'}^* + \left( \sum_{i''=1}^{T} \mathbf{\Sigma}_{i''} \right)^{-1} \sum_{i=1}^{T} \mathbf{\Sigma}_i \mathbf{w}_i^* \right) \right] \right\| \quad (54)
\end{aligned}
$$

where in equation 54 we have swapped the summations. This implies

$$
\begin{aligned}
\vartheta &= \left\| \mathbf{C}^{-1} \frac{1}{T} \sum_{i'=1}^{T} \left[ \mathbf{Z}_{i'} \left( \sum_{i=1}^{T} \mathbf{\Sigma}_i \right)^{-1} \sum_{i=1}^{T} \mathbf{\Sigma}_i (\mathbf{w}_i^* - \mathbf{w}_{i'}^*) \right] \right\| \\
&\le \|\mathbf{C}^{-1}\| \frac{1}{T} \sum_{i'=1}^{T} \|\mathbf{Z}_{i'}\| \left\| \left( \sum_{i=1}^{T} \mathbf{\Sigma}_i \right)^{-1} \sum_{i=1}^{T} \mathbf{\Sigma}_i (\mathbf{w}_i^* - \mathbf{w}_{i'}^*) \right\| \quad (55)
\end{aligned}
$$

where (55) follows by the Cauchy-Schwarz and triangle inequalities. Using these inequalities and Assumptions 1 and 2 we can further bound $\vartheta$ as:

$$\vartheta \leq \|\mathbf{C}^{-1}\|\frac{1}{T}\sum_{i'=1}^{T}\|\mathbf{Z}_{i'}\|\left\|\left(\sum_{i=1}^{T}\mathbf{\Sigma}_i\right)^{-1}\right\|\sum_{i=1}^{T}\|\mathbf{\Sigma}_i(\mathbf{w}_i^* - \mathbf{w}_{i'}^*)\|$$

$$\leq \|\mathbf{C}^{-1}\|\frac{1}{T}\sum_{i'=1}^{T}\|\mathbf{Z}_{i'}\|\left\|\left(\sum_{i=1}^{T}\mathbf{\Sigma}_i\right)^{-1}\right\|\sum_{i=1}^{T}\|\mathbf{\Sigma}_i\|\|\mathbf{w}_i^* - \mathbf{w}_{i'}^*\|$$

$$\leq \|\mathbf{C}^{-1}\|\frac{1}{T}\sum_{i'=1}^{T}\|\mathbf{Z}_{i'}\|\left\|\left(\sum_{i=1}^{T}\mathbf{\Sigma}_i\right)^{-1}\right\|2TLB$$

Next, by the dual Weyl inequality for Hermitian matrices, we have $\lambda_{\min}\left(\sum_{i=1}^{T}\mathbf{\Sigma}_i\right) \geq \sum_{i=1}^{T}\lambda_{\min}(\mathbf{\Sigma}_i)$. Thus by Assumption 2, we have $\lambda_{\min}\left(\sum_{i=1}^{T}\mathbf{\Sigma}_i\right) \geq T\beta$, so

$$\vartheta \leq \|\mathbf{C}^{-1}\|\frac{1}{T}\sum_{i=1}^{T}\|\mathbf{Z}_i\|2LB/\beta = \|\mathbf{C}^{-1}\|\frac{2LB}{T\beta}\sum_{i=1}^{T}\|\mathbf{Z}_i\| \tag{56}$$

By a union bound and Lemma 27 from Tripuraneni et al. (2020) with each $a_i = 1$, the probability that any

$$\|\mathbf{Z}_i\| \geq c_1\lambda_{\max}(\mathbf{\Sigma}_i)\max\left(c_2(\sqrt{d/n} + t_i/n), c_2^2(\sqrt{d/n} + t_i/n)^2\right) \tag{57}$$

is at most $2\sum_{i=1}^{T}\exp(-t_i^2)$. Thus, with probability at least $1 - 2\sum_{i=1}^{T}\exp(-t_i^2)$

$$\vartheta \leq \|\mathbf{C}^{-1}\|\frac{2LB}{T\beta}\sum_{i=1}^{T}c_1\lambda_{\max}(\mathbf{\Sigma}_i)\max\left(c_2(\sqrt{d/n} + t_i/n), c_2^2(\sqrt{d/n} + t_i/n)^2\right)$$

Let $t := \max_i t_i$, then using Assumption 2 we have

$$\vartheta \leq \|\mathbf{C}^{-1}\|\frac{2L^2B}{\beta}c_1\max\left(c_2(\sqrt{d/n} + t/n), c_2^2(\sqrt{d/n} + t/n)^2\right) \tag{58}$$

with probability at least $1 - 2T\exp(-t^2)$ for some absolute constants $c_1$ and $c_2$ and any $t > 0$.

Next we bound $\|\mathbf{C}^{-1}\|$, where $\mathbf{C}$ is the random matrix

$$\mathbf{C} = \frac{1}{Tn}\sum_{i=1}^{T}\mathbf{X}_i^\top\mathbf{X}_i \tag{59}$$

Using the dual Weyl inequality again, we have

$$\lambda_{\min}(\mathbf{C}) \geq \frac{1}{T}\sum_{i=1}^{T}\lambda_{\min}\left(\frac{1}{n}\mathbf{X}_i^\top\mathbf{X}_i\right) \tag{60}$$

Next, using again using Lemma 27 from Tripuraneni et al. (2020) with each $a_i = 1$, as well as Weyl's Inequality (Theorem 4.5.3 in Vershynin (2018)), we have

$$\lambda_{\min}\left(\frac{1}{n}\mathbf{X}_i^\top\mathbf{X}_i\right) \geq \lambda_{\min}(\mathbf{\Sigma}_i) - c_1\|\mathbf{\Sigma}_i\|\max\left(c_2(\sqrt{d/n} + s/n), c_2^2(\sqrt{d/n} + t/n)^2\right)$$

$$\geq \beta - c_1L\max\left(c_2(\sqrt{d/n} + t/n), c_2^2(\sqrt{d/n} + t/n)^2\right)$$

$$= \beta - cL(\sqrt{d/n} + t/n) =: \phi \tag{61}$$

with probability at least $1 - 2\exp(-t^2)$ for any $t > 0$ and sufficiently large $n$ such that $\phi > 0$, where $c_1$, $c_2$, and $c$ are absolute constants (note that since $L \geq \beta$, in order for $\phi$ to be positive $n$ must be such that $c_2(\sqrt{d/n} + t/n) \leq 1$ assuming $c_1 \geq 1$, so we can eliminate the maximization. In

particular, we must have $n \geq \left( \frac{cL\sqrt{d} + \sqrt{c^2 L^2 d + 4\beta cLt}}{2\beta} \right)^2$. Now combining (61) with (60) and using a union bound over $i$, we have

$$
\begin{aligned}
\|\mathbf{C}^{-1}\| = \frac{1}{\lambda_{\min}(\mathbf{C})} &\leq \frac{T}{\sum_i^T \beta - cL(\sqrt{d/n} + s/n)} \\
&= \frac{1}{\beta - cL(\sqrt{d/n} + t/n)}
\end{aligned}
\tag{62}
$$

for any $t > 0$ and $n$ sufficiently large, where $c$ is an absolute constant, with probability at least $1 - 2T \exp\left(-t^2\right)$. Using (62) with (58), and noting that both inequalities are implied by the same event so no union bound is necessary, and $n \geq \left( \frac{cL\sqrt{d} + \sqrt{c^2 L^2 d + 4\beta cLt}}{2\beta} \right)^2$ sufficiently large, we have

$$
\vartheta \leq \frac{c'(\sqrt{d/n} + t/n)}{\beta - cL(\sqrt{d/n} + t/n)} \frac{L^2 B}{\beta}
\tag{63}
$$

with probability at least $1 - 2T \exp\left(-t^2\right)$ for some absolute constants $c$ and $c'$ and any $t > 0$.

So far, we derived an upper bound for the first term in (47) which we denoted by $\vartheta$. Next, we consider the second term of (47), which is due to the effect on the additive noise on the empirical solution. To be more precise, we proceed to provide an upper bound for $\left\| \left( \frac{1}{Tn} \sum_{i=1}^T \mathbf{X}_i^\top \mathbf{X}_i \right)^{-1} \left( \sum_{i=1}^T \frac{1}{Tn} \mathbf{X}_i^\top \mathbf{z}_i \right) \right\|$. Using the Cauchy-Schwarz inequality, we can bound this term as

$$
\begin{aligned}
&\left\| \left( \frac{1}{T} \sum_{i=1}^T \mathbf{X}_i^\top \mathbf{X}_i \right)^{-1} \left( \frac{1}{T} \sum_{i=1}^T \mathbf{X}_i^\top \mathbf{z}_i \right) \right\| \\
&\leq \left\| \left( \frac{1}{T} \sum_{i=1}^T \frac{1}{n} \mathbf{X}_i^\top \mathbf{X}_i \right)^{-1} \right\| \left\| \frac{1}{T} \sum_{i=1}^T \frac{1}{n} \mathbf{X}_i^\top \mathbf{z}_i \right\| \leq \|\mathbf{C}^{-1}\| \frac{1}{T} \sum_{i=1}^T \left\| \frac{1}{n} \mathbf{X}_i^\top \mathbf{z}_i \right\|
\end{aligned}
\tag{64}
$$

We have already bounded $\|\mathbf{C}^{-1}\|$, so we proceed to bound the term $\frac{1}{T} \sum_{i=1}^T \|\frac{1}{n} \mathbf{X}_i^\top \mathbf{z}_i\|$. Note that each element of the vector $\mathbf{X}_i^\top \mathbf{z}_i$ is the sum of products of a Gaussian random variable with another Gaussian random variable. Namely, denoting the $(h, s)$-th element of the matrix $\mathbf{X}_i$ as $x_i^{(h)}(s)$ and the $s$-th element of the vector $\mathbf{z}_i$ as $z_i(s)$, then the $s$-th element of $\mathbf{X}_i^\top \mathbf{z}_i$ is $\sum_{s=1}^d x_i^{(h)}(s) z_i(s)$. The products $x_i^{(h)}(s) z_i(s)$ are each sub-exponential, since the products of subgaussian random variables is sub-exponential (Lemma 2.7.7 in Vershynin (2018)), have mean zero, and are independent from each other. Thus by the Bernstein Inequality,

$$
\mathbb{P}\left( \left| \sum_{s=1}^d x_i^{(h)}(s) z_i(s) \right| \geq b \right) \leq 2 \exp\left( -c'' \min\left( \frac{b^2}{\sum_{s=1}^d K_s}, \frac{b}{\max_s K_s} \right) \right)
\tag{65}
$$

for some absolute constant $c''$ and any $b > 0$, where $K_s$ is the sub-exponential norm of the random variable $x_i^{(h)}(s) z_i(s)$ (for any $h$, since the above random variables indexed by $h$ are i.i.d.). Define $K := \max_s K_s$ Using a union bound over $h \in [n]$, we have

$$
\begin{aligned}
\mathbb{P}\left( \frac{1}{\sqrt{n}} \|\mathbf{X}_i^\top \mathbf{z}_i\| \geq b \right) = \mathbb{P}\left( \sum_{l=1}^n \left( \sum_{s=1}^d x_i^{(h)}(s) z_i(s) \right)^2 \geq n b^2 \right) \\
\leq \sum_{l=1}^n \mathbb{P}\left( \left| \sum_{s=1}^d x_i^{(h)}(s) z_i(s) \right| \geq b \right) \\
\leq 2n \exp\left( -c'' \min\left( \frac{b^2}{dK}, \frac{b}{K} \right) \right)
\end{aligned}
$$

for any $b > 0$. Thus we have $\frac{1}{n}\|\mathbf{X}_i^\top \mathbf{z}_i\| \leq b/\sqrt{n}$ with probability at least $1 - 2n\exp\left(-c''\min(\frac{b^2}{dK}, \frac{b}{K})\right)$. Combining this result with (64) with (47) and (63), we obtain

$$
\theta \leq \frac{c'(\sqrt{d/n}+t/n)}{\beta - cL(\sqrt{d/n}+t/n)}\frac{L^2 B}{\beta} + \frac{b}{\sqrt{n}}\frac{1}{\beta - cL(\sqrt{d/n}+t/n)}
$$

$$
= \frac{1}{\beta - cL(\sqrt{d/n}+t/n)}\left(\frac{c'\sqrt{d}+\beta b}{\beta\sqrt{n}} + \frac{tL^2 B}{\beta n}\right) \tag{66}
$$

with probability at least

$$
1 - 2T\exp\left(-t^2\right) - 2n\exp\left(-c''\min\left(\frac{b^2}{dK}, \frac{b}{K}\right)\right) \tag{67}
$$

for any $t, b > 0$ and $n \geq \left(\frac{cL\sqrt{d}+\sqrt{c^2 L^2 d + 4\beta cLt}}{2\beta}\right)^2$.

Now that we have bounds for the terms in (47), we proceed to bound the second term in (46). We have

$$
\|\mathbf{w}_{NAL}^{(T)*} - \mathbf{w}_{NAL}^*\|
$$

$$
= \left\|\left(\frac{1}{T}\sum_{i=1}^{T}\boldsymbol{\Sigma}_i\right)^{-1}\frac{1}{T}\sum_{i=1}^{T}\boldsymbol{\Sigma}_i\mathbf{w}_i^* - \mathbb{E}_i[\boldsymbol{\Sigma}_i]^{-1}\mathbb{E}_i[\boldsymbol{\Sigma}_i\mathbf{w}_i^*]\right\|
$$

$$
= \left\|\left(\frac{1}{T}\sum_{i=1}^{T}\boldsymbol{\Sigma}_i\right)^{-1}\frac{1}{T}\sum_{i=1}^{T}\boldsymbol{\Sigma}_i\mathbf{w}_i^* - \left(\frac{1}{T}\sum_{i=1}^{T}\boldsymbol{\Sigma}_i\right)^{-1}\mathbb{E}_i[\boldsymbol{\Sigma}_i\mathbf{w}_i^*]\right.
$$

$$
\left. + \left(\frac{1}{T}\sum_{i=1}^{T}\boldsymbol{\Sigma}_i\right)^{-1}\mathbb{E}_i[\boldsymbol{\Sigma}_i\mathbf{w}_i^*] - \mathbb{E}_i[\boldsymbol{\Sigma}_i]^{-1}\mathbb{E}_i[\boldsymbol{\Sigma}_i\mathbf{w}_i^*]\right\|
$$

$$
\leq \left\|\left(\frac{1}{T}\sum_{i=1}^{T}\boldsymbol{\Sigma}_i\right)^{-1}\frac{1}{T}\sum_{i=1}^{T}\boldsymbol{\Sigma}_i\mathbf{w}_i^* - \left(\sum_{i=1}^{T}\boldsymbol{\Sigma}_i\right)^{-1}\mathbb{E}_i[\boldsymbol{\Sigma}_i\mathbf{w}_i^*]\right\|
$$

$$
+ \left\|\left(\frac{1}{T}\sum_{i=1}^{T}\boldsymbol{\Sigma}_i\right)^{-1}\mathbb{E}_i[\boldsymbol{\Sigma}_i\mathbf{w}_i^*] - \mathbb{E}_i[\boldsymbol{\Sigma}_i]^{-1}\mathbb{E}_i[\boldsymbol{\Sigma}_i\mathbf{w}_i^*]\right\| \tag{68}
$$

$$
\leq \left\|\left(\frac{1}{T}\sum_{i=1}^{T}\boldsymbol{\Sigma}_i\right)^{-1}\right\|\left\|\frac{1}{T}\sum_{i=1}^{T}\boldsymbol{\Sigma}_i\mathbf{w}_i^* - \mathbb{E}_i[\boldsymbol{\Sigma}_i\mathbf{w}_i^*]\right\|
$$

$$
+ \left\|\left(\frac{1}{T}\sum_{i=1}^{T}\boldsymbol{\Sigma}_i\right)^{-1} - \mathbb{E}_i[\boldsymbol{\Sigma}_i]^{-1}\right\|\|\mathbb{E}_i[\boldsymbol{\Sigma}_i\mathbf{w}_i^*]\| \tag{69}
$$

$$
= \frac{1}{\beta}\left\|\frac{1}{T}\sum_{i=1}^{T}\boldsymbol{\Sigma}_i\mathbf{w}_i^* - \mathbb{E}_i[\boldsymbol{\Sigma}_i\mathbf{w}_i^*]\right\| + \left\|\left(\frac{1}{T}\sum_{i=1}^{T}\boldsymbol{\Sigma}_i\right)^{-1} - \mathbb{E}_i[\boldsymbol{\Sigma}_i]^{-1}\right\|LB \tag{70}
$$

where in equations (68) and (69) we have used the triangle and Cauchy-Schwarz inequalities, respectively, and in (70) we have used the dual Weyl's inequality and Assumption 2. We first consider the second term in (70). We can bound this term as

$$
\left\|\left(\frac{1}{T}\sum_{i=1}^{T}\boldsymbol{\Sigma}_i\right)^{-1} - \mathbb{E}_i[\boldsymbol{\Sigma}_i]^{-1}\right\| = \left\|\left(\frac{1}{T}\sum_{i=1}^{T}\boldsymbol{\Sigma}_i\right)^{-1}\left(\frac{1}{T}\sum_{i=1}^{T}(\boldsymbol{\Sigma}_i - \mathbb{E}_{i'}[\boldsymbol{\Sigma}_{i'}])\right)\mathbb{E}_i[\boldsymbol{\Sigma}_i]^{-1}\right\|
$$

$$
\leq \left\|\left(\frac{1}{T}\sum_{i=1}^{T}\boldsymbol{\Sigma}_i\right)^{-1}\right\|\left\|\frac{1}{T}\sum_{i=1}^{T}\left(\boldsymbol{\Sigma}_i - \mathbb{E}_{i'}[\boldsymbol{\Sigma}_{i'}]\right)\right\|\left\|\mathbb{E}_i[\boldsymbol{\Sigma}_i]^{-1}\right\|
$$

$$
\leq \frac{1}{\beta^2}\left\|\frac{1}{T}\sum_{i=1}^{T}(\boldsymbol{\Sigma}_i - \mathbb{E}_{i'}[\boldsymbol{\Sigma}_{i'}])\right\| \tag{71}
$$

using Assumptions 1 and 2.

We use the matrix Bernstein inequality (Theorem 6.5 in Tropp (2015)) to bound $\|\frac{1}{T}\sum_{i=1}^{T}(\mathbf{\Sigma}_i - \mathbb{E}_{i'}[\mathbf{\Sigma}_{i'}])\|$, noting that each $\mathbf{\Sigma}_i - \mathbb{E}_{i'}[\mathbf{\Sigma}_{i'}]$ is an iid matrix with mean zero. Recall that $\mathrm{Var}(\mathbf{\Sigma}_i) := \|\mathbb{E}_i[(\mathbf{\Sigma}_i - \mathbb{E}_{i'}[\mathbf{\Sigma}_{i'}])^2]\|$. By matrix Bernstein and equation (71) we obtain

$$\left\|\left(\frac{1}{T}\sum_{i=1}^{T}\mathbf{\Sigma}_i\right)^{-1} - \mathbb{E}_i[\mathbf{\Sigma}_i]^{-1}\right\| \leq \frac{1}{\beta^2}\frac{\delta\ \mathrm{Var}(\mathbf{\Sigma}_i)}{\sqrt{T}} \tag{72}$$

with probability at least $1 - 2d\exp\left(-\frac{\delta^2\ \mathrm{Var}(\mathbf{\Sigma}_i)/2}{1+L\delta/(3\sqrt{T})}\right)$, for any $\delta > 0$. Similarly, we have that

$$\frac{1}{\beta}\left\|\frac{1}{T}\sum_{i=1}^{T}\mathbf{\Sigma}_i\mathbf{w}_i^* - \mathbb{E}_i[\mathbf{\Sigma}_i\mathbf{w}_i^*]\right\| \leq \frac{1}{\beta}\frac{\delta\ \mathrm{Var}(\mathbf{\Sigma}_i\mathbf{w}_{i,*})}{\sqrt{T}} \tag{73}$$

with probability at least $1 - 2d\exp\left(-\frac{\delta^2\ \mathrm{Var}(\mathbf{\Sigma}_i\mathbf{w}_i^*)/2}{1+LB\delta/(3\sqrt{T})}\right)$.

Thus (70) reduces to:

$$\|\mathbf{w}_{NAL}^{(T)*} - \mathbf{w}_{NAL}^*\| \leq \frac{1}{\beta}\frac{\delta\ \mathrm{Var}(\mathbf{\Sigma}_i\mathbf{w}_{i,*})}{\sqrt{T}} + \frac{LB}{\beta^2}\frac{\delta\ \mathrm{Var}(\mathbf{\Sigma}_i)}{\sqrt{T}} \tag{74}$$

with probability at least $1 - 2d\exp\left(-\frac{\delta^2\ \mathrm{Var}(\mathbf{\Sigma}_i)/2}{1+L\delta/(3\sqrt{T})}\right) - 2d\exp\left(-\frac{\delta^2\ \mathrm{Var}(\mathbf{\Sigma}_i\mathbf{w}_i^*)/2}{1+LB\delta/(3\sqrt{T})}\right)$.

We combine this result with (47) and (66) via a union bound to obtain

$$\|\mathbf{w}_{NAL} - \mathbf{w}_{NAL}^*\| \leq \frac{1}{\beta - cL(\sqrt{d/n}+t/n)}\left(\frac{c'\sqrt{d}+\beta b}{\beta\sqrt{n}} + \frac{tL^2B}{\beta n}\right)$$
$$+ \frac{\delta\ \mathrm{Var}(\mathbf{\Sigma}_i\mathbf{w}_{i,*})}{\beta\sqrt{T}} + \frac{\delta\ \mathrm{Var}(LB\mathbf{\Sigma}_i)}{\beta^2\sqrt{T}} \tag{75}$$

with probability at least

$$1 - 2T\exp(-t^2) - 2n\exp\left(-c''\min\left(\frac{b^2}{dK},\frac{b}{K}\right)\right) - 2d\exp\left(-\frac{\delta^2\ \mathrm{Var}(\mathbf{\Sigma}_i)/2}{1+L\delta/(3\sqrt{T})}\right)$$
$$- 2d\exp\left(-\frac{\delta^2\ \mathrm{Var}(\mathbf{\Sigma}_i\mathbf{w}_i^*)/2}{1+LB\delta/(3\sqrt{T})}\right) \tag{76}$$

as long as $n \geq \left(\frac{cL\sqrt{d}+\sqrt{c^2L^2d+4\beta cLt}}{2\beta}\right)^2$. Finally, choose $t = \sqrt{\log(200T)}$, $b = \sqrt{\frac{dK}{c''}\log(200n)} + \frac{K}{c''}\log(200n)$ and $\delta = (\sqrt{\mathrm{Var}(\mathbf{\Sigma}_i) + \mathrm{Var}(\mathbf{\Sigma}_i\mathbf{w}_i^*)})^{-1}\log(200d)$ and restrict $n \geq 4\left(\frac{cL\sqrt{d}+\sqrt{c^2L^2d+4\beta cLt}}{2\beta}\right)^2$ such that $\frac{1}{\beta - cL(\sqrt{d/n}+t/n)} \leq \frac{2}{\beta}$ and $T > L^2B^2\log(200d)/(9(\mathrm{Var}(\mathbf{\Sigma}_i) + \mathrm{Var}(\mathbf{\Sigma}_i\mathbf{w}_i^*)))$. This ensures that each negative term in the high probability bound (124) is at most 0.01 and thereby completes the proof. $\qquad\square$

## D.2 MAML CONVERGENCE

We first state and prove the following lemma.

**Lemma 2.** *Let $\mathbf{A}$ be a fixed symmetric matrix in $\mathbb{R}^{d \times d}$, and let $\mathbf{X}$ be a random matrix in $\mathbb{R}^{n \times d}$ whose rows are i.i.d. multivariate Gaussian random vectors with mean $\mathbf{0}$ and diagonal covariance $\mathbf{\Sigma}$. Then*

$$\mathbf{E_X}\left[\left(\frac{1}{n}\mathbf{X}^\top\mathbf{X}\right)\mathbf{A}\left(\frac{1}{n}\mathbf{X}^\top\mathbf{X}\right)\right] = \mathbf{\Sigma}\mathbf{A}\mathbf{\Sigma} + \frac{1}{n}\left(tr(\mathbf{\Sigma}\mathbf{A})\mathbf{I}_d + \mathbf{\Sigma}\mathbf{A}\right)\mathbf{\Sigma} \tag{77}$$

*Proof.* Letting $\mathbf{x}_k$ denote the $k$-th row of $\mathbf{X}$, we have

$$\begin{aligned}
\mathbf{E_X}&\left[\left(\frac{1}{n}\mathbf{X}^\top\mathbf{X}\right)\mathbf{A}\left(\frac{1}{n}\mathbf{X}^\top\mathbf{X}\right)\right] \\
&= \mathbf{E_X}\left[\left(\frac{1}{n}\sum_{k=1}^n\mathbf{x}_k^\top\mathbf{x}_k\right)\mathbf{A}\left(\frac{1}{n}\sum_{k=1}^n\mathbf{x}_k^\top\mathbf{x}_k\right)\right] \\
&= \mathbf{E}_{\mathbf{x}_k}\left[\frac{1}{n^2}\sum_{k=1}^n\sum_{k'=1,k'\neq k}^n\mathbf{x}_k^\top\mathbf{x}_k\mathbf{A}\mathbf{x}_{k'}^\top\mathbf{x}_{k'}\right] + \mathbf{E}_{\mathbf{x}_k}\left[\frac{1}{n^2}\sum_{k=1}^n\mathbf{x}_k^\top\mathbf{x}_k\mathbf{A}\mathbf{x}_k^\top\mathbf{x}_k\right] \\
&= \frac{n-1}{n}\mathbf{\Sigma}\mathbf{A}\mathbf{\Sigma} + \frac{1}{n}\sum_{k=1}^n\mathbf{E}_{\mathbf{x}_k}\left[\mathbf{x}_k^\top\mathbf{x}_k\mathbf{A}\mathbf{x}_k^\top\mathbf{x}_k\right]
\end{aligned} \tag{78}$$

Let $\mathbf{C}_k = \mathbf{x}_k^\top\mathbf{x}_k\mathbf{A}\mathbf{x}_k^\top\mathbf{x}_k$ for $k \in [n]$, and let $\lambda_i$ be the $i$-th diagonal element of $\mathbf{\Sigma}$ for $i \in [d]$. Then for any $k$, using the fact that the elements of $\mathbf{x}_k$ are independent, have all odd moments equal to 0, and have fourth moment equal to $3\lambda_i^2$ for the corresponding $i$, it follows that

$$\mathbb{E}[C_{r,s}] = \begin{cases} \lambda_r\sum_{j=1}^d\lambda_j a_{j,j} + 2\lambda_r^2 A_{r,r}, & \text{if } r = s \\ \lambda_r\lambda_s(A_{r,s} + A_{s,r}), & \text{otherwise} \end{cases} \tag{79}$$

Using (79), we can write

$$\begin{aligned}
\mathbb{E}[\mathbf{C}_k] &= \text{tr}(\mathbf{\Sigma}\mathbf{A})\mathbf{\Sigma} + \mathbf{\Sigma}(\mathbf{A} + \mathbf{A}^\top)\mathbf{\Sigma} \\
&= \text{tr}(\mathbf{\Sigma}\mathbf{A})\mathbf{\Sigma} + 2\mathbf{\Sigma}\mathbf{A}\mathbf{\Sigma}
\end{aligned} \tag{80}$$

where (80) follows by the symmetry of $\mathbf{A}$. Plugging (80) into (78) completes the proof. $\square$

Now we have the main convergence result. Analogously to Theorem 2, we define $\text{Var}(\mathbf{Q}_i) := \|\mathbb{E}_i[(\mathbf{Q}_i - \mathbb{E}_{i'}[\mathbf{Q}_{i'}])^2]\|_2$ and $\text{Var}(\mathbf{Q}_i\mathbf{w}_{i,*}) := \|\mathbb{E}_i[(\mathbf{Q}_i\mathbf{w}_{i,*} - \mathbb{E}_{i'}[\mathbf{Q}_{i'}\mathbf{w}_{i,*}])^2]\|_2$.

**Theorem 3.** *(MAML Convergence, General Statement) Define $\hat{\beta} := \beta(1-\alpha L)^2 + \frac{\alpha^2\beta^3(d+1)}{n_2}$ and $\hat{L} := L(1-\alpha\beta)^2 + \frac{\alpha^2 L^3(d+1)}{n_2}$. If Assumptions 1 and 2 hold, the distance of the MAML training solution (4) to its population-optimal value (5) is bounded as*

$$\begin{aligned}
\|\mathbf{w}_{MAML} - \mathbf{w}_{MAML}^*\| &\leq \frac{16\hat{L}Bd\sqrt{\frac{\alpha^4 L^6}{c}}\log^3(100Td)}{\hat{\beta}^2\sqrt{\tau}} \\
&\quad + \frac{\sqrt{Var(\mathbf{Q}_i\mathbf{w}_i^*)}\log(200d)}{\hat{\beta}\sqrt{T}} + \frac{\hat{L}B\sqrt{Var(\mathbf{Q}_i)}\log(200d)}{\hat{\beta}^2\sqrt{T}}
\end{aligned} \tag{81}$$

*with probability at least 0.96, for some absolute constants $c$ and $c'$, and any $\tau > 32d^3\alpha^4 L^6\log^6(100Td)/(c\hat{\beta})$ and $T > \hat{L}^2 B^2\log(200d)/(9(Var(\mathbf{Q}_i) + Var(\mathbf{Q}_i\mathbf{w}_i^*)))$.*

*Proof.* The proof follows the same form as the proof of Theorem 2. Here the empirical covariance matrices $\frac{1}{n}\mathbf{X}_i^\top\mathbf{X}_i$ and their means $\mathbf{\Sigma}_i$ are replaced by the empirical *preconditioned* covariance matrices $\hat{\mathbf{Q}}_{i,j}$ and their means $\mathbf{Q}_i$, respectively. As before, a critical aspect of the proof will be to

show concentration of the empirical (preconditioned) covariance matrix to its mean. To do so, we re-define the perturbation matrices $\mathbf{Z}_i$:

$$\mathbf{Z}_i := \frac{1}{\tau n_1} \sum_{j=1}^{\tau} \hat{\mathbf{Q}}_{i,j} - \mathbf{Q}_i \tag{82}$$

where

$$\hat{\mathbf{Q}}_{i,j} := \mathbf{P}_{i,j}^{\top} (\mathbf{X}_{i,j}^{out})^{\top} \mathbf{X}_{i,j}^{out} \mathbf{P}_{i,j}$$

and

$$
\begin{aligned}
\mathbf{Q}_i &:= \mathbb{E}\left[ \frac{1}{\tau n_1} \sum_{j=1}^{\tau} \hat{\mathbf{Q}}_{i,j} \right] \\
&= \left( \frac{1}{\tau n_1} \sum_{j=1}^{\tau} \mathbb{E}[\mathbf{P}_{i,j}^{\top} (\mathbf{X}_{i,j}^{out})^{\top} \mathbf{X}_{i,j}^{out} \mathbf{P}_{i,j}] \right) \\
&= \frac{1}{\tau n_1} \sum_{j=1}^{\tau} \mathbb{E}\left[ \left( \mathbf{I}_d - \frac{\alpha}{n_2}(\mathbf{X}_{i,j}^{in})^{\top} \mathbf{X}_{i,j}^{in} \right) \mathbf{\Sigma}_i \left( \mathbf{I}_d - \frac{\alpha}{n_2}(\mathbf{X}_{i,j}^{in})^{\top} \mathbf{X}_{i,j}^{in} \right) \right] \\
&= \frac{1}{\tau n_1} \sum_{j=1}^{\tau} \left( \mathbf{\Sigma}_i - 2\alpha \mathbf{\Sigma}_i^2 + \frac{\alpha^2}{n_2^2} \mathbb{E}\left[ (\mathbf{X}_{i,j}^{in})^{\top} \mathbf{X}_{i,j}^{in} (\mathbf{X}_{i,j}^{out})^{\top} \mathbf{X}_{i,j}^{out} (\mathbf{X}_{i,j}^{in})^{\top} \mathbf{X}_{i,j}^{in} \right] \right)
\end{aligned} \tag{83}
$$

$$= (\mathbf{I}_d - \alpha \mathbf{\Sigma}_i) \mathbf{\Sigma}_i (\mathbf{I}_d - \alpha \mathbf{\Sigma}_i) + \frac{\alpha^2}{n_2} (\mathrm{tr}(\mathbf{\Sigma}_i^2) \mathbf{\Sigma}_i + \mathbf{\Sigma}_i^3) \tag{84}$$

where (84) follows by Lemma 2.

Note that here $\mathbf{Z}_i$ has higher-order matrices than previously. Lemma 4 nevertheless gives the key concentration result for the $\mathbf{Z}_i$'s, which we will use later. For now, we argue as in Theorem 2: for any $T, \tau \geq 1$, we define

$$
\begin{aligned}
&\mathbf{w}_{MAML}^{(T,\tau)} \\
&:= \left( \frac{1}{T\tau n_1} \sum_{i=1}^{T} \sum_{j=1}^{\tau} \hat{\mathbf{Q}}_{i,j} \right)^{-1} \frac{1}{T\tau n_1} \sum_{i=1}^{T} \sum_{j=1}^{\tau} (\hat{\mathbf{Q}}_{i,j} \mathbf{w}_i^* + \mathbf{P}_{i,j} \mathbf{X}_{i,j}^{\top} \mathbf{z}_{i,j}^{out} - \frac{\alpha}{n_2} \mathbf{P}_{i,j} \mathbf{X}_{i,j}^{\top} \mathbf{X}_{i,j} \hat{\mathbf{X}}_{i,j}^{\top} \hat{\mathbf{z}}_{i,j}^{in}).
\end{aligned}
$$

We next fix $T$ and define the asymptotic solution over $T$ tasks as $\tau \to \infty$, namely $\mathbf{w}_{MAML}^{(T)*} := (\frac{1}{T} \sum_{i=1}^{T} \mathbf{Q}_i)^{-1} \frac{1}{T} \sum_{i=1}^{T} \mathbf{Q}_i \mathbf{w}_i^*$. Again using the triangle inequality we have

$$
\begin{aligned}
\|\mathbf{w}_{MAML}^{(T,\tau)} - \mathbf{w}_{MAML}^*\| &= \|\mathbf{w}_{MAML}^{(T,\tau)} - \mathbf{w}_{MAML}^{(T)*} + \mathbf{w}_{MAML}^{(T)*} - \mathbf{w}_{MAML}^*\| \\
&\leq \|\mathbf{w}_{MAML}^{(T,\tau)} - \mathbf{w}_{MAML}^{(T)*}\| + \|\mathbf{w}_{MAML}^{(T)*} - \mathbf{w}_{MAML}^*\|
\end{aligned} \tag{85}
$$

The first term in (85) captures the error due to have limited samples per task during training, and the second term captures the error from having limited tasks from the environment during training. We first bound the first term in (85), which we denote as $\theta = \|\mathbf{w}_{MAML}^{(T,\tau)} - \mathbf{w}_{MAML}^{(T)*}\|$ for convenience. Note that by Assumption 2, we have

$$\hat{\beta} \mathbf{I}_d \preceq \mathbf{Q}_i \preceq \hat{L} \mathbf{I}_d \qquad \forall i \tag{86}$$

where $\hat{\beta} := \beta(1 - \alpha L)^2 + \frac{\alpha^2 \beta^3 (d+1)}{n_2}$ and $\hat{L} := L(1 - \alpha \beta)^2 + \frac{\alpha^2 L^3 (d+1)}{n_2}$. Thus, using the argument from (47) to (56) in the proof of Theorem 2, with $\frac{1}{n} \mathbf{X}_{i,j}^{\top} \mathbf{X}_{i,j}$ replaced by $\frac{1}{\tau n_1} \sum_{j=1}^{\tau} \hat{\mathbf{Q}}_{i,j}$ and $\mathbf{\Sigma}_i$ replaced by $\mathbf{Q}_i$, we obtain

$$
\begin{aligned}
\theta \leq \vartheta + &\left\| \left( \frac{1}{T\tau n_1} \sum_{i=1}^{T} \sum_{j=1}^{\tau} \hat{\mathbf{Q}}_{i,j} \right)^{-1} \right. \\
&\left. \frac{1}{T\tau n_1} \sum_{i=1}^{T} \sum_{j=1}^{\tau} \left( \mathbf{P}_{i,j} (\mathbf{X}_{i,j}^{out})^{\top} \mathbf{z}_{i,j}^{out} - \frac{\alpha}{n_2} \mathbf{P}_{i,j} (\mathbf{X}_{i,j}^{out})^{\top} \mathbf{X}_{i,j}^{out} (\mathbf{X}_{i,j}^{in})^{\top} \mathbf{z}_{i,j}^{in} \right) \right\|
\end{aligned} \tag{87}
$$

where

$$\vartheta := \left\| \sum_{i=1}^{T} \left[ \left( \frac{1}{T} \sum_{i'=1}^{T} \frac{1}{n_1 \tau} \sum_{j=1}^{\tau} \hat{\mathbf{Q}}_{i',j} \right)^{-1} \frac{1}{T} \hat{\mathbf{Q}}_{i,j} - \left( \frac{1}{T} \sum_{i'=1}^{T} \mathbf{Q}_{i'} \right)^{-1} \frac{1}{T} \mathbf{Q}_i \right] \mathbf{w}_i^* \right\|.$$

Defining $\mathbf{C} := \frac{1}{T} \sum_{i'=1}^{T} \frac{1}{n_1 \tau} \sum_{j=1}^{\tau} \hat{\mathbf{Q}}_{i',j}$, we have,

$$\vartheta = \left\| \sum_{i=1}^{T} \left[ \frac{1}{T} \mathbf{C}^{-1} \hat{\mathbf{Q}}_{i,j} - \left( \frac{1}{T} \sum_{i'=1}^{T} \mathbf{Q}_{i'} \right)^{-1} \frac{1}{T} \mathbf{Q}_i \right] \mathbf{w}_i^* \right\| \leq \|\mathbf{C}^{-1}\| \frac{2\hat{L}B}{T\hat{\beta}} \sum_{i=1}^{T} \|\mathbf{Z}_i\| \tag{88}$$

where to obtain the inequality we have swapped the order of the summations as in (54). Next we bound each $\|\mathbf{Z}_i\|$ with high probability, by first showing element-wise convergence to 0.

**Lemma 3.** *Consider a fixed $i \in [T]$, $k \in [d]$, and $s \in [d]$, and let $\hat{Q}_{i,j}(k,s)$ be the $(k,s)$-th element of the matrix $\hat{\mathbf{Q}}_{i,j}$ defined in (93). The following concentration result holds for the random variable $\frac{1}{\tau n_1} \sum_{j=1}^{\tau} \hat{Q}_{i,j}(k,s)\hat{Q}_{i,j}(k,s)$:*

$$\mathbb{P}\left( \left| \frac{1}{\tau n_1} \sum_{j=1}^{\tau} \hat{Q}_{i,j}(k,s) - Q_i(k,s) \right| \leq \gamma \right) \geq e^2 \exp\left( -\left( \frac{\tau \gamma^2}{c\alpha^4 d^2 L^6} \right)^{1/6} \right) \tag{89}$$

*for some any $\gamma > 0$, where $\mathbf{Q}_i = \mathbb{E}[\frac{1}{\tau n_1} \sum_{j=1}^{\tau} \hat{\mathbf{Q}}_{i,j}]$ as defined in e.g. (83).*

*Proof.* We start by computing the $(k,l)$-th element of $\mathbf{Z}_i$. First we compute the $(k,l)$-th element of the matrix $\mathbf{D}_{i,j} := (\mathbf{I}_d - \frac{\alpha}{n_2}(\mathbf{X}_{i,j}^{in})^\top \mathbf{X}_{i,j}^{in})(\mathbf{X}_{i,j}^{out})^\top \mathbf{X}_{i,j}^{out}$. Here, we define $\mathbf{x}_{i,j}^{(r)} \in \mathbb{R}^{n_1}$ as the $n_1$-dimensional vector of the $r$-th-dimensional elements from all of the $n_1$ outer samples for the $j$-th instance of task $i$, and $\hat{\mathbf{x}}_{i,j}^{(k)} \in \mathbb{R}^{n_2}$ as the $n_2$-dimensional vector of the $k$-th-dimensional elements from all of the $n_2$ outer samples for task $i$. Then we have

$$D_{i,j}(k,l) = \sum_{r=1}^{d} (\mathbb{1}_{r=k} - \frac{\alpha}{n_2} \langle \hat{\mathbf{x}}_{i,j}^{(k)}, \hat{\mathbf{x}}_{i,j}^{(r)} \rangle) \langle \mathbf{x}_{i,j}^{(r)}, \mathbf{x}_{i,j}^{(l)} \rangle$$

$$= \langle \mathbf{x}_{i,j}^{(k)}, \mathbf{x}_{i,j}^{(l)} \rangle - \sum_{r=1}^{d} \frac{\alpha}{n_2} \langle \hat{\mathbf{x}}_{i,j}^{(k)}, \hat{\mathbf{x}}_{i,j}^{(r)} \rangle \langle \mathbf{x}_{i,j}^{(r)}, \mathbf{x}_{i,j}^{(l)} \rangle \tag{90}$$

Then, the $(k,s)$-th element of $\hat{\mathbf{Q}}_{i,j}$ is

$$\hat{\mathbf{Q}}_{i,j}(k,s)$$

$$= \left[ \mathbf{P}_{i,j}^\top \mathbf{X}_{i,j}^\top \mathbf{X}_{i,j} \mathbf{P}_{i,j} \right](k,s)$$

$$= \sum_{l=1}^{d} D_{i,j}(h,l)(\mathbb{1}_{l=s} - \frac{\alpha}{n_2} \langle \hat{\mathbf{x}}_{i,j}^{(l)}, \hat{\mathbf{x}}_{i,j}^{(s)} \rangle)$$

$$= \sum_{l=1}^{d} (\mathbb{1}_{l=s} - \frac{\alpha}{n_2} \langle \hat{\mathbf{x}}_{i,j}^{(l)}, \hat{\mathbf{x}}_i^{(s)} \rangle) \left( \langle \mathbf{x}_{i,j}^{(k)}, \mathbf{x}_i^{(l)} \rangle - \sum_{r=1}^{d} \frac{\alpha}{n_2} \langle \hat{\mathbf{x}}_{i,j}^{(k)}, \hat{\mathbf{x}}_{i,j}^{(r)} \rangle \langle x_{i,j}^{(r)}, \mathbf{x}_{i,j}^{(l)} \rangle \right)$$

$$= \left( \langle \mathbf{x}_{i,j}^{(k)}, \mathbf{x}_{i,j}^{(s)} \rangle - \sum_{r=1}^{d} \frac{\alpha}{n_2} \langle \hat{\mathbf{x}}_{i,j}^{(k)}, \hat{\mathbf{x}}_i^{(r)} \rangle \langle \mathbf{x}_{i,j}^{(r)}, \mathbf{x}_{i,j}^{(s)} \rangle \right)$$

$$\quad - \frac{\alpha}{n_2} \sum_{l=1}^{d} \langle \hat{\mathbf{x}}_{i,j}^{(l)}, \hat{\mathbf{x}}_{i,j}^{(s)} \rangle \left( \langle \mathbf{x}_{i,j}^{(k)}, \mathbf{x}_{i,j}^{(l)} \rangle - \sum_{r=1}^{d} \frac{\alpha}{n_2} \langle \hat{\mathbf{x}}_{i,j}^{(k)}, \hat{\mathbf{x}}_{i,j}^{(r)} \rangle \langle \mathbf{x}_{i,j}^{(r)}, \mathbf{x}_{i,j}^{(l)} \rangle \right)$$

$$= \langle \mathbf{x}_{i,j}^{(k)}, \mathbf{x}_{i,j}^{(s)} \rangle - \frac{2\alpha}{n_2} \sum_{r=1}^{d} \langle \hat{\mathbf{x}}_{i,j}^{(k)}, \hat{\mathbf{x}}_{i,j}^{(r)} \rangle \langle \mathbf{x}_{i,j}^{(r)}, \mathbf{x}_{i,j}^{(s)} \rangle + \frac{\alpha^2}{n_2^2} \sum_{l=1}^{d} \sum_{r=1}^{d} \langle \hat{\mathbf{x}}_{i,j}^{(l)}, \hat{\mathbf{x}}_{i,j}^{(s)} \rangle \langle \hat{\mathbf{x}}_{i,j}^{(k)}, \hat{\mathbf{x}}_i^{(r)} \rangle \langle \mathbf{x}_{i,j}^{(r)}, \mathbf{x}_{i,j}^{(l)} \rangle$$

$$\tag{91}$$

Therefore the $(k, s)$ element of $\frac{1}{\tau n_1} \sum_{j=1}^{\tau} \hat{\mathbf{Q}}_{i,j}$ is

$$\left[ \frac{1}{\tau n_1} \sum_{j=1}^{\tau} \hat{\mathbf{Q}}_{i,j} \right](k, s) = \frac{1}{\tau n_1} \sum_{j=1}^{\tau} \left[ \langle \mathbf{x}_{i,j}^{(k)}, \mathbf{x}_{i,j}^{(s)} \rangle - \frac{2\alpha}{n_2} \sum_{r=1}^{d} \langle \hat{\mathbf{x}}_{i,j}^{(k)}, \hat{\mathbf{x}}_{i,j}^{(r)} \rangle \langle \mathbf{x}_{i,j}^{(r)}, \mathbf{x}_{i,j}^{(s)} \rangle \right.$$
$$\left. + \frac{\alpha^2}{n_2^2} \sum_{l=1}^{d} \sum_{r=1}^{d} \langle \hat{\mathbf{x}}_{i,j}^{(l)}, \hat{\mathbf{x}}_{i,j}^{(s)} \rangle \langle \hat{\mathbf{x}}_{i,j}^{(k)}, \hat{\mathbf{x}}_{i,j}^{(r)} \rangle \langle \mathbf{x}_{i,j}^{(r)}, \mathbf{x}_{i,j}^{(l)} \rangle \right] \qquad (92)$$

As we can see, this is a polynomial in the independent Gaussian random variables $\{\hat{x}_{i,j}^{(r)}(q)\}_{j \in [\tau], r \in [d], q \in [n_2]}$ and $\{x_{i,j}^{(r)}(q)\}_{j \in [\tau], r \in [d], q \in [n_1]}$, for a total of $d\tau(n_1 + n_2)$ random variables. Moreover, $\frac{1}{\tau n_1} \sum_{j=1}^{\tau} \hat{Q}_{i,j}(k, s)$ is the average of $\tau$ i.i.d. random variables indexed by $j$. Define these random variables as $u_j$ for $j \in [\tau]$, i.e.,

$$u_j = \frac{1}{n_1} \langle \mathbf{x}_{i,j}^{(k)}, \mathbf{x}_{i,j}^{(s)} \rangle - \frac{2\alpha}{n_1 n_2} \sum_{r=1}^{d} \langle \hat{\mathbf{x}}_{i,j}^{(k)}, \hat{\mathbf{x}}_{i,j}^{(r)} \rangle \langle \mathbf{x}_{i,j}^{(r)}, \mathbf{x}_{i,j}^{(s)} \rangle$$
$$+ \frac{\alpha^2}{n_1 n_2^2} \sum_{l=1}^{d} \sum_{r=1}^{d} \langle \hat{\mathbf{x}}_{i,j}^{(l)}, \hat{\mathbf{x}}_{i,j}^{(s)} \rangle \langle \hat{\mathbf{x}}_{i,j}^{(k)}, \hat{\mathbf{x}}_{i,j}^{(r)} \rangle \langle \mathbf{x}_{i,j}^{(r)}, \mathbf{x}_{i,j}^{(l)} \rangle \qquad (93)$$

such that

$$\frac{1}{\tau n_1} \sum_{j=1}^{\tau} \hat{Q}_{i,j}(k, s) = \frac{1}{\tau} \sum_{j=1}^{\tau} u_j \qquad (94)$$

Then the variance of $\frac{1}{\tau n_1} \sum_{j=1}^{\tau} \hat{Q}_{i,j}(k, s)$ decreases linearly with $\tau$, since

$$\text{Var}\left( \frac{1}{\tau n_1} \sum_{j=1}^{\tau} \hat{Q}_{i,j}(k, s) \right) = \mathbb{E}\left[ \left( \frac{1}{\tau n_1} \sum_{j=1}^{\tau} \hat{Q}_{i,j}(k, s) - Q_i(k, s) \right)^2 \right]$$
$$= \mathbb{E}\left[ \left( \frac{1}{\tau} \sum_{j=1}^{\tau} u_j - \frac{1}{\tau} \sum_{j=1}^{\tau} \mathbb{E}[u_j] \right)^2 \right]$$
$$= \frac{1}{\tau^2} \sum_{j=1}^{\tau} \mathbb{E}[(u_j - \mathbb{E}[u_j])^2]$$
$$+ \frac{1}{\tau^2} \sum_{j=1}^{\tau} \sum_{j'=1, j' \neq j}^{\tau} \mathbb{E}[(u_j - \mathbb{E}[u_j])] \mathbb{E}[(u_{j'} - \mathbb{E}[u_j])] \qquad (95)$$
$$= \frac{1}{\tau} \mathbb{E}[(u_1 - \mathbb{E}[u_1])^2]$$
$$\leq \frac{1}{\tau} \mathbb{E}[u_1^2]$$

where (95) follows from the independence of the $u_j$'s. Next, recall the definition of $u_1$ given in (93):

$$u_1 = \frac{1}{n_1} \langle \mathbf{x}_{i,1}^{(k)}, \mathbf{x}_{i,1}^{(s)} \rangle - \frac{2\alpha}{n_1 n_2} \sum_{r=1}^{d} \langle \hat{\mathbf{x}}_{i,1}^{(k)}, \hat{\mathbf{x}}_{i,1}^{(r)} \rangle \langle \mathbf{x}_{i,1}^{(r)}, \mathbf{x}_{i,1}^{(s)} \rangle$$
$$+ \frac{\alpha^2}{n_1 n_2^2} \sum_{l=1}^{d} \sum_{r=1}^{d} \langle \hat{\mathbf{x}}_{i,1}^{(l)}, \hat{\mathbf{x}}_{i,1}^{(s)} \rangle \langle \hat{\mathbf{x}}_{i,1}^{(k)}, \hat{\mathbf{x}}_{i,1}^{(r)} \rangle \langle \mathbf{x}_{i,1}^{(r)}, \mathbf{x}_{i,1}^{(l)} \rangle \qquad (96)$$

The second moment of $u_1$ is dominated by the higher-order terms in $u_1^2$, since these terms have the largest expectation (being of the highest order) and are the most populous. To see that they are the most populous, note that $u_1$ has $d^2 n_1 n_2^2$ monomials with six random variables, and only $d n_1 n_2$ monomials with four random variables and $n_1$ monomials with two random variables. Thus $\mathbb{E}[u_1^2]$ is at most a constant times the expectation of the $d^4 n_1^2 n_2^4$ monomials of 12 variables in $u_1^2$. Each of

these monomials in 12 variables is the product of 12 one-dimensional, mean-zero Gaussian random variables, 8 corresponding to inner-loop samples and 4 corresponding to outer-loop samples. The maximum expected value of each of these monomials is thus the eighth moment of the inner loop random variable with maximum variance times the fourth moment of the outer-loop random variable with maximum variance. Since the variables are Gaussian with maximum variance $L$, the maximum expected value of each monomial is $105L^4 \times 3L^2 = 315L^6$. This yields our upper bound on the variance. The following analysis formalizes this argument:

$$
\mathrm{Var}\left( \frac{1}{\tau n_1} \sum_{j=1}^{\tau} \hat{Q}_{i,j}(k,s) \right)
$$

$$
\leq \frac{1}{\tau} \mathbb{E}[u_1^2]
$$

$$
= \frac{1}{\tau} \mathbb{E}\Bigg[ \Bigg( \frac{1}{n_1} \sum_{h=1}^{n_1} \Big( x_{i,1}^{(k)}(h) x_{i,1}^{(s)}(h) - \frac{2\alpha}{n_2} \sum_{r=1}^{d} x_{i,j}^{(r)}(h) x_{i,j}^{(s)}(h) \langle \hat{\mathbf{x}}_{i,j}^{(k)}, \hat{\mathbf{x}}_{i,j}^{(r)} \rangle
$$

$$
+ \frac{\alpha^2}{n_2^2} \sum_{l=1}^{d} \sum_{r=1}^{d} x_{i,j}^{(r)}(h) x_{i,j}^{(l)}(h) \langle \hat{\mathbf{x}}_{i,j}^{(h)}, \hat{\mathbf{x}}_{i,j}^{(s)} \rangle \langle \hat{\mathbf{x}}_{i,j}^{(k)}, \hat{\mathbf{x}}_{i,j}^{(r)} \rangle \Big) \Bigg)^2 \Bigg]
$$

$$
\leq \frac{c}{n_1^2 \tau} \mathbb{E}\Bigg[ \Bigg( \frac{\alpha^2}{n_2^2} \sum_{h=1}^{n_1} \sum_{l=1}^{d} \sum_{r=1}^{d} x_{i,j}^{(r)}(h) x_{i,j}^{(l)}(h) \langle \hat{\mathbf{x}}_{i,j}^{(l)}, \hat{\mathbf{x}}_{i,j}^{(s)} \rangle \langle \hat{\mathbf{x}}_{i,j}^{(k)}, \hat{\mathbf{x}}_{i,j}^{(r)} \rangle \Bigg)^2 \Bigg]
$$

$$
= \frac{c\alpha^4}{n_1^2 n_2^4 \tau} \mathbb{E}\Bigg[ \sum_{h=1}^{n_1} \sum_{h'=1}^{n_1} \sum_{l=1}^{d} \sum_{r=1}^{d} \sum_{l'=1}^{d} \sum_{r'=1}^{d} x_{i,j}^{(r)}(h) x_{i,j}^{(l)}(h) \langle \hat{\mathbf{x}}_{i,j}^{(l)}, \hat{\mathbf{x}}_{i,j}^{(s)} \rangle \langle \hat{\mathbf{x}}_{i,j}^{(k)}, \hat{\mathbf{x}}_{i,j}^{(r)} \rangle
$$

$$
x_{i,j}^{(r')}(h') x_{i,j}^{(l')}(h') \langle \hat{\mathbf{x}}_{i,j}^{(l')}, \hat{\mathbf{x}}_{i,j}^{(s)} \rangle \langle \hat{\mathbf{x}}_{i,j}^{(k)}, \hat{\mathbf{x}}_{i,j}^{(r')} \rangle \Bigg]
$$

$$
= \frac{c\alpha^4}{n_1^2 n_2^4 \tau} \mathbb{E}\Bigg[ \sum_{h=1}^{n_1} \sum_{h'=1}^{n_1} \sum_{l=1}^{d} \sum_{r=1}^{d} \sum_{l'=1}^{d} \sum_{r'=1}^{d} x_{i,j}^{(r)}(h) x_{i,j}^{(l)}(h) \Bigg( \sum_{q=1}^{n_2} \hat{x}_{i,j}^{(l)}(q) \hat{x}_{i,j}^{(s)}(q) \Bigg)
$$

$$
\Bigg( \sum_{q=1}^{n_2} \hat{x}_{i,j}^{(k)}(q) \hat{x}_{i,j}^{(r)}(q) \Bigg) x_{i,j}^{(r')}(h') x_{i,j}^{(l')}(h') \Bigg( \sum_{q=1}^{n_2} \hat{x}_{i,j}^{(l')}(q) \hat{x}_{i,j}^{(s)}(q) \Bigg) \Bigg( \sum_{q=1}^{n_2} \hat{x}_{i,j}^{(k)}(q) \hat{x}_{i,j}^{(r')}(q) \Bigg) \Bigg]
$$

$$
\tag{97}
$$

By independence, each term in the above summation has expectation zero unless (i) $r = l$ and $r' = l'$, or (ii) $r = r'$, $l = l'$ and $h = h'$. There are $n_1^2 d^2$ distinct values of $(h, h', r, r', l, l')$ that satisfy (i), and $n_1 d^2 \leq n_1^2 d^2$ distinct values of $(h, h', r, r', l, l')$ that satisfy (ii). For simplicity we treat each term in the summations over $n_2$ as having non-zero mean, although this can tightened. Since the expectation of each non-zero-mean monomial is at most $315L^6$, we have

$$
\mathrm{Var}\left( \frac{1}{\tau n_1} \sum_{j=1}^{\tau} \hat{Q}_{i,j}(k,s) \right) \leq \frac{730 c d^2 \alpha^4}{\tau} \tag{98}
$$

for some absolute constant $c$. With this bound on $\mathrm{Var}(\frac{1}{\tau n_1} \sum_{j=1}^{\tau} \hat{Q}_{i,j}(k,s))$, the result directly follows from Theorem 1.9 in Schudy & Sviridenko (2012), noting that $\frac{1}{\tau n_1} \sum_{j=1}^{\tau} \hat{Q}_{i,j}(k,s) - Q_i(k,s)$ is a degree 6 polynomial in centered independent Gaussian random variables. Note that by the argument in Schudy & Sviridenko (2012), the bound on the elements of $\mathbf{Z}_i$ given in Lemma 3 is tight up to logarithmic factors. □

Next, we show that this result implies a high probability bound on each $\|\mathbf{Z}_i\|$.

**Lemma 4.** *For all task indices $i$, any $\gamma > 0$, and some absolute constant $c$, the following holds:*

$$
\mathbb{P}\left( \|\mathbf{Z}_i\| \leq \gamma \right) \geq 1 - e^2 d^2 \exp\left( -\left( \frac{\tau \gamma^2}{c \alpha^4 d^2 L^6} \right)^{1/6} \right) \tag{99}
$$

*Proof.* By Lemma 3, we have a high-probability bound on the size of the elements of $\mathbf{Z}_i$.

Using this bound, we can bound the spectral norm of $\mathbf{Z}_i$ using the Gershgorin Circle Theorem, which says that every eigenvalue of $\mathbf{Z}_i$ lies within one of the Gershgorin disks $\mathfrak{D}_k$ defined by

$$\mathfrak{D}_k := \{\lambda : \lambda \in [Z_i(k,k) - R_k, Z_i(k,k) + R_k]\} \tag{100}$$

where $R_k = \sum_{s=1, s \neq k}^d |Z_i(k,s)|$. This implies that

$$\|\mathbf{Z}_i\| \leq \max_{k \in d} Z_i(k,k) + R_k \tag{101}$$

so we attempt to bound the RHS of the above. By a union bound over $k$, we have that for any $\gamma > 0$,

$$\mathbb{P}\left(R_k \geq (d-1)\gamma\right) \leq \mathbb{P}(\cup_{s \neq k}\{|\tilde{Q}_i(k,s) - Q(k,s)| \geq \gamma\})$$

$$\leq \sum_{s=1, s \neq k}^d e^2 \exp\left(-\left(\frac{\tau\gamma^2}{c\alpha^4 d^2 L^6}\right)^{1/6}\right)$$

$$= (d-1)e^2 \exp\left(-\left(\frac{\tau\gamma^2}{c\alpha^4 d^2 L^6}\right)^{1/6}\right) \tag{102}$$

where (102) follows from Lemma 3. We also have that

$$\mathbb{P}\left(|Z_i(k,k)| \geq \gamma\right) \leq e^2 \exp\left(-\left(\frac{\tau\gamma^2}{c\alpha^4 d^2 L^6}\right)^{1/6}\right) \tag{103}$$

again using Lemma 3. Combining (102) and (103) via a union bound yields for some particular $s$

$$\mathbb{P}\left(Z_i(k,k) + R_k \leq \gamma\right) \geq \mathbb{P}\left(\{R_k \leq (d-1)\gamma\} \cap \{Z_i(k,k) \leq \gamma\}\right)$$

$$= 1 - \mathbb{P}\left(\{R_k \geq (d-1)\gamma\} \cup \{Z(k,k) \geq \gamma\}\right)$$

$$\geq 1 - \mathbb{P}\left(\{R_k \geq (d-1)\gamma\}\right) - \mathbb{P}\left(\{Z(k,k) \geq \gamma\}\right)$$

$$= 1 - e^2 d \exp\left(-\left(\frac{\tau\gamma^2}{c\alpha^4 d^2 L^6}\right)^{1/6}\right) \tag{104}$$

Therefore, via a union bound over $k$, we have

$$\mathbb{P}\left(\max_{k \in [d]}\{Z_i(k,k) + R_k\} \leq \gamma\right) = 1 - \mathbb{P}\left(\cup_{k \in [d]}\{Z_i(k,k) + R_k \geq \gamma\}\right)$$

$$\geq 1 - e^2 d^2 \exp\left(-\left(\frac{\tau\gamma^2}{c\alpha^4 d^2 L^6}\right)^{1/6}\right) \tag{105}$$

for any $\gamma > 0$. As a result, using (101) we have

$$\mathbb{P}\left(\|\mathbf{Z}_i\| \leq \gamma\right) \geq 1 - e^2 d^2 \exp\left(-\left(\frac{\tau\gamma^2}{c\alpha^4 d^2 L^6}\right)^{1/6}\right) \tag{106}$$

$\square$

We continue with the proof of Theorem 3 by bounding $\|\mathbf{C}^{-1}\|$. Using an analogous argument as in the proof of Theorem 2 based on Weyl's Inequality (Theorem 4.5.3 in (Vershynin, 2018)), we have

$$\lambda_{\min}(\mathbf{C}) \geq \lambda_{\min}(\frac{1}{T}\sum_{i=1}^T \mathbf{Q}_i) - \|\mathbf{Z}_i\|$$

$$\geq \hat{\beta} - \gamma \tag{107}$$

where (107) follows by (86) and Lemma 4, with high probability, for any $\gamma > 0$. Thus choosing $\gamma < \hat{\beta}$, we have

$$\|\mathbf{C}^{-1}\| \leq \frac{1}{\hat{\beta} - \gamma} \tag{108}$$

Using this with (88) and Lemma 4, and using a union bound over $i \in [T]$, we have

$$\mathbb{P}\left(\vartheta \geq \frac{2\hat{L}B\gamma}{\hat{\beta}}\frac{1}{\hat{\beta}-\gamma}\right) \leq e^2 T d^2 \exp\left(-\left(\frac{\tau\gamma^2}{c\alpha^4 d^2 L^6}\right)^{1/6}\right) \tag{109}$$

Next, we turn to bounding the second term in (87), which is the error term due to the additive noise in the linear regression setting. We will do this by again making an argument based on the concentration of a polynomial in independent, centered Gaussian random variables around its mean. First, note that the term we must bound is

$$\left\|\left(\frac{1}{Tn_1\tau}\sum_{i=1}^{T}\sum_{j=1}^{\tau}\mathbf{P}_{i,j}(\mathbf{X}_{i,j}^{out})^\top \mathbf{X}_{i,j}^{out}\mathbf{P}_{i,j}\right)^{-1}\right.$$

$$\left.\left(\frac{1}{Tn_1\tau}\sum_{i=1}^{T}\sum_{j=1}^{\tau}\mathbf{P}_{i,j}(\mathbf{X}_{i,j}^{out})^\top \mathbf{z}_{i,j}^{out} - \frac{\alpha}{n_2}\mathbf{P}_{i,j}(\mathbf{X}_{i,j}^{out})^\top \mathbf{X}_{i,j}^{out}(\mathbf{X}_{i,j}^{in})^\top \mathbf{z}_{i,j}^{in}\right)\right\|$$

$$\leq \left\|\frac{1}{Tn_1\tau}\left(\sum_{i=1}^{T}\sum_{j=1}^{\tau}\mathbf{P}_{i,j}(\mathbf{X}_{i,j}^{out})^\top \mathbf{X}_{i,j}^{out}\mathbf{P}_{i,j}\right)^{-1}\right\|$$

$$\left\|\frac{1}{Tn_1\tau}\left(\sum_{i=1}^{T}\sum_{j=1}^{\tau}\mathbf{P}_{i,j}(\mathbf{X}_{i,j}^{out})^\top \mathbf{z}_{i,j}^{out} - \frac{\alpha}{n_2}\mathbf{P}_{i,j}(\mathbf{X}_{i,j}^{out})^\top \mathbf{X}_{i,j}^{out}(\mathbf{X}_{i,j}^{in})^\top \mathbf{z}_{i,j}^{in}\right)\right\| \tag{110}$$

$$= \left\|\mathbf{C}^{-1}\right\|\left\|\frac{1}{Tn_1\tau}\left(\sum_{i=1}^{T}\sum_{j=1}^{\tau}\mathbf{P}_{i,j}(\mathbf{X}_{i,j}^{out})^\top \mathbf{z}_{i,j}^{out} - \frac{\alpha}{n_2}\mathbf{P}_{i,j}(\mathbf{X}_{i,j}^{out})^\top \mathbf{X}_{i,j}^{out}(\mathbf{X}_{i,j}^{in})^\top \mathbf{z}_{i,j}^{in}\right)\right\| \tag{111}$$

where (110) follows from the Cauchy-Schwarz Inequality. Define

$$g(k) := \left[\frac{1}{Tn_1\tau}\left(\sum_{i=1}^{T}\sum_{j=1}^{\tau}\mathbf{P}_{i,j}(\mathbf{X}_{i,j}^{out})^\top \mathbf{z}_{i,j}^{out} - \frac{\alpha}{n_2}\mathbf{P}_{i,j}(\mathbf{X}_{i,j}^{out})^\top \mathbf{X}_{i,j}^{out}(\mathbf{X}_{i,j}^{in})^\top \mathbf{z}_{i,j}^{in}\right)\right](k) \tag{112}$$

for each $k \in [d]$, i.e. $g(k)$ is the $k$-th element of the $d$-dimensional vector $\left(\frac{1}{Tn_1\tau}\sum_{i=1}^{T}\sum_{j=1}^{\tau}\mathbf{P}_{i,j}(\mathbf{X}_{i,j}^{out})^\top \mathbf{z}_{i,j}^{out} - \frac{\alpha}{n_2}\mathbf{P}_{i,j}(\mathbf{X}_{i,j}^{out})^\top \mathbf{X}_{i,j}^{out}(\mathbf{X}_{i,j}^{in})^\top \mathbf{z}_{i,j}^{in}\right)$, and let $g := [g(k)]_{1 \leq k \leq d}$. Note that each $g(k)$ is a degree-6 polynomial in independent, centered Gaussian random variables. One can see that it is degree-6 because after expanding the $\mathbf{P}_{i,j}$ matrices, there are five data matrices and a noise vector (six total matrices) in the highest-order product. Also note that this polynomial has mean zero since the noise has mean zero and is independent of the data. Its variance $\mathbb{E}[(g(k) - \mathbb{E}[g(k)])^2]$ can be upper bounded using a similar argument as in Lemma 3. We will not write the full calculations and argument since they are very similar to those in Lemma 3. We again use the fact that the polynomial in question is the sum of $n_1\tau$ i.i.d. random variables to obtain variance decreasing linearly in $n_1\tau$. The differences are that in the highest-order monomial of $g(k)^2$, there are there are six inner-loop samples, four outer-loop samples, and two noise samples, for a maximum expectation of $15L^3 \times 3L^2 \times \sigma^2$. There are $T^2\tau^2 n_1^2 d^4 n_2^4$ of these terms, but as before, only $T^2\tau d^4 n_1^2 n_2^4$ of these terms have nonzero expectation due to independence, and the $n_1^2 n_2^4$ coefficient cancels. Thus, the variance is upper bounded as

$$\text{Var}(g(k)) \leq c\frac{\alpha^4 d^2 \sum_{i=1}^{T}\sum_{j=1}^{\tau} L^3\sigma^2}{T^2\tau^2} = c\frac{\alpha^4 d^2 L^5 \sigma^2}{\tau} \tag{113}$$

for some absolute constant $c$, which implies, via Theorem 1.9 in Schudy & Sviridenko (2012), that

$$\mathbb{P}(|g(k)| \geq b) \leq e^2 \exp\left(-\left(\frac{\tau b^2}{c\alpha^4 d^2 L^5 \sigma^2}\right)^{1/6}\right) \tag{114}$$

thus we have via a union bound over $k \in [d]$,

$$\mathbb{P}\left(\|g\| \leq \sqrt{d}b\right) \geq 1 - e^2 d \exp\left(-\left(\frac{\tau b^2}{c\alpha^4 d^2 L^5 \sigma^2}\right)^{1/6}\right) \tag{115}$$

for any $b > 0$. Combining this with (111) via a union bound, we have that

$$
\left\| \left( \frac{1}{Tn_1\tau} \sum_{i=1}^{T} \sum_{j=1}^{\tau} \mathbf{P}_{i,j}(\mathbf{X}_{i,j}^{out})^\top \mathbf{X}_{i,j}^{out} \mathbf{P}_{i,j} \right)^{-1} \right.
$$
$$
\left. \left( \frac{1}{Tn_1\tau} \sum_{i=1}^{T} \sum_{j=1}^{\tau} \mathbf{P}_{i,j}(\mathbf{X}_{i,j}^{out})^\top \mathbf{z}_{i,j}^{out} - \frac{\alpha}{n_2} \mathbf{P}_{i,j}(\mathbf{X}_{i,j}^{out})^\top \mathbf{X}_{i,j}^{out}(\mathbf{X}_{i,j}^{in})^\top \mathbf{z}_{i,j}^{in} \right) \right\|
$$
$$
\leq \frac{\sqrt{d}b}{\hat{\beta} - \gamma} \tag{116}
$$

with probability at least

$$
\geq 1 - e^2 d \exp\left( -\left( \frac{\tau b^2}{c\alpha^4 d^2 L^5 \sigma^2} \right)^{1/6} \right) - e^2 T d^2 \exp\left( -\left( \frac{\tau\gamma^2}{c\alpha^4 d^2 L^6} \right)^{1/6} \right) \tag{117}
$$

This completes the bound on $\theta$. Next, we must bound $\|\mathbf{w}_{MAML}^{(T)*} - \mathbf{w}_{MAML}^*\|$. We have

$$
\|\mathbf{w}_{MAML}^{(T)*} - \mathbf{w}_{MAML}^*\|
$$
$$
= \left\| \left( \frac{1}{T} \sum_{i=1}^{T} \mathbf{Q}_i \right)^{-1} \frac{1}{T} \sum_{i=1}^{T} \mathbf{Q}_i \mathbf{w}_i^* - \mathbb{E}_i[\mathbf{Q}_i]^{-1} \mathbb{E}_i[\mathbf{Q}_i \mathbf{w}_i^*] \right\|
$$
$$
= \left\| \left( \frac{1}{T} \sum_{i=1}^{T} \mathbf{Q}_i \right)^{-1} \frac{1}{T} \sum_{i=1}^{T} \mathbf{Q}_i \mathbf{w}_i^* - \left( \frac{1}{T} \sum_{i=1}^{T} \mathbf{Q}_i \right)^{-1} \mathbb{E}_i[\mathbf{Q}_i \mathbf{w}_i^*] \right.
$$
$$
\left. + \left( \frac{1}{T} \sum_{i=1}^{T} \mathbf{Q}_i \right)^{-1} \mathbb{E}_i[\mathbf{Q}_i \mathbf{w}_i^*] - \mathbb{E}_i[\mathbf{Q}_i]^{-1} \mathbb{E}_i[\mathbf{Q}_i \mathbf{w}_i^*] \right\|
$$
$$
\leq \left\| \left( \frac{1}{T} \sum_{i=1}^{T} \mathbf{Q}_i \right)^{-1} \frac{1}{T} \sum_{i=1}^{T} \mathbf{Q}_i \mathbf{w}_i^* - (\sum_{i=1}^{T} \mathbf{Q}_i)^{-1} \mathbb{E}_i[\mathbf{Q}_i \mathbf{w}_i^*] \right\|
$$
$$
+ \left\| \left( \frac{1}{T} \sum_{i=1}^{T} \mathbf{Q}_i \right)^{-1} \mathbb{E}_i[\mathbf{Q}_i \mathbf{w}_i^*] - \mathbb{E}_i[\mathbf{Q}_i]^{-1} \mathbb{E}_i[\mathbf{Q}_i \mathbf{w}_i^*] \right\| \tag{118}
$$
$$
\leq \left\| \left( \frac{1}{T} \sum_{i=1}^{T} \mathbf{Q}_i \right)^{-1} \right\| \left\| \frac{1}{T} \sum_{i=1}^{T} \mathbf{Q}_i \mathbf{w}_i^* - \mathbb{E}_i[\mathbf{Q}_i \mathbf{w}_i^*] \right\| + \left\| \left( \frac{1}{T} \sum_{i=1}^{T} \mathbf{Q}_i \right)^{-1} - \mathbb{E}_i[\mathbf{Q}_i]^{-1} \right\| \|\mathbb{E}_i[\mathbf{Q}_i \mathbf{w}_i^*]\| \tag{119}
$$
$$
= \frac{1}{\hat{\beta}} \left\| \frac{1}{T} \sum_{i=1}^{T} \mathbf{Q}_i \mathbf{w}_i^* - \mathbb{E}_i[\mathbf{Q}_i \mathbf{w}_i^*] \right\| + \left\| \left( \frac{1}{T} \sum_{i=1}^{T} \mathbf{Q}_i \right)^{-1} - \mathbb{E}_i[\mathbf{Q}_i]^{-1} \right\| LB \tag{120}
$$

where in equations (118) and (119) we have used the triangle and Cauchy-Schwarz inequalities, respectively, and in (120) we have used the dual Weyl's inequality and Assumption 2. We first consider the second term in (120). Continuing to argue as in the proof of Theorem 2, we have

$$
\|\mathbf{w}_{MAML}^{(T)*} - \mathbf{w}_{MAML}^*\| \leq \frac{1}{\hat{\beta}} \left\| \frac{1}{T} \sum_{i=1}^{T} \mathbf{Q}_i \mathbf{w}_i^* - \mathbb{E}_i[\mathbf{Q}_i \mathbf{w}_i^*] \right\| + \frac{\hat{L}B}{\hat{\beta}^2} \left\| \frac{1}{T} \sum_{i=1}^{T} \mathbf{Q}_i - \mathbb{E}_i[\mathbf{Q}_i] \right\| \tag{121}
$$

As in the proof of Theorem 2, will use the matrix Bernstein inequality (Theorem 6.1.1 in Tropp (2015)) to upper bound $\|\frac{1}{T} \sum_{i=1}^{T} \mathbf{Q}_i \mathbf{w}_i^* - \mathbb{E}_i[\mathbf{Q}_i \mathbf{w}_i^*]\|$ and $\|\frac{1}{T} \sum_{i=1}^{T} \mathbf{Q}_i - \mathbb{E}_i[\mathbf{Q}_i]\|$ with high probability. Doing so yields

$$
\|\mathbf{w}_{MAML}^{(T)*} - \mathbf{w}_{MAML}^*\| \leq \frac{1}{\hat{\beta}} \frac{\delta \operatorname{Var}(\mathbf{Q}_i \mathbf{w}_i^*)}{\sqrt{T}} + \frac{\hat{L}B}{\hat{\beta}^2} \frac{\delta \operatorname{Var}(\mathbf{Q}_i)}{\sqrt{T}} \tag{122}
$$

with probability at least $1 - 2d \exp\left( -\frac{\delta^2 \operatorname{Var}(\mathbf{Q}_i \mathbf{w}_{i,*})/2}{1 + \hat{L}\delta/(3\sqrt{T})} \right) - 2d \exp\left( -\frac{\delta^2 \operatorname{Var}(\mathbf{Q}_i)/2}{1 + \hat{L}B\delta/(3\sqrt{T})} \right)$ for any $\delta > 0$.

Combining (122) with our bound on $\theta$ (from (87), (109), and (116)) via a union bound and rescaling $\gamma = \gamma_{\text{new}} = \gamma_{\text{old}}\sqrt{\tau}$ yields

$$\|\mathbf{w}_{ERM} - \mathbf{w}^*_{ERM}\| \leq \frac{2\hat{L}B\gamma}{\hat{\mu}\sqrt{\tau}}\frac{1}{\hat{\mu} - \gamma/\sqrt{\tau}} + \frac{\sqrt{d}b}{\sqrt{\tau}}\frac{1}{\hat{\mu} - \gamma/\sqrt{\tau}} + \frac{1}{\hat{\beta}}\frac{\delta \text{ Var}(\mathbf{Q}_i\mathbf{w}^*_i)}{\sqrt{T}} + \frac{\hat{L}B}{\hat{\beta}^2}\frac{\delta \text{ Var}(\mathbf{Q}_i)}{\sqrt{T}}$$

$$(123)$$

with probability at least

$$1 - e^2 T d^2 \exp\left(-\left(\frac{c\gamma^2}{\alpha^4 d^2 L^6}\right)^{1/6}\right) - e^2 d \exp\left(-\left(\frac{c'b^2}{\alpha^4 d^2 L^5 \sigma^2}\right)^{1/6}\right)$$

$$- 2d \exp\left(-\frac{\delta^2 \text{ Var}(\mathbf{Q}_i\mathbf{w}_{i,*})/2}{1 + \hat{L}\delta/(3\sqrt{T})}\right) - 2d \exp\left(-\frac{\delta^2 \text{ Var}(\mathbf{Q}_i)/2}{1 + \hat{L}B\delta/(3\sqrt{T})}\right), \quad (124)$$

for some absolute constants $c$ and $c'$, and any $b$ and $\delta > 0$, and $\gamma \in (0, \sqrt{\tau}\hat{\mu})$.

Finally, choose $\gamma = 4d\sqrt{\frac{\alpha^4 L^6}{c}}\log^3(100Td)$, $b = 2d\sqrt{\frac{\alpha^4 L^5 \sigma^2}{c'}}\log^3(100d)$, and $\delta = (\sqrt{\text{Var}(\mathbf{Q}_i) + \text{Var}(\mathbf{Q}_i\mathbf{w}^*_i)})^{-1}\sqrt{\log(200d)}$ and restrict $\tau > 32d^2\alpha^4 L^6 \log^6(100Td)/(c\hat{\beta})$ such that $\frac{1}{\hat{\beta}-\gamma/\sqrt{\tau}} \leq \frac{2}{\hat{\beta}}$ and $T > \hat{L}^2 B^2 \log(200d)/(9(\text{Var}(\mathbf{Q}_i) + \text{Var}(\mathbf{Q}_i\mathbf{w}^*_i)))$. This ensures that each negative term in the high probability bound (124) is at most 0.01 and thereby completes the proof.

$\square$

