# OpenReview forum: "How Does the Task Landscape Affect MAML Performance?"
_ICLR.cc/2022/Conference — ICLR 2022 Submitted_

### Official Review · Reviewer_EaUw · 2021-10-31

**Correctness:** 3
**Technical Novelty And Significance:** 2
**Empirical Novelty And Significance:** 2
**Recommendation:** 5
**Confidence:** 3

**Details Of Ethics Concerns:**

No ethical concerns.

**Main Review:**

# Strengthes
1. I think the research topic is interesting and valuable. Understanding adaptive learning methods and their relationships/tradeoffs with the tasks/data is important. The topic is certainly relevant to the ICLR audience.

2. The text is clear and I can easily follow the story.

3. The authors provide the code and a comprehensive appendix for reproducibility.

4. I think the analysis of MAML at the beginning of Section 3.1 is interesting. I would be interested to see it extended for multiple tasks, with different degrees of hardness and possibly orthogonal relationships between them (i.e., $w_i$ orthogonal to $w_j$).

# Weaknesses
1. The setting seems to be a bit oversimplified. I believe that the authors should explain/justify why a linear setting is a useful framework to validate their ideas and to justify their theory. Why wouldn't it fail in a more complicated setting?

2. I suspect there might be an unfair advantage for the MAML model in the demonstrated comparison. It stems from the problem definition described at the beginning of Section 3. In this section, it is mentioned that each task $y_i$ is concentrated around a dot product $\langle w_{i,\*},xi\rangle$, where $w_{i,\*}$ is some ground-truth parameter associated with the $i$'th task. In addition, the MAML setting allows each task ($i$) to be solved by a function with a different parameter that depends on the data of the specific task. On the other hand, the NAL model does not enjoy the same degree of freedom and all of the tasks share the same solution w that is not dependent on the data of a specific task, rather the aggregation of all data. For instance, in the NAL case, we expect w to converge to their mean (assuming that $\Sigma_i=I$ for all $i$). In particular, if $w_{i,\*}$ are very far from each other, then, we do not really learn anything useful. It stems from the fact that NAL tries to model a function $h(x,i)=\langle w_{i,\*},x\rangle$ that depends on the tasks using a function that is not dependent on $i$.

Suggestions:
a. Emphasize why it is a fair/interesting comparison between the two models.
b. Reconsider the comparison with NAL and compare it with an adaptive method instead. For instance, in a typical transfer/meta-learning setting, we have a function $f:\mathbb{R}^d\to \mathbb{R}^k$ that is shared between the tasks and a specific classifier $g_i:\mathbb{R}^k\to\mathcal{Y}$ for each task individually (e.g., $g_i$=linear). Would it be possible to replace w in the NAL case with a function of the form $\langle v,Wx\rangle$, where $W$ is a matrix shared between tasks and v is solved using linear regression on $(Wx_{ij},y_{ij})^{n}_{j=1}$? If so, would the theory change? how?

4. Maybe I am missing something, in the proof of Lemma 1, what are $g_1$ and $g_2$? It seems that only $g_1(w_{MAML})$ and $g_2(w_{MAML})$ are defined but not $g_i(w)$. It makes it very hard to judge the soundness of this proof.

5. The $O(\alpha^2)$ in Theorem 1 (and Lemma 1) measures the scale w.r.t. alpha. Does it depend also on $\|w_1\|$ and $\|w_2\|$? If it may depend on these, the argument that $\|\nabla f_1(w_1)\| \leq \|\nabla f_2(w_2)\|$ could be false. This should be explained.

6. It seems that the task hardness is measured by the variance of the samples in the given task. Why is it referred to as the 'rate at which gradient descent converges to the optimal solution for the task'? I think it should be explained at least intuitively.

7. What does it mean: "NAL is trained equivalently to MAML with no inner loop"?
In MAML, we have a shared model $f_{w}$ and for each task we adapt w given the corresponding data.
How do we compute the predictions for each one of the tasks with NAL?
Suppose we have two binary classification tasks $T_1$ and $T_2$ that are exactly the same but the labels are switched, how $f_{w}$ handles them?

8. "The terms ‘Large’ and ‘Small’ describe the hypothesized size of $r_H$ in each experiment: the optimal solutions of hard tasks drawn
from ten (train and test) different alphabets are presumably more dispersed than hard tasks drawn from only two (train and test) similar alphabets. By our previous analysis, we expect MAML to achieve more gain over NAL in the Small $r_H$ setting."
Are there any experiments supporting this statement?

# Clarity/minor comments
1. "has large *energy* for easy tasks and small energy for hard tasks". -- what does energy mean here?
2. "after one step of gradient descent *from far away*, which is not true for hard tasks". -- unclear to me.
3. Maybe I am missing something, but I am not sure what is $\| \dots \|_{Q^{(m)}_i}$ in (6-7).
It seems to be a non-standard notation, which needs to be defined before the statements of (6-7).
4. What is the difference between $w_{i,\*}$ in the definition of $y_i$ and $w^*_i$ in (5)?
5. I think that the term 'geography' might not be the best choice. As far I as know, it is not a standard mathematical term.
6. What is the inequality in (29) standing for? Is it smaller up to multiplicative constant? If so, why do you need $O(\alpha)$ instead of $\alpha$?
7. Several undefined notations: $\sigma_{\min}, \tilde{\Omega}, \tilde{O}$.
8. In Theorem 2: is there a reason why you chose prob $\geq 0.96$ instead of a parameter delta as we typically see in standard generalization/concentration bounds? What would be the dependence on such delta?


**Summary Of The Paper:**

The authors aim to compare theoretically and empirically the ability of MAML and Non-Adaptive Learning (NAL) to different tasks in a multi-task setting, where tasks are sampled independently from the same distribution. They argue that MAML is better suitable for adapting to hard tasks, while NAL is unable to do so. Task hardness is measured by the noise of the samples in each task.

**Summary Of The Review:**

I believe that the overall direction is interesting and there is room to study which kind of adaptive learning method is better suitable for a different set of tasks (having many similar tasks/multiple hard and multiple easy tasks/etc). In addition, the paper makes an effort for laying some definitions of task hardness and tries to demonstrate different tradeoffs between adaptivity and hardness.

However, I believe that the comparison itself is not in the right place and MAML should be compared within a set of adaptive methods. I do not think it is impressive that MAML is more adaptive than the non-adaptive learning method, which does not seem to be practical. I also think that the definition of hardness is a bit oversimplistic (or at least not sufficiently motivated) as it measures the variance of the samples in the task.

---

> ### Author Response · Authors · 2021-11-16
> **Response to Reviewer EaUw**
>
> Thank you for your detailed feedback.
>
> 1. **Usefulness of linear setting.** Little is understood about MAML even for mixed linear regression. Moreover,
> recent work has demonstrated that the dynamics of gradient descent (GD) on overparameterized deep NNs can be effectively captured by the dynamics of GD on linear models, e.g. [1,2,3]. Therefore, it is important to thoroughly understand the linear regression case before moving on to more complex problems.
>
> 2. **Unfair comparison.**  In fact, both MAML and NAL must choose a single model. During training, they are both given the same amount of data to choose their $\mathbf{w}$. For these reasons, we believe the comparison is fair. Since MAML factors the test-time adaptation evaluation procedure into its training, it always does at least as well as NAL, the question is how much better? If the task solutions are far apart, isotropic around their mean, and $\mathbf{\Sigma}\_i = \mathbf{I}\_d$, then our analysis shows NAL and MAML perform equally poorly.
>
> Response to suggestions:
>
> We appreciate the thoughtful suggestions. For fixed $\mathbf{W}$, if $\mathbf{w}_{\ast,i}\in \text{row}(\mathbf{W})\forall i$, then the proposed approach can solve every task up the the noise floor, making the problem uninteresting. If $\mathbf{W}$ is fixed and we must learn a single $\mathbf{v}$, as in multi-task random feature regression, and $\mathbf{w}\_{\ast,i}\in \text{row}(\mathbf{W})\forall i$, then the analysis straightforwardly reduces to our current analysis with $\mathbf{\Sigma}_i$ replaced by $\mathbf{W}\mathbf{\Sigma}\_i\mathbf{W}^\top$. If the $\mathbf{w}\_{\ast,i}$'s do not belong to the row space of $\mathbf{W}$, then the tasks cannot be solved simultaneously (since $\mathbf{W}$ restricts the regressors for all tasks to belong to row($\mathbf{W}$)).
>
> Another case is when $\mathbf{W}$ can be learned, but as mentioned in first paragraph of page 2, the focus of our work is not on feature/representation learning, as this is studied in other works. Rather, we take a general approach to determining the benefit of MAML's distinctive feature (inner loop), for which the natural baseline is NAL.
>
> 4. **Typo regarding $\mathbf{g}\_i(\mathbf{w})$.** This line should read $\mathbf{g}\_1(\mathbf{w}):=\nabla f\_1(\mathbf{w})$ and $\mathbf{g}\_2(\mathbf{w}):=\nabla f\_2(\mathbf{w})$. We apologize for the confusion.
>
> 5. **Dependence on $||\mathbf{w}||$.** The Taylor expansion of $f_i(\mathbf{w} - \alpha \nabla f_i(\mathbf{w}))$ around $\mathbf{w}$ does not depend on the norm of $\mathbf{w}$, rather, it depends on higher-order derivatives of $f_i(\mathbf{w})$ and higher-order tensor products of the vector $\mathbf{w} - \alpha \nabla f_i(\mathbf{w}) - \mathbf{w} = -\alpha \nabla f_i(\mathbf{w})$.
>
> 6. **Why more data variance makes task easier.** More variance implies larger strong convexity parameter for the task's loss function (note that the Hessian is equal to $\mathbf{\Sigma}_i$) so the steeper the task's loss function and the faster GD converges. We explain this in the first paragraph on pg 5.
>
> 7. **NAL training.** Like MAML, on every iteration NAL samples a batch of tasks. Then, instead of executing an inner loop, NAL simply takes one step of minibatch SGD from the current iterate on each sampled task, then averages the updates to compute the next iterate. This corresponds to minibatch SGD on the NAL objective. We ensure MAML and NAL use the same number of samples during training.
>
> 8. **MAML's relative improvement for small $r_H$.** In Table 1, MAML's accuracy on the hard test tasks jumps from 81.2 to 95.4 when $r_H$ is small, while easy task performance and NAL performance is roughly the same regardless of $r_H$.
>
> Minor comments:
> 1. We mean the size of $\mathbf{\Sigma}_i$, i.e. the larger $\mathbf{\Sigma}_i$, the more weight that NAL places on the $i$-th task-optimal solution.
> 2. For easy tasks, the loss function is steep, so only one GD step may be sufficient to reach the optimal solution even if the initial point is far from the optimal solution (for fixed step size).
> 3. This is the Mahalanobis norm, i.e. $\|\mathbf{x}\|_\mathbf{S} := \sqrt{\mathbf{x}^\top \mathbf{S}\mathbf{x}}$ for symmetric matrix $\mathbf{S}$. We will explicitly define this in the revision.
> 4. This is a typo, it should read $\mathbf{w}_{i,\ast}$ in (5). We will fix this.
> 6. Thank you for noticing, this should be $\leq$, we will fix it.
> 7. Thank you for pointing this out, we will explicitly define them in the revision.
> 8. We choose a constant probability to simplify the high probability bound. Since we have exponential convergence (76), we would have a $\log(1/\delta)$ factor in the RHS of (44) as usual.
>
> [1] Jacot et al., Neural Tangent Kernel: Convergence and Generalization in Neural Networks, 2018.
>
> [2] Chizat et al., On lazy training in differentiable programming, 2018.
>
> [3] Mei and Montanari, The generalization error of random features regression: Precise asymptotics and double
> descent curve, 2019.

---

> > ### Comment · Reviewer_EaUw · 2021-11-29
> > **Thanks for the clarifications**
> >
> > Thanks for the clarifications. I appreciate the time and thoughtfulness responding my review.
> >
> > I agree with the authors that the comparison is valid in terms of the input for training the model. In that sense the analysisis not skewed. My main concern is whether the conclusions of this comparison are interesting or trivial. Since NAL is a very limited model, in my intuition, it is a bit obvious that MAML would perform better. NAL in a sense just performs some weighted average over the classifiers corresponding to the given tasks. This is non-adaptive by definition. MAML on the other hand had the capability to do something more adaptable by definition.
> >
> > Would be delighted to be convinced otherwise.

---

> > > ### Author Response · Authors · 2021-11-30
> > > **Response**
> > >
> > > Thank you for your reply. You are correct that NAL is non-adaptive by definition, and because of this it should never outperform MAML. However, what is surprising is that sometimes it performs just as well as MAML, for instance when all tasks are equally hard. This begs the question of when it makes sense to use MAML or use NAL to get similar performance at a cheaper cost. Our paper aims to answer this by illuminating the scenarios in which MAML achieves large gain over NAL and in which it does not, and quantify how much gain it achieves.
> > >
> > > We would appreciate any further feedback you would like to share.

---

### Official Review · Reviewer_g4W9 · 2021-11-01

**Correctness:** 4
**Technical Novelty And Significance:** 3
**Empirical Novelty And Significance:** 2
**Recommendation:** 6
**Confidence:** 3

**Main Review:**

**Strengths:**
- The authors analyze the effect of the hardness of task loss on the solution of MAML algorithm, and suggested situations where MAML gains advantage over NAL.
- The paper shows that MAML is going to pay more attention to the hard tasks to facilitate the boost of overall performance.
- The paper shows how the geometry of optimal solutions can have impact the effectiveness of MAML, which is novel

**Weaknesses:**
- The hardness of tasks is characterized by the eigenvalue of Hessian, which is a local quantity. This notion of task hardness seems to be difficult to generalize to other tasks with rough landscape and varying Hessian.
- The analysis shows that having the optimal solutions of hard problems packed together would benefit more from MAML, and in practice we do not get to know where are the optimal solutions for each task. The theoretical result does not seem to be able to guide practical applications of MAML effectively.
- It is not convincing enough why NAL is made into comparison as we do not often use NAL for meta learning.

**Correctness:**
- There is no false claims to the best of my knowledge.

**Clarity:**
- The arrangement of the paper is clear and structured.

**Additional Comments:**
- What are the possible suggestions that this theoretical conclusion could be able to make on applications of MAML tasks, given that the pool of tasks is often determined not by their hardness of each task but their usefulness in practice? Would adding more simple tasks beyond target tasks help the learning?

**Summary Of The Paper:**

This paper analyzes the performance of MAML algorithm under linear regression setting and compares that with the NAL under the same tasks. The authors show that the excess risk is smaller if there is more discrepency in the hardness of tasks and if the optimal solutions of hard tasks locate close to each other while those for easy tasks are far away. Related analysis and simulation results are also available for two-layer neural networks.

**Summary Of The Review:**

The conclusions of this paper are interesting, and some of the analyses come from a novel perspective. However, the contribution of this paper still remains in doubt as it is unclear how the insights can prove to be useful in practical applications.

---

> ### Author Response · Authors · 2021-11-16
> **Response to Reviewer g4W9**
>
> Thank you for your helpful feedback.
>
> **Local analysis in Section 4.** Indeed, our result only applies for a local region in which the Hessians are positive definite. Nevertheless, this type of local analysis is a standard first step for analyzing  many nonconvex problems (e.g. [1]). We need to operate in this region in order to come to any provable conclusions.
>
> **Practicality.** It is true that we do not get to know whether the hard task optimal solutions are packed together or not in practice, but we do get to make estimates of task similarity, which, even when estimated by semantic means, can be a good proxy for whether the optimal solutions are closely packed, as we demonstrate in our experiments in Section 5. Our work also suggests the merits of training on more hard tasks during training in order to find a model that is closer to the hard task solutions.
>
> **Why NAL.** We compare with NAL because it is a natural approach to solve multi-task learning problems (e.g. FedAvg in federated learning) and is equivalent to MAML without MAML's defining property (inner gradient step). Thus comparing with it exposes what makes MAML special compared to standard learning methods. We would like to note that prior work has demonstrated the competitiveness of NAL for meta-learning (e.g. [2,3,4]). Please also see our response to Reviewer oNs8.
>
> [1] Chi et al., Nonconvex Optimization Meets Low-Rank Matrix Factorization: An Overview, 2019.
>
> [2] Gao and Sener, Modeling and Optimization Trade-off in Meta-learning, 2020.
>
> [3] Bai et al., How Important is the Train-Validation Split in Meta-Learning?, 2021.
>
> [4] Bernacchia, Meta-Learning with Negative Learning Rates, 2021.

---

> > ### Comment · Reviewer_g4W9 · 2021-11-30
> > **Thank you for your reply**
> >
> > I appreciate the detailed reply from the authors and the response seems to be convincing to me. I will keep my original rating of the paper.

---

### Official Review · Reviewer_Y5kt · 2021-11-03

**Correctness:** 4
**Technical Novelty And Significance:** 3
**Empirical Novelty And Significance:** 3
**Recommendation:** 5
**Confidence:** 3

**Main Review:**

MAML is a widely-used meta-learning method that can quickly adapt to new tasks via one or a few stochastic gradient descent steps, so it is interesting and important to understand on which conditions MAML works well, and based on which, it may be further improved. The conditions provided in this paper are well supported by both numerical and analytical results in the linear regression setting. Since, in most scenarios, deep neural networks (DNNs) are used, the conditions would be more practically useful if the authors could provide more experimental results in the case where DNNs are applied.

One more suggestion: the analysis is based on the ratio of excess risk. It would be more convincing if the authors could explain why this measure is used and what are the other possible measures.


**Summary Of The Paper:**

This paper is about finding the conditions under which MAML outperforms standard multi-task learning. In particular, the authors focus on a linear regression setting and show that MAML outperforms NAL under the following two conditions: (i) there must be some discrepancy in hardness among the tasks, and (ii) the optimal solutions of the hard tasks must be closely packed with the center far from the center of the easy tasks optimal solutions. The authors also give numerical and analytical results in two-layer neural networks.


**Summary Of The Review:**

Interesting problem, solid numerical and analytical results in the linear regression setting, need more experimental results in the case where DNNs are used

---

> ### Author Response · Authors · 2021-11-16
> **Response to Reviewer Y5kt**
>
> Thank you for your positive feedback.
>
> **Choice of metric.** We compare the ratio of the excess risks in order to best highlight the relative performance of MAML and NAL.
> We could compare the values of the excess risk instead of studying their ratio but our conclusions would not change. We could compare the risks themselves, but this is not standard and would result in uncontrollable noise terms that would blur our vision of the discrepancy in performance between the two methods.

---

### Official Review · Reviewer_oNs8 · 2021-11-07

**Correctness:** 3
**Technical Novelty And Significance:** 2
**Empirical Novelty And Significance:** 2
**Recommendation:** 5
**Confidence:** 4

**Main Review:**

## Strengths
1. Gain of MAML over NAL based on the dispersion of hard tasks is interesting. However, I have concerns that the assumptions are too strong (see below).
2. Experimental comparison of MAML and NAL on real datasets is interesting. Particularly, it is interesting to see that MAML gets smaller error on hard tasks under some conditions from the theoretical results. However, this is not entirely surprising as NAL is never used in practice.
3. This is also validates the popular notion that MAML is better at adapting to hard tasks than baselines.

## Weaknesses
1. Sec 3.1:
a. $\rho_H/\rho_E(1-\rho_H/\rho_E)^2 \gg d/m$ is a very strong condition, even if $d$ is the effective dimension. This seems like a negative result (not that there is anything wrong with negative results), because when $d/m=o(1)$, each of the tasks may be separately solved using standard linear regression algorithm. For example, in Fig 1(b) with $d=10$ and $m=2000$, it would be interesting to know the average Excess risk of separately solving each of the tasks. This is an important baseline missing from all analysis and the experiments. I hope the authors can provide these in the next revision.

b. It is not clear what the hardness means. Why can’t we use different stepsize for each task during adaptation?

c. Why isn’t the main result collected into a well-defined theorem?

d. It is not clear if the same conclusion holds true when first coordinate of $\Sigma_i$ is $\alpha^{-1}$. In that case, for large $m$, $a_H = \rho_H (1-\alpha \rho_H)^2$, $a_E = \rho_E (1-\alpha \rho_E)^2$.

e. The problem considered seems too simple. For example, when the eigenvalues of covaraince matrices are different, or if the eigenvectors of the tasks do not match, the conclusions will differ.


2. Sec 4:
a. Thm 1: It is not clear why, when $m=\infty$, we have to use MAML instead of standard regression. This assumption on $m$ is too strong for the results to be interesting.

b. Interpretation: Since hard tasks have smaller Lipschitz constant than strong convexity of easy tasks, it is not too surprising that the gradient of the hard tasks are smaller than gradient of the easy tasks.

c. Interpretation: How are they the strong convexity and smoothness of the function? This needs to be derived.

d. What happens here when there are multiple tasks? Can we make similar comparison between MAML and NAL then?

e. Fig 4: Specific experimental details are missing.

f. Fig 4: Why does the Thm 1 and Fig 4 use different notion of hardness. In Thm 1, $\Sigma_i$ are identical, but later $W_i$ controls the hard

3. Sec 5: Current theory seems too simplistic to capture the reality of non-convex MAML. The non-convex experiment shows huge improvement of MAML over NAL, which is expected and known. However, in convex theory and experiments it is not clear whether the simple baseline of separately learning the tasks could be better or comparable to MAML.

4. Overall it is not clear why NAL is the only baseline that is compared with MAML.

## Minor comment
1. Pg 5, after eqn (8): Value of $a_H$ when settiing $\alpha$ is wrong.
2. Fg 2(c): Is $m=500$ or $m=100$?
3. Fig 1: What are the values of $R$, $r_H$, $r_E$ used?
4. Sec 4: Is the neural mapping a sum of activations or activation of the sum?
5. Interpretation: For ease of reading please prove that NAL gradients are the same for hard and easy tasks.


**Summary Of The Paper:**

This paper analytically compares the excess risk of two meta-learning methods: MAML and NAL. NAL is the simple baseline where the initialization for optimizing the test tasks is learned as parameter which minimizes the average loss of train tasks.

1. For a particular simple setting of linear regression, authors show that MAML achieves lower excess risk than NAL when (1) there is a considerable difference in hardness of the tasks, and (2) if the true parameter of the hard tasks are closer together relative to the distance between the mean of the hard and easy tasks.
2. For another case of simple neural network, where the output is the sum of many neurons outputs, authors show that MAML achieves smaller gradient for harder tasks than easier tasks under some condition.


**Summary Of The Review:**

Although the motivation of the paper is valid, it is not clear if the conclusion is interesting due to strong assumptions.

---

> ### Author Response · Authors · 2021-11-16
> **Response to Reviewer oNs8**
>
> Thank you for your thorough feedback.
>
> 1.a. **Learn separate model for each task.** With many samples/task, the tasks could in theory be solved separately. However, in meta-learning settings with many tasks that may be arriving dynamically, it may not be possible to learn separate models for each task due to computational constraints. Moreover, meta-learning analysis often studies the infinite samples/task case in order to make meaningful formal conclusions, e.g. [1,2].
>
> b. **Task hardness, step sizes.** Task hardness means the rate at which GD converges to solve the task. MAML uses the same adaptive step size for all tasks. As a result, this step size must be upper bounded by the inverse of the largest smoothness parameter across all tasks so that task-specific adaptation always converges.
>
> c. **Why not one theorem.** Our work provides multiple key insights, which we believe are more accessible when explained distinctly rather than condensed in one large theorem.
>
> d. **First coordinate of $\mathbf{\Sigma}_i$ is $\sigma^{-1}$.** Here, the task could be solved in the first dimension in one step. This does not change our conclusions. Footnote 1 addresses the issue of rescaling the data such that it has covariance $\sigma^{-1}\mathbf{I}_d$.
>
> e. **Data covariance assumption.** The assumption of scalar-times-identity covariance in Sec. 3.1-3.2 is for ease of presentation. Removing it does not fundamentally change conclusions or contribute new insights, but requires even more notation. We are not sure what is meant by ''eigenvectors of the tasks do not match'', so clarification would be helpful.
>
> 2. a. **Relevance of Sec. 4 population analysis.** The $m=\infty$ case yields insights about the finite-sample case, which is why it is studied in many works, e.g. in meta-learning [1,2].  Also, as mentioned earlier, even with many samples/task, it may not be possible to learn task-specific models in practice.
>
> b. **Theorem 1 not surprising.** While the assumptions of the theorem imply that most points have first task gradient norm less than second task gradient norm, they do not imply this for all points. So it remains non-trivial to prove this for $\mathbf{w}\_{MAML}$. For example, it is not true for $\mathbf{w}\_{NAL}$.
>
> c. **Smoothness and strong convexity params.**  We show these parameters bound the min and max singular values of the Hessian in the proof of Theorem 1 (Sec. B, pg 18). In the revision we will explicitly mention this in the main body.
>
> d. **Multi-task case for NNs.** We expect similar conclusions to hold, just as the linear two-task case generalizes to the multi-task case. Please see Fig. 4 and 6 which consider multi-task settings with NNs.
>
> e. **Fig. 4 details.** Please see Sec. C.3. for details. We will be more explicit in the main body in the revision.
>
> f. **Two hardness measures.** Both $\mathbf{\Sigma}\_i$ and the conditioning of $\mathbf{W}\_{\ast,i}$ affect task hardness in the 2-layer NN setting. Theorem 1 considers a setting in which the effects of $\mathbf{\Sigma}\_i$ are nullified, as they are all equal to one. Meanwhile, $M$ may be larger than one, so the conditioning of $\mathbf{W}\_{\ast,i}$ is important. In Fig. 4, $M=1$, so the condition number of $\mathbf{W}\_{\ast,i}$ is always 1 and does not impact task hardness, therefore varying $\mathbf{\Sigma}\_1$ controls task hardness.
>
> 3. **Relevance of analysis to Sec. 5 experiments.** Our analysis gives an explanation for the marginal MAML performance increases. Since little is known in this direction, our work provides a valuable early step in understanding MAML's behavior on complex models.
>
> 4. **Why only NAL.** Our objective is not to develop a new algorithm; the goal is to understand the role of adaptation in MAML. Thus, NAL is the only baseline compared with MAML. More specifically, to understand the role of adaptation in MAML, we must study why its inner loop confers performance benefits. This can only be done by comparing against the algorithm that is identical to MAML but lacks inner loop adaptation, i.e. NAL. Please also see our response to Reviewer g4W9.
>
> Minor
> 1. Thank you for noticing this, we will correct it ($\rho_E^3$ will become $\rho_E^2$).
> 2. The correct value is $m=100$ as stated in the figure and caption; we will fix the typo in the main body.
> 3.
>
> | | Large | Small |
> |-|-|-|
> |$R$  |  5 |  0 |
> |$r_H$ | 20 | 0.5 |
> |$r_E$ | 20 | 0.5 |
> We will include these in the revision.
>
> 4. Sum of neuron outputs; we will fix this in the definition of $f_i(\mathbf{W})$.
> 5. $\mathbf{W}$ is a stationary point of the NAL objective if and only if $\nabla F_\infty(\mathbf{W})=\mathbf{0} \iff \nabla f_1(\mathbf{W})+ \nabla f_2(\mathbf{W}) = \mathbf{0}$, which implies $||f_1(\mathbf{W})||_2 = ||f_2(\mathbf{W})||_2$. We will include this in the revision.
>
> [1] Saunshi et al., A Sample Complexity Separation between Non-Convex and Convex Meta-Learning, 2020.
>
> [2] Arnold et al., When MAML Can Adapt Fast and How to Assist When It Cannot, 2020.

---

> > ### Comment · Reviewer_oNs8 · 2021-12-03
> > **Follow-up**
> >
> > I thank the authors for their response. They have clarified some of my concerns and therefore I am increasing my score. However I am not fully convinced if the paper is ready to published at this venue due to the reasons detailed below.
> >
> > - Thanks for the clarification that with computational constraints it may not be possible to separately train each of the problems. This important point must be explicitly discussed in the next revision and tthe computational complexities should be compared.
> > - Lack of a well-defined theorem. Currently paper looks too hand-wavy.
> > - Paper assumes that MAML (with single adaptation step with constant stepsize) is needed for some problems and the only other baseline is NAL. This is a big assumption which requires more justification.
> > (a) Some of the baselines I can think of are meta-representation learning [A, 1] and transfer learning [B]. [1] shows that for convex models (like in this paper) MAML is not better than separately learning the tasks in the worst-case, but meta-representation learning (non-convex model) could be better. [B] shows that transfer learning may be better than MAML at MAML benchmarks.
> > (b) We could have also ran MAML with multiple steps [1] and with adaptive stepsizes like adagrad.
> >
> > - “eigenvectors of the tasks do not match”: I meant that eigenspace and eigenvalues of the two Hessians don’t match. I think authors can strengthen the paper by analyzing more general cases.
> >
> > - It is not clear how [1,2] justify infinite samples per task. There should be more discussion regarding why this choice was made.
> >
> > I highly encourage the authors to revise the paper to present their ideas better.
> >
> > [A] Javed et al., “Meta-Learning Representations for Continual Learning”, NeurIPS 2019
> >
> > [B] Dumoulin et al., “A Unified Few-Shot Classification Benchmark to Compare Transfer and Meta Learning Approaches”, NeurIPS 2021 Track on Datasets and Benchmark

---

> > > ### Author Response · Authors · 2021-12-06
> > > **Response**
> > >
> > > Thank you for considering our response and for your detailed feedback.
> > >
> > > - Regarding comparison with other baselines: Ultimately our aim is to determine when and why MAML is effective. To do so, we need to compare against the algorithm that does not make MAML's distinctive inner loop update. Unlike [1], we perform this analysis in the general setting which does not rely on the existence of a ground-truth representation existing among the tasks. In this case, it does not make sense to learn a representation. Considering the (meta-)representation learning case may be interesting, but as we mention in the introduction, it is not the focus of this work.
> > >
> > > - Thank you for the clarification on the case of eigenvectors not matching. We calculate the MAML and NAL solutions and excess risks for arbitrary covariance matrices, and then consider a special case in order to interpret these quantities. It is not clear why any of the insights gleaned from the specific case would not also apply to the general case with arbitrary covariance matrices or why mismatched eigenspaces should cause any fundamental change.
> > >
> > > - As in [1,2], we choose to study the case of infinite samples per task to uncover the underlying algorithm behavior in the ideal, noiseless case. Moreover, in the linear case, our analysis considers the finite-sample regime.
> > >
> > > Please let us know if we can further address your concerns.

---

### Decision · Program_Chairs · 2022-01-20

**Decision:**

Reject

**Comment:**

The paper compares MAML and NAL for meta-learning, and provides theoretical explanations on some very simple models when MAML can be significantly better than NAL, related to a definition of task hardness. The findings are also supported by experimental results.
While the results are plausible and can mark the starting point of a useful analysis, the models analyzed in the paper are too simplistic to warrant publication at ICLR. The authors are encouraged to extend their methodology to more complicated task models, as well as to, e.g., multi-step versions of MAML (since the considered version of MAML makes a single step, the proposed problem hardness may not be applicable in more general situations). It is also not clear how the derived insights can guide the practical applications of MAML.